# *Unlearning Isn't Deletion*: Investigating Reversibility of Machine Unlearning in LLMs

**Xiaoyu Xu** [1]  **Xiang Yue** [2]  **Yang Liu** [3]  **Qingqing Ye** [1]  **Huadi Zheng** [4]  **Peizhao Hu** [4]  **Minxin Du** [1]  **Haibo Hu** [1,5]

## Abstract

Unlearning in large language models (LLMs) aims to remove specified data, but its efficacy is typically assessed with task-level metrics like accuracy and perplexity. We show that these metrics can be misleading, as models can appear to forget while their original behavior is easily restored through minimal fine-tuning. This *reversibility* suggests that information is merely suppressed, not genuinely erased. To address this critical evaluation gap, we introduce a *representation-level analysis framework*. Our toolkit comprises PCA similarity and shift, centered kernel alignment (CKA), and Fisher information, complemented by a summary metric, the mean PCA distance, to measure representational drift. Applying this framework across multiple unlearning methods, data domains, and LLMs, we identify four distinct forgetting regimes based on their *reversibility* and *catastrophicity*. We compare recovery strategies and show that relearning efficiency relies on the data source. We also find that irreversible, non-catastrophic forgetting is exceptionally challenging. By probing unlearning limits, we identify a case of seemingly irreversible, targeted forgetting, offering insights for more robust erasure algorithms. Overall, our findings expose a gap in current evaluation and establish a representation-level foundation for trustworthy unlearning.

## 1. Introduction

Large language models (LLMs), trained on massive corpora, have achieved remarkable success across diverse tasks, yet their capacity to memorize training snippets poses acute ethical, legal, and security risks. Memorization can unintentionally disclose sensitive, harmful, or copyrighted text (Nasr et al., 2023; Karamolegkou et al., 2023; Wen et al., 2023), conflicting with emerging regulations, such as the EU's *Right to be Forgotten* (Ginart et al., 2019).

*Machine unlearning* seeks to address this challenge by algorithmically erasing the influence of specified data, making a model behave as if it had never been trained on that data (Bourtoule et al., 2021). While numerous unlearning methods have been developed for LLMs (Yao et al., 2024; Wuerkaixi et al., 2025; Li et al., 2024b;a; Xu et al., 2025), their efficacy is typically assessed using task-level metrics, such as accuracy on a held-out "forget set."

However, these evaluations overlook a pivotal question: ***Does LLM unlearning achieve genuine erasure, or merely suppress information that can resurface?*** If supposedly erased knowledge is readily revived, unlearning constitutes a shallow perturbation with limited safety.

Emerging evidence indicates that many unlearning methods are superficially effective. After unlearning, models often show degraded performance on the forget set; yet, the "forgotten" knowledge can be rapidly recovered through minimal fine-tuning even on unrelated data (Lo et al., 2024; Lynch et al., 2024) (see Figure 1), low-bit quantization (Zhang et al., 2025), or adversarial prompting (Patil et al., 2024; Liu et al., 2023). Although previous studies have identified this *reversibility* and the risks of catastrophic forgetting under accumulated updates (of repeated requests) (Shi et al., 2025), they primarily treat these issues as behavioral phenomena. The representational dynamics governing these regimes have yet to be investigated.

This paper presents the **first systematic, representation-level analysis of LLM unlearning reversibility**. We demonstrate that task-level metrics (*e.g.*, forget accuracy) are insufficient to distinguish reversible forgetting from catastrophic failure, as surface-level performance collapse may occur while internal representations remain intact. To move beyond surface effects, we introduce a unified diagnostic toolkit that jointly captures feature geometry, activation-subspace preservation, and parameter sensitivity. PCA sub-

---

[1]The Hong Kong Polytechnic University [2]Carnegie Mellon University [3]University of California, Santa Cruz [4]Huawei Technologies [5]Research Centre for Privacy and Security Technologies in Future Smart Systems, PolyU. Correspondence to: Minxin Du <minxin.du@polyu.edu.hk>, Haibo Hu <haibo.hu@polyu.edu.hk>.

*Proceedings of the 43rd International Conference on Machine Learning*, Seoul, South Korea. PMLR 306, 2026. Copyright 2026 by the author(s).

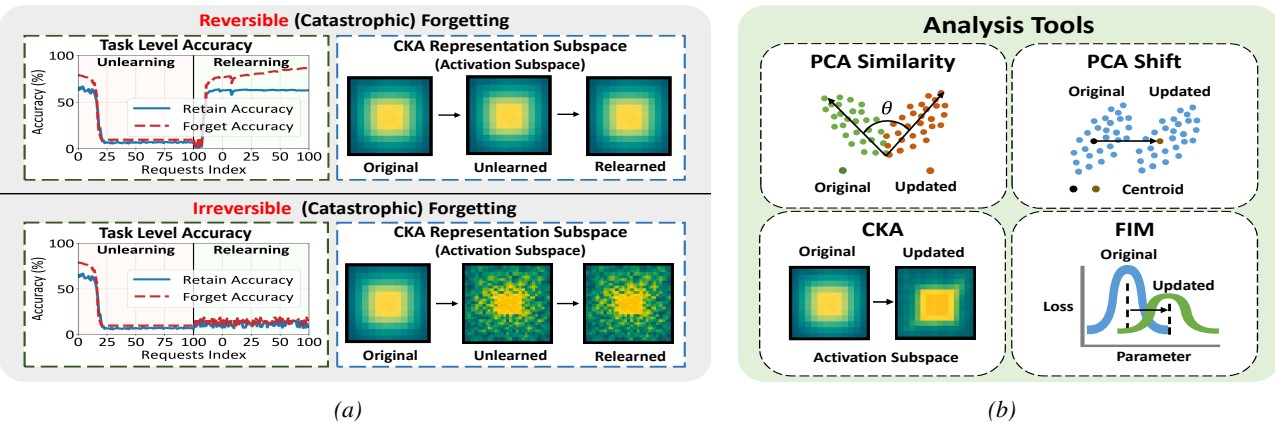

*Figure 1.* (a) task-level accuracy and CKA subspaces of **reversible** (top) vs. **irreversible** (bottom) catastrophic forgetting due to *continual unlearning* then *relearning*, (b) Our four diagnostic tools: PCA Similarity, PCA Shift, CKA, and FIM.

space similarity and shift (Zheng et al., 2025) measure directional alignment and translational drift, centered kernel alignment (CKA) (Kornblith et al., 2019) assesses activation-subspace preservation, and Fisher information (FIM) (Cha et al., 2025) tracks changes in the local loss landscape. We further introduce *mean PCA distance* as a compact measure of representational drift, helping reveal how different unlearning regimes emerge. With our toolkit, we build a taxonomy characterizing unlearning along two axes: reversibility and catastrophicity (collateral damage to retained knowledge). This allows us to distinguish four regimes:

1) *Reversible, Catastrophic:* Global performance collapse that is fully recoverable via relearning. 2) *Reversible, Non-Catastrophic:* Targeted performance modest degradation that is easily restored. 3) *Irreversible, Catastrophic:* Permanent and unrecoverable global performance collapse. 4) *Irreversible, Non-Catastrophic:* Ideally, permanent erasure of target data without collateral damage.

Crucially, we find that alternative relearning strategies such as prompt attacks (Patil et al., 2024), jailbreaking (Liu et al., 2023), quantization (Zhang et al., 2025), and in-context recovery (with five-shot demonstrations of the forget-set) "fail" once the model enters *reversible, catastrophic* forgetting. Since these methods involve minimal or no parameter updates (on unlearned models), they cannot restore the lost representations. Consequently, we employ relearning as a robust probe to investigate unlearning behavior. This approach allows us to unify single and continual unlearning under a single taxonomy, elucidating how distinct forgetting regimes emerge from request volume, hyperparameters, and the unlearning method itself. By further analyzing sample efficiency across data types, we conclude that genuine unlearning demands irreversible, non-catastrophic erasure rather than superficial degradation in task-level metrics.

We summarize our main contributions as follows: [1]

- We present the *first* systematic study of *reversibility* in both *single* and *continual* LLM unlearning. We introduce a representation-level diagnostic toolkit and a quantitative metric, the *mean PCA distance*, to analyze representational drift and distinguish four regimes of forgetting.

- We conduct extensive experiments across multiple unlearning methods, data domains, and LLM variants. Our results demonstrate that standard task-level metrics (*e.g.*, accuracy, perplexity, MIA susceptibility) are insufficient for assessing the true extent of unlearning. And we further find that relearning exhibits different sample efficiencies depending on the type of data.

- We provide a perturbation-based mechanistic view to explain how widespread vs. localized parameter changes relate to (ir)reversible forgetting. Small perturbations near the logits can distort task-level metrics despite intact features, hence leading to misleading assessments.

- We identify a case of *seemingly irreversible, non-catastrophic forgetting*, offering insights for designing more robust unlearning algorithms. We also highlight the potential for unlearning to serve as a form of data augmentation, improving model representations upon relearning.

## 2. Preliminaries and Our Formulation

**LLM unlearning** aims to remove the influence of specific data from a trained model to enhance privacy, improve safety, or mitigate bias (Yao et al., 2024; Jang et al., 2023; Wuerkaixi et al., 2025; Li et al., 2024b;a; 2025b). The standard paradigm involves a training corpus $\mathcal{D}$, from

---

[1]Our code is available at `https://github.com/XiaoyuXU1/Representational_Analysis_Tools`.

*Table 1.* Four regimes of forgetting characterized by the *reversibility* and *catastrophicity*: ● denotes regimes commonly observed in practice, and ◑ denotes the ideal but elusive regime.

| Regime | Obs. | Description |
|---|---|---|
| Reversible, Catastrophic | ● | Performance on both forget and retain sets collapses, but is recoverable via relearning. |
| Reversible, Non-Catastrophic | ● | Targeted performance drops on the forget set, which can be easily restored. |
| Irreversible, Catastrophic | ● | Global, unrecoverable performance collapse on both forget and retain sets. |
| Irreversible, Non-Catastrophic | ◑ | Targeted, permanent erasure of forget-set knowledge with no collateral damage. |

which $\mathcal{D}_f \subseteq \mathcal{D}$ is designated as the *forget set*. A model $\mathcal{M}$ is first trained on $\mathcal{D}$ via an algorithm $\mathcal{A}$. An unlearning procedure $\mathcal{U}$ then transforms $\mathcal{M}$ into *unlearned* $\mathcal{M}_f$, which should ideally behave as if it were trained only on the *retain set* $\mathcal{D}_r = \mathcal{D} \setminus \mathcal{D}_f$. Formally, the goal is to statistically approximate the retrained model $\mathcal{M}_r$: $\mathcal{M}_f = \mathcal{U}(\mathcal{M}, \mathcal{D}_f) \approx \mathcal{M}_r = \mathcal{A}(\mathcal{D}_r)$.

Retraining LLMs is prohibitively costly, so most studies rely on empirical proxies rather than formal statistically-indistinguishable guarantees (Maini et al., 2024; Li et al., 2024b; Gandikota et al., 2024). Evaluations track *forget quality* on the forget set, *utility*, and downstream task *accuracy* on the retain set, aiming to preserve both dimensions.

While current methods can achieve reasonable balances between forgetting and utility in *single-shot* scenarios (Barez et al., 2025; Gao et al., 2025), they often falter in the practical *continual* setting, where removal requests arrive sequentially over time (Barez et al., 2025). For a sequence of forget sets $\mathcal{D}_f^{(1)}, \mathcal{D}_f^{(2)}, \ldots, \mathcal{D}_f^{(t)}$ (the union is $\mathcal{D}_f$), the retain set is $\mathcal{D}_r^{(t)}$ after $t$ rounds. The model is then updated recursively: $\mathcal{M}_f^{(t)} = \mathcal{U}(\mathcal{M}_f^{(t-1)}, \mathcal{D}_f^{(t)})$, which should be similar to $\mathcal{M}_r^{(t)} = \mathcal{A}(\mathcal{D}_r^{(t)}), \forall t$. However, empirically, it often leads to *catastrophic forgetting*–a severe decline in performance on both forgotten and retained knowledge (Barez et al., 2025; Shi et al., 2025; Gao et al., 2025).

Single-shot unlearning is "fragile:" fine-tuning, even on benign, unrelated data, can rapidly restore the supposedly "forgotten" knowledge (Barez et al., 2025; Lynch et al., 2024; Lo et al., 2024). Such fragility persists in *continual* unlearning as well. Prior work has noted this phenomenon but has not deeply investigated its underlying mechanics.

### 2.1. A Taxonomy of Forgetting Regimes

We hypothesize that this performance collapse does not necessarily equate to true information erasure; the knowledge might merely become latent or suppressed. To formalize this hypothesis, we introduce a taxonomy of forgetting based on two axes: **catastrophicity** (the extent of collateral damage to retained knowledge) and **reversibility** (whether forgotten knowledge can be recovered).

Let $\theta_0$ be the initial model parameters, $\theta_u$ be the parameters after unlearning, and $\theta_r$ be the parameters after a subsequent *relearning* phase (defined below). We use $E(\theta, \mathcal{T})$ to denote a performance metric (*e.g.*, accuracy) evaluated on a task set $\mathcal{T}$, which can be partitioned into a forget-related task $\mathcal{T}_f$ and a retain-related task $\mathcal{T}_r$. We define four distinct regimes of forgetting, summarized in Table 1.

**Definition 2.1** (**Four Regimes of Forgetting**). Let $\Delta_u(\mathcal{T}) = E(\theta_0, \mathcal{T}) - E(\theta_u, \mathcal{T})$ be the performance drop after unlearning, and $\Delta_r(\mathcal{T}) = E(\theta_0, \mathcal{T}) - E(\theta_r, \mathcal{T})$ be the change after relearning. The nature of forgetting is determined by changes on forget set ($\mathcal{T}_f$) and retain set ($\mathcal{T}_r$).

**Catastrophic** vs. **Non-Catastrophic**: Forgetting is *catastrophic* if both $\Delta_u(\mathcal{T}_r)$ *and* $\Delta_u(\mathcal{T}_f) \gg 0$ and *non-catastrophic* otherwise ($\Delta_u(\mathcal{T}_r) \approx 0$).

**Reversible** vs. **Irreversible**: Forgetting is *reversible* if relearning almost recovers initial performance on forget set ($\Delta_r(\mathcal{T}_f) \approx 0$) and *irreversible* if a significant performance on the forget set drop persists ($\Delta_r(\mathcal{T}_f) \gg 0$).

The combination of these two properties yields four regimes, among which the *irreversible, non-catastrophic* forgetting is deemed ideal, but remains challenging to achieve in practice.

*Relearning Restriction.* Comparative analysis (see Appendix A.4.1) reveals that only relearning attacks reliably restore forgotten knowledge; we therefore employ relearning as our primary empirical probe to investigate forgetting regimes. To rigorously test the reversibility, we define a constrained relearning protocol that is distinct from full retraining. Given an unlearned model parameterized by $\theta_u$, we obtain the recovered model $\theta_r$ via brief fine-tuning on a restricted dataset, without access to the raw pre-training corpus. The relearning budget is strictly matched to the forget set size ($|\mathcal{D}_f|$), with data drawn from one of three sources: (i) the forget set $\mathcal{D}_f$ itself (representing a worst-case recovery scenario), (ii) a domain-aligned retain subset $\mathcal{D}_r^{(t)}$, or (iii) general out-of-distribution (or unrelated) data. Appendix A.4.2 shows a clear sample-efficiency hierarchy: relearning on the forget set recovers fastest and brings the unlearned model close to the original with less samples, while retain-set or unrelated relearning improves more slowly.

## 3. Classic (Task-Level) Evaluation Can Be Deceptive

### 3.1. Experiment setup

**Models and Datasets.** We adopt two open-source LLMs: Yi-6B (Young et al., 2024) and Qwen-2.5-7B (Yang et al., 2024). To assess the generality of our findings, we employ two distinct dataset types for unlearning: (i) *simple tasks*, comprising arXiv abstracts and GitHub code from (Yao et al., 2024), and (ii) a *complex task*, NuminaMath-1.5, a

*Table 2.* Yi-6B: MIA / F.Acc / R.Acc (%) simple task using four LRs under single unlearning, relearned by fine-tuning on the whole forget set once. Values in parentheses denote the change from the original model (red: negative, blue: positive).

| Phase | Method | LR=$3\times10^{-6}$ | | | LR=$4\times10^{-6}$ | | | LR=$5\times10^{-6}$ | | | LR=$6\times10^{-6}$ | | |
|---|---|---|---|---|---|---|---|---|---|---|---|---|---|
| | | MIA | F.Acc | R.Acc | MIA | F.Acc | R.Acc | MIA | F.Acc | R.Acc | MIA | F.Acc | R.Acc |
| Original | – | 70.9 | 78.9 | 65.5 | 70.9 | 78.9 | 65.5 | 70.9 | 78.9 | 65.5 | 70.9 | 78.9 | 65.5 |
| Unlearn | GA | 45.5 (-25.4) | 65.4 (-13.5) | 54.0 (-11.5) | 43.8 (-27.1) | 62.4 (-16.5) | 52.3 (-13.2) | 41.2 (-29.7) | 60.3 (-18.6) | 50.9 (-14.6) | 38.6 (-32.3) | 58.2 (-20.7) | 49.5 (-16.0) |
| | GA+GD | 65.4 (-5.5) | 75.1 (-3.8) | 64.6 (-0.9) | 58.2 (-12.7) | 73.8 (-5.1) | 65.8 (+0.3) | 55.3 (-15.6) | 68.5 (-10.4) | 63.5 (-2.0) | 52.4 (-18.5) | 63.2 (-15.7) | 61.2 (-4.3) |
| | GA+KL | 48.9 (-22.0) | 71.0 (-7.9) | 58.5 (-7.0) | 47.6 (-23.3) | 70.6 (-8.3) | 58.1 (-7.4) | 44.8 (-26.1) | 68.4 (-10.5) | 55.4 (-10.1) | 42.0 (-28.9) | 66.2 (-12.7) | 52.7 (-12.8) |
| | NPO | 67.2 (-3.7) | 76.2 (-2.7) | 64.7 (-0.8) | 65.2 (-5.7) | 75.8 (-3.1) | 62.8 (-2.7) | 62.2 (-8.7) | 75.2 (-3.7) | 62.7 (-2.8) | 59.2 (-11.7) | 74.6 (-4.3) | 62.6 (-2.9) |
| | NPO+KL | 66.5 (-4.4) | 76.3 (-2.6) | 64.8 (-0.7) | 67.2 (-3.7) | 76.4 (-2.5) | 63.2 (-2.3) | 64.5 (-6.4) | 75.6 (-3.3) | 61.2 (-4.3) | 61.8 (-9.1) | 74.8 (-4.1) | 59.2 (-6.3) |
| | RLabel | 69.6 (-1.3) | 77.7 (-1.2) | 64.7 (-0.8) | 69.2 (-1.7) | 76.5 (-2.4) | 64.5 (-1.0) | 68.7 (-2.2) | 75.4 (-3.5) | 63.3 (-2.2) | 68.2 (-2.7) | 74.3 (-4.6) | 62.1 (-3.4) |
| Relearn | GA | 67.2 (-3.7) | 76.6 (-2.3) | 65.2 (-0.3) | 68.6 (-2.3) | 77.6 (-1.3) | 62.8 (-2.7) | 67.6 (-3.3) | 76.9 (-2.0) | 65.5 (0.0) | 66.6 (-4.3) | 76.2 (-2.7) | 65.2 (-0.3) |
| | GA+GD | 68.6 (-2.3) | 77.0 (-1.9) | 65.3 (-0.2) | 68.8 (-2.1) | 76.9 (-2.0) | 65.3 (-0.2) | 68.8 (-2.1) | 77.2 (-1.7) | 65.3 (-0.2) | 68.8 (-2.1) | 76.5 (-2.4) | 65.3 (-0.2) |
| | GA+KL | 67.9 (-3.0) | 77.6 (-1.3) | 65.3 (-0.2) | 68.3 (-2.6) | 75.5 (-3.4) | 65.2 (-0.3) | 67.7 (-3.2) | 77.2 (-1.7) | 65.2 (-0.3) | 67.1 (-3.8) | 76.9 (-2.0) | 65.2 (-0.3) |
| | NPO | 68.2 (-2.7) | 77.1 (-1.8) | 65.3 (-0.2) | 68.3 (-2.6) | 77.2 (-1.7) | 65.2 (-0.3) | 68.3 (-2.6) | 77.0 (-1.9) | 65.1 (-0.4) | 68.4 (-2.5) | 76.8 (-2.1) | 65.0 (-0.5) |
| | NPO+KL | 68.9 (-2.0) | 77.1 (-1.8) | 65.3 (-0.2) | 67.9 (-3.0) | 76.3 (-2.6) | 63.0 (-2.5) | 68.6 (-2.3) | 76.9 (-2.0) | 65.2 (-0.3) | 69.3 (-1.6) | 77.5 (-1.4) | 65.4 (-0.1) |
| | RLabel | 68.3 (-2.6) | 78.8 (-0.1) | 65.6 (+0.1) | 68.9 (-2.0) | 76.4 (-2.5) | 65.3 (-0.2) | 68.8 (-2.1) | 78.9 (0.0) | 65.2 (-0.3) | 68.7 (-2.2) | 78.4 (-0.5) | 65.1 (-0.4) |

recent benchmark for mathematical reasoning (LI et al., 2024). All experiments are performed on NVIDIA H100 GPUs. (Additional results on TOFU (Maini et al., 2024) and the Traditional-Chinese corpus[2] are in Appendix A.5)

**Unlearning algorithms.** We primarily evaluate six canonical unlearning methods, organized into three families. To test whether our taxonomy generalizes across broader paradigms, we further include four representative methods: RMU (Li et al., 2024b), which steers representations toward random targets; UnDIAL (Dong et al., 2025), which performs unlearning via self-distillation with adjusted logits; AltPO (Mekala et al., 2025), which combines negative and positive feedback through preference optimization; and PDU (Entesari et al., 2025), which formulates unlearning as constrained primal-dual optimization. Their detailed results are reported in Appendix A.6 Table 15.

1) Gradient-Ascent (GA) family. The unified goal is $\mathcal{L} = \mathcal{L}_{\text{forget}}(\mathcal{D}_f) + \lambda\,\mathcal{L}_{\text{retain}}(\mathcal{D}_r)$, where $\mathcal{L}_{\text{forget}}$ maximizes the loss on the forget set via GA, $\mathcal{L}_{\text{retain}}$ preserves utility on the retain set, and $\lambda > 0$ balances the two. Choices for $\mathcal{L}_{\text{retain}}$ give three variants: i) GA ($\mathcal{L}_{\text{retain}} = 0$), ii) GA+GD (standard cross-entropy on $\mathcal{D}r$), and iii) GA+KL (KL divergence to the reference model on $\mathcal{D}r$) (Yao et al., 2024).

2) Negative Preference Optimization (NPO) family. GA is replaced by an NPO loss that penalizes agreement with the forget set (Zhang et al., 2024): $\mathcal{L} = \mathcal{L}_{\text{NPO}}(\mathcal{D}_f) + \lambda\,\mathcal{L}_{\text{retain}}(\mathcal{D}_r)$, Variants mirror those above: NPO ($\mathcal{L}_{\text{retain}} = 0$) and NPO+KL (retain-set KL regularization).

3) Random Label (RLabel). To mimic a model that never saw $\mathcal{D}_f$, true labels are replaced with random ones: $\mathcal{L} = \mathcal{L}_{\text{RLabel}}(\mathcal{D}_f)$, inducing near-uniform predictions without GA/negative rewards (Yao et al., 2024).

**Unlearning Scenarios.** We consider two standard settings: i) **Single unlearning:** A trained model $\mathcal{M}$ receives exactly one request to remove $\mathcal{D}_f \subset \mathcal{D}$, and ii) **Con-**

tinual unlearning: The model processes a stream of requests $\mathcal{D}_f^{(1)}, \dots, \mathcal{D}_f^{(t)}$, yielding a sequence of models where $\mathcal{M}_f^{(t)} = \mathcal{U}(\mathcal{M}_f^{(t-1)}, \mathcal{D}_f^{(t)})$. For *simple* tasks, we benchmark all six algorithms. For the *complex* math reasoning task, where a well-defined retain set is not available, we evaluate the core GA, NPO, and RLabel methods.

**Evaluation Metrics.** In *single*-step unlearning (unlearned only on simple tasks), we measure forget-set accuracy (F.Acc), retain-set accuracy (R.Acc), and privacy leakage via min-$k$%-prob MIA AUC (Shi et al., 2024).

In *continual* unlearning (both task types), we provide a more comprehensive evaluation. For simple tasks, we report: F.Acc / R.Acc, forget/retain perplexity (F.Ppl / R.Ppl), downstream accuracy on CommonsenseQA (CSQA) and $\text{GSM8K}_{\text{0-shot}}$ (Talmor et al., 2019; Cobbe et al., 2021), and min-$k$%-prob MIA AUC. For the complex task, we employ $\text{MATH}_{\text{0-shot}}$ (Hendrycks et al., 2021) and $\text{GSM8K}_{\text{0-shot}}$ as the primary math utility benchmarks.

**Relearning Setting.** To assess the *reversibility* of unlearning, each run is followed by a controlled *relearning* phase. The unlearned model is briefly fine-tuned on specific data without access to the full pre-training corpus. For **single unlearning**, we fine-tune once on the entire forget set $\mathcal{D}_f$. For **continual unlearning**, we evaluate three conditions: (i) the cumulative forget set $\bigcup_t \mathcal{D}_f^{(t)}$, representing a worst-case adversarial scenario, (ii) the corresponding retain subset $\mathcal{D}_r^{(t)}$, as a proxy for the data distribution, and (iii) unrelated out-of-distribution data (general-domain samples explicitly different from $\mathcal{D}_f$). Each relearning dataset is size-matched to its corresponding unlearning request.

**Hyperparameter Configuration.** To comprehensively evaluate the effects of unlearning, we design multiple hyperparameter configurations that vary both the learning rate and the number of unlearning requests. For single unlearning we sweep the learning rate over LR $\in \{3, 4, 5, 6\} \times 10^{-6}$ while fixing the request count to $N = 1$. For continual unlearning we vary both knobs: on the simple task (Yi-6B) we test LR $\in \{3, 5\} \times 10^{-6} \cup \{3 \times 10^{-5}\}$ with

---

[2] https://huggingface.co/datasets/taide/taide-bench

$N \in \{6 \rightarrow 100\}$; on the complex task (Qwen-2.5-7B) we use LR $\in \{3, 5\} \times 10^{-6}$ and $3 \times 10^{-5}$ together with $N \in \{6 \rightarrow 100\}$. All runs adopt the optimizer settings of (Touvron et al., 2023): AdamW (Loshchilov & Hutter, 2019) ($\beta_1 = 0.9$, $\beta_2 = 0.95$, $\varepsilon = 10^{-8}$), a cosine schedule with 10% warm-up followed by decay to 10% of peak, weight decay 0.1, and gradient clipping at 1.0.

### 3.2. Evaluation Results

We report quantitative results for single and continual unlearning on Yi-6B and Qwen-2.5-7B under various settings. Complete results are provided in Appendix Tables 6-7.

**Single Unlearning.** On Yi-6B, all six methods successfully reduce MIA and F.Acc, indicating a certain degree of forgetting (Table 2). The impact on the retain set is modest, with R.Acc dropping by only 2–5%. However, relearning often restores original performance; for instance, GA+KL and RLabel recover F.Acc to approximately 77%. These findings suggest that single unlearning achieves superficial forgetting, as the underlying representations remain largely intact (Section 4.2.1). This outcome characterizes the *reversible, non-catastrophic forgetting* regime.

**Continual Unlearning.** Post-relearning analysis (Tables 3, Appendix Table 6-7) reveals two forms of reversible forgetting. In *reversible, catastrophic forgetting*, both utility (*e.g.*, F.Acc, R.Acc) and privacy metrics drop sharply during unlearning but are fully restored after relearning. This is observed in GA and RLabel with moderate hyperparameters. Besides, *reversible, non-catastrophic forgetting* entails only a mild, easily recoverable performance degradation, as seen with NPO at LR $= 3 \times 10^{-5}$, $N = 6$.

Conversely, *irreversible, catastrophic forgetting* occurs when relearning fails to restore utility, leaving F.Acc and R.Acc low despite partial MIA recovery. This pattern is common for GA and RLabel under aggressive hyperparameters (*e.g.*, LR $= 3 \times 10^{-5}$, $N = 100$), where cumulative updates lead to irreversible representational collapse. The MIA AUC metric behaves erratically in this regime: it may fall below 50% during unlearning but misleadingly rebounds to high values after relearning, even after the model's capabilities have been permanently lost. These empirical results on single and continual unlearning are consistent with the theoretical framework in Section 5, which shows that small perturbations to the model weights can trigger disproportionately large drops in accuracy.

We conducted additional single-unlearning experiments on Qwen2.5-7B for the *complex task*, evaluating GA under two learning rates. We also performed continual-unlearning experiments ($N = 100$) on additional model variants for the *simple task*, including Llama-3-8B, Llama-3-8B-Instruct (Dubey et al., 2024), Qwen2.5-3B (Yang et al.,

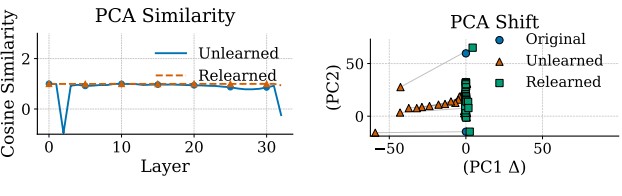

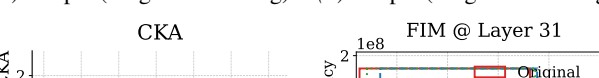

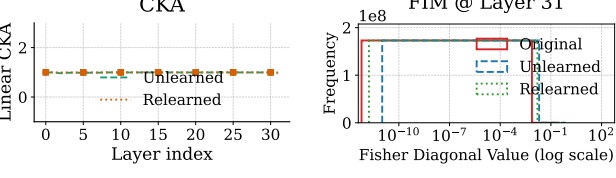

*(a)* Simple (Single Unlearning)     *(b)* Simple (Single Unlearning)

*(c)* Simple (Single Unlearning)     *(d)* Simple (Single Unlearning)

*Figure 2.* Single unlearning analysis on Yi-6B with GA under a simple task. In reversible non-catastrophic forgetting, PCA Similarity and Shift, CKA, and FIM across layers show only minor changes. Input queries are drawn from the forget set.

2024), Qwen3-8B-Base, and Qwen3-14B (Yang et al., 2025). Appendix Table 8 reports GA results under the same settings, while Appendix Tables 9 and 10 further include NPO results on Qwen3-14B and Llama-3-8B-Instruct. These results suggest that task-level metrics alone remain insufficient for reliably assessing unlearning reversibility across model scales and tuning variants. The observed behaviors are qualitatively consistent with those in Tables 3 and 2.

## 4. Representation-level Evaluation

### 4.1. Representational Analysis Tools

To analyze representational drift, we employ four complementary diagnostics, as summarized in Figure 1(b). PCA similarity and shift characterize feature-geometry changes, CKA measures activation-subspace preservation, and FIM reflects parameter sensitivity and local loss-landscape changes. Together, they help distinguish reversible suppression from irreversible representational collapse beyond task-level metrics. Precise definitions and implementation details are deferred to Appendix A.3.

**PCA Similarity, Shift, and Mean Distance.** For each layer $i$, we collect activation matrices $\mathbf{H}_i^{\text{orig}}$, $\mathbf{H}_i^{\text{unl}}$, and $\mathbf{H}_i^{\text{rel}}$ on a probe set $\mathcal{X}$ for the original, unlearned, and relearned models, respectively. Let $\mathbf{c}_{i,1}^{(*)}$ denote the PC1 direction at layer $i$ for state $(*) \in \{\text{orig}, \text{unl}, \text{rel}\}$, and let $p_{i,12}^{(*)}$ be the corresponding 2D drift coordinate in the PCA plane defined by the top-2 PCs of the original model. PCA Similarity is the cosine between $\mathbf{c}_{i,1}^{\text{orig}}$ and $\mathbf{c}_{i,1}^{(*)}$, while PCA Shift is the 2D drift coordinate $p_{i,12}^{(*)}$. Small values indicate stable representations, whereas large, unrecovered shifts suggest irreversible changes (Zheng et al., 2025). In our

*Table 3.* Yi-6B: MIA / F.Acc / R.Acc (%) for simple task under four unlearning settings. Bold numbers indicate improvements over the Original baseline in F.Acc or R.Acc. The relearning phase uses the cumulative forget set. Values in parentheses denote the change from the Original baseline (red: negative, blue: positive).

| Phase | Method | LR=$3\times10^{-5}$, N=100 | | | LR=$5\times10^{-6}$, N=100 | | | LR=$3\times10^{-6}$, N=100 | | | LR=$3\times10^{-5}$, N=6 | | |
|---|---|---|---|---|---|---|---|---|---|---|---|---|---|
| | | MIA | F.Acc | R.Acc | MIA | F.Acc | R.Acc | MIA | F.Acc | R.Acc | MIA | F.Acc | R.Acc |
| Original | —— | 70.8 | 78.9 | 65.5 | 70.8 | 78.9 | 65.5 | 70.8 | 78.9 | 65.5 | 70.8 | 78.9 | 65.5 |
| Unlearn | GA | 26.1 (-44.7) | 0.0 (-78.9) | 0.0 (-65.5) | 23.2 (-47.6) | 9.1 (-69.8) | 6.2 (-59.3) | 25.2 (-45.6) | 16.8 (-62.1) | 14.4 (-51.1) | 29.6 (-41.2) | 36.3 (-42.6) | 36.1 (-29.4) |
| | GA+GD | 16.8 (-54.0) | 9.7 (-69.2) | 2.3 (-63.2) | 28.7 (-42.1) | 3.6 (-75.3) | 3.1 (-62.4) | 69.4 (-1.4) | 78.8 (-0.1) | 65.5 (0.0) | 66.9 (-3.9) | 77.0 (-1.9) | 64.0 (-1.5) |
| | GA+KL | 17.8 (-53.0) | 9.0 (-69.9) | 6.2 (-59.3) | 27.3 (-43.5) | 9.1 (-69.8) | 6.2 (-59.3) | 18.9 (-51.9) | 3.8 (-75.1) | 3.2 (-62.3) | 29.5 (-41.3) | 52.9 (-26.0) | 41.5 (-24.0) |
| | NPO | 60.1 (-10.7) | 37.8 (-41.1) | 37.9 (-27.6) | 50.6 (-20.2) | 51.0 (-27.9) | 52.3 (-13.2) | 68.4 (-2.4) | 78.3 (-0.6) | 64.1 (-1.4) | 68.7 (-2.1) | 71.6 (-7.3) | 59.4 (-6.1) |
| | NPO+KL | 59.0 (-11.8) | 64.3 (-14.6) | 55.9 (-9.6) | 65.4 (-5.4) | 77.6 (-1.3) | 64.3 (-1.2) | 66.7 (-4.1) | 78.8 (-0.1) | 65.5 (0.0) | 67.9 (-2.9) | 67.6 (-11.3) | 56.1 (-9.4) |
| | RLabel | 65.1 (-5.7) | 0.0 (-78.9) | 0.0 (-65.5) | 63.6 (-7.2) | 0.1 (-78.8) | 0.4 (-65.1) | 61.4 (-9.4) | 0.4 (-78.5) | 0.7 (-64.8) | 62.7 (-8.1) | 72.7 (-6.2) | 61.1 (-4.4) |
| Relearn | GA | 74.5 (+3.7) | 2.1 (-76.8) | 1.8 (-63.7) | 68.0 (-2.8) | **80.0** (+1.1) | 65.0 (-0.5) | 68.6 (-2.2) | **80.8** (+1.9) | 65.2 (-0.3) | 68.2 (-2.6) | 70.5 (-8.4) | 58.7 (-6.8) |
| | GA+GD | 68.1 (-2.7) | 2.2 (-76.7) | 2.6 (-62.9) | 69.8 (-1.0) | **81.2** (+2.3) | 65.1 (-0.4) | 70.0 (-0.8) | **81.8** (+2.9) | 65.5 (0.0) | 67.0 (-3.8) | 61.6 (-17.3) | 54.4 (-11.1) |
| | GA+KL | 70.7 (-0.1) | 1.7 (-77.2) | 1.6 (-63.9) | 68.3 (-2.5) | **81.1** (+2.2) | 64.8 (-0.7) | 70.7 (-0.1) | **81.0** (+2.1) | 63.2 (-2.3) | 65.0 (-5.8) | 66.6 (-12.3) | 56.2 (-9.3) |
| | NPO | 70.0 (-0.8) | 57.0 (-21.9) | 45.6 (-19.9) | 68.0 (-2.8) | **82.7** (+3.8) | 65.5 (0.0) | 69.9 (-0.9) | **81.2** (+2.3) | 65.5 (0.0) | 68.4 (-2.4) | 71.2 (-7.7) | 59.4 (-6.1) |
| | NPO+KL | 67.7 (-3.1) | 60.7 (-18.2) | 54.2 (-11.3) | 69.5 (-1.3) | **83.8** (+4.9) | **65.6** (+0.1) | 69.9 (-0.9) | **83.8** (+4.9) | 65.5 (0.0) | 69.0 (-1.8) | 67.6 (-11.3) | 56.1 (-9.4) |
| | RLabel | 69.5 (-1.3) | 4.3 (-74.6) | 2.8 (-62.7) | 70.4 (-0.4) | **80.8** (+1.9) | 65.3 (-0.2) | 70.0 (-0.8) | **80.5** (+1.6) | 65.3 (-0.2) | 65.2 (-5.6) | 72.7 (-6.2) | 61.1 (-4.4) |

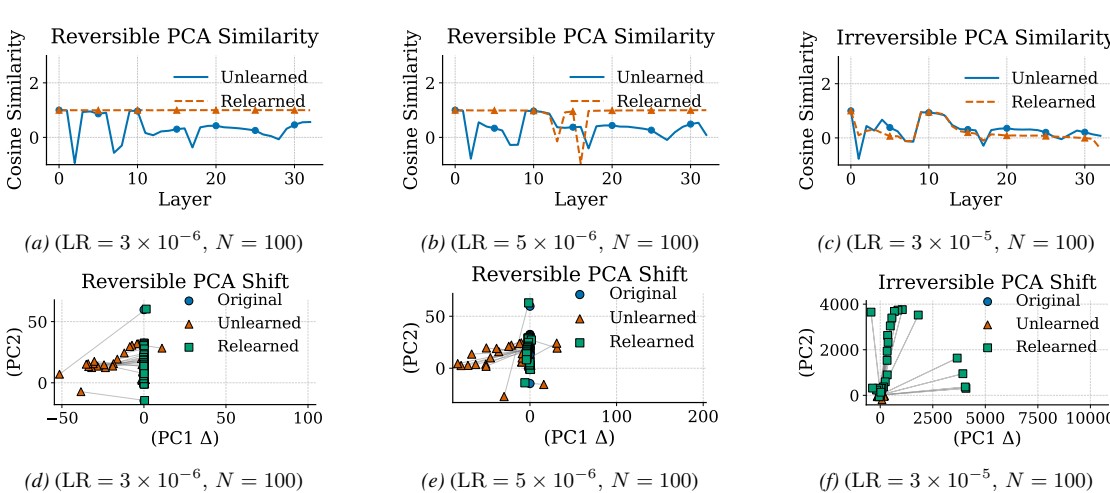

*(a)* (LR = $3 \times 10^{-6}$, $N = 100$)   *(b)* (LR = $5 \times 10^{-6}$, $N = 100$)   *(c)* (LR = $3 \times 10^{-5}$, $N = 100$)

*(d)* (LR = $3 \times 10^{-6}$, $N = 100$)   *(e)* (LR = $5 \times 10^{-6}$, $N = 100$)   *(f)* (LR = $3 \times 10^{-5}$, $N = 100$)

*Figure 3.* Layer-wise PCA Similarity and Shift for GA on Yi-6B (simple task). Vary LR $\{3 \times 10^{-6}, 5 \times 10^{-6}, 3 \times 10^{-5}\}$ at $N = 100$. Sustained low similarity or large shifts signal severe, irreversible catastrophic forgetting, whereas partial similarity or small shifts indicate mild, reversible catastrophic forgetting. Input queries are drawn from the forget set.

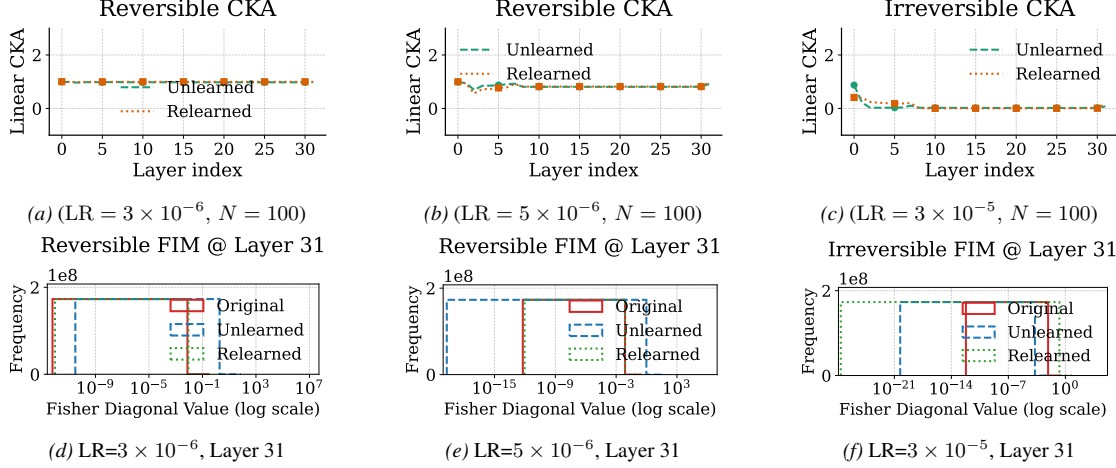

*(a)* (LR = $3 \times 10^{-6}$, $N = 100$)   *(b)* (LR = $5 \times 10^{-6}$, $N = 100$)   *(c)* (LR = $3 \times 10^{-5}$, $N = 100$)

*(d)* LR=$3 \times 10^{-6}$, Layer 31   *(e)* LR=$5 \times 10^{-6}$, Layer 31   *(f)* LR=$3 \times 10^{-5}$, Layer 31

*Figure 4.* CKA and FIM for GA on Yi-6B, simple task. Vary LR $\{3 \times 10^{-6}, 5 \times 10^{-6}, 3 \times 10^{-5}\}$ with $N = 100$. High CKA ($\approx 1$) and concentrated FIM spectra indicate reversible catastrophic forgetting, while persistently low CKA and large-shifted, flattened spectra denote severe representational drift and irreversible catastrophic forgetting. Input queries are drawn from the forget set.

visualizations, each PCA-shift point corresponds to one layer. We further report the *mean PCA distance* (PCA dist),

defined as the layer-averaged Euclidean distance between $p_{i,12}^{(*)}$ and $p_{i,12}^{\text{orig}}$, to summarize overall representation drift.

**Centered Kernel Alignment (CKA).** Given centered activation matrices $X_i^{\text{orig}}$ and $Y_i^{(*)}$, we compute $\text{CKA}(X_i^{\text{orig}}, Y_i^{(*)}) \in [0, 1]$. Values $\approx 1$ mean nearly identical subspaces, those $\approx 0$ are orthogonal.

**Fisher information (FIM).** We estimate the diagonal of the empirical FIM by averaging squared gradients over the probe set $\mathcal{X}$. Comparing $\text{FIM}^{\text{orig}}$, $\text{FIM}^{\text{unl}}$, and $\text{FIM}^{\text{rel}}$ reveals how unlearning alters the loss landscape and whether relearning restores parameter importance (Kirkpatrick et al., 2016; Hsu et al., 2022).

All diagnostics are computed not only on the forget set but also on the retain set and unrelated data to distinguish targeted unlearning from general representational degradation.

### 4.2. Representational Results

#### 4.2.1. SINGLE UNLEARNING

Figure 2 demonstrates feature-level changes under single unlearning. (a) PCA Similarity remains near 1.0 across all layers, with minor, reversible dips, indicating that dominant activation directions are preserved. Slight dips in shallow and final layers are rapidly restored after relearning, suggesting minimal and reversible drift. (b) PCA shifts are minimal, and relearned representations closely realign with the original. (c) CKA values are nearly 1.0 for all model states, confirming that subspace structures remain intact. (d) FIM spectra show only mild, temporary shifts that are fully restored after relearning. These results, combined with the task-level evaluation in Section 3.2, demonstrate that single unlearning induces *reversible, non-catastrophic forgetting*. This highlights the limitation of classic (task-level) metrics, which fail to capture the superficial nature of the forgetting.

#### 4.2.2. CONTINUAL UNLEARNING

As shown in Figures 3 and Appendix A.7 Figure 5, PCA Similarity and Shift offer complementary views of representational change: similarity reflects global alignment, and shift is sensitive to local variations. Relying on PCA similarity alone can obscure subtle effects; employing both avoids overlooking fine-grained distinctions, enabling a more comprehensive assessment. Higher learning rates or more requests cause sharp drops in similarity and large, unrecovered shifts, characteristic of *irreversible catastrophic forgetting*. In contrast, milder hyperparameters lead to high similarity and bounded shifts that are restored after relearning, consistent with *reversible, catastrophic forgetting*. This pattern is consistent across probe sets from forget set, retain set, and unrelated data. (Appendix A.7 Figures 10 and 14).

Figure 4 and Appendix A.7 Figure 6 integrate CKA and FIM analyses. CKA reveals that mild unlearning maintains stable alignment that recovers post-relearning, while aggressive unlearning causes irreversible degradation. The FIM spectra complement this by showing that continual unlearning flattens the loss landscape. Extreme hyperparameters induce a permanent leftward shift in sensitivity distributions, whereas moderate settings permit recovery. Together, these diagnostics suggest that observed performance loss is often due to temporary suppression rather than permanent erasure of knowledge. For conciseness, we present results on forget-set queries in the main text; retain-set queries, which yield similar conclusions under catastrophic forgetting, are in Appendix A.7 (*e.g.*, Figures 18 and 34).

**Mean PCA Distance Analysis.** To summarize representation-level drift with a single scalar metric, we use the *mean PCA distance*. We report its mean, standard deviation, and 95% confidence intervals across four random seeds, while also shuffling the order of unlearning requests to assess robustness to request ordering. As shown in Table 4, higher learning rates consistently lead to larger PCA distances for both unlearned and relearned models, indicating stronger and less recoverable representational drift. At lower learning rates (*e.g.*, $3 \times 10^{-6}$ and $5 \times 10^{-6}$), mean PCA distances remain small with limited variance, suggesting stable and reproducible recovery toward the original representation space. In contrast, at the high learning rate ($3 \times 10^{-5}$), PCA distances remain large even after relearning, with substantially higher variability, reflecting persistent representational distortion and incomplete recovery.

Table 4 further compares task-level metrics with mean PCA distance under identical GA continual-unlearning settings. Task scores alone are not sufficiently discriminative: both reversible and irreversible cases can exhibit severe post-unlearning collapse, and their distinction becomes clear only after relearning. By contrast, mean PCA distance provides an informative pre-relearning signal. Large pre-relearning distances at $3 \times 10^{-5}$ predict poor recovery and irreversible catastrophic forgetting, whereas smaller distances at $5 \times 10^{-6}$ and $3 \times 10^{-6}$ correspond to strong recovery and reduced representational drift. These results suggest that mean PCA distance captures recoverability information beyond task-level metrics, consistent with the perturbation view in Section 5: small perturbations induce limited feature-space changes, while larger accumulated perturbations lead to persistent drift and irreversibility.

## 5. Theoretical Analysis

### 5.1. A Perturbation Model of Unlearning

To interpret the empirical distinction between *reversible* and *irreversible* catastrophic forgetting, we view unlearning as a

*Table 4.* Yi-6B (GA) continual-unlearning results under different learning rates. We report task-level accuracy, mean PCA distance on forget and retain sets, and 95% confidence intervals across four random seeds. Confidence intervals are computed using a normal approximation and may slightly cross zero for small-distance cases.

| Model | Learning Rate | Phase | F.Acc | R.Acc | PCA dist (forget) | 95% CI (forget) | PCA dist (retain) | 95% CI (retain) |
|---|---|---|---|---|---|---|---|---|
| **Yi-6B (GA)** | $3 \times 10^{-5}$ | Unlearn | $0.05 \pm 0.02$ | $133.20 \pm 45.81$ | $[60.31, 206.09]$ | $121.45 \pm 38.60$ | $[60.03, 182.87]$ | |
| | $3 \times 10^{-5}$ | Relearn | $2.10 \pm 0.83$ | $1.80 \pm 0.73$ | $104.58 \pm 39.70$ | $[41.41, 167.75]$ | $95.34 \pm 32.40$ | $[43.78, 146.90]$ |
| | $5 \times 10^{-6}$ | Unlearn | $9.10 \pm 2.50$ | $6.20 \pm 2.07$ | $11.52 \pm 6.19$ | $[1.67, 21.37]$ | $8.79 \pm 5.20$ | $[0.52, 17.06]$ |
| | $5 \times 10^{-6}$ | Relearn | $80.00 \pm 1.21$ | $65.00 \pm 0.65$ | $1.37 \pm 0.74$ | $[0.19, 2.55]$ | $1.05 \pm 0.58$ | $[0.13, 1.97]$ |
| | $3 \times 10^{-6}$ | Unlearn | $16.80 \pm 3.07$ | $14.40 \pm 3.04$ | $9.62 \pm 5.66$ | $[0.61, 18.63]$ | $6.85 \pm 4.10$ | $[0.33, 13.37]$ |
| | $3 \times 10^{-6}$ | Relearn | $80.80 \pm 1.10$ | $65.20 \pm 0.80$ | $2.11 \pm 1.42$ | $[-0.15, 4.37]$ | $1.64 \pm 1.12$ | $[-0.14, 3.42]$ |

sequence of layer-wise perturbations that may alter representations across the network. For clarity, consider an $L$-layer feed-forward network $f(x) = \sigma(W_L \sigma(\cdots \sigma(W_1 x)\cdots))$, with activation function $\sigma$ and weights $\{W_i\}_{i=1}^L$. We model the unlearned model as a perturbed network with $\widetilde{W}_i = W_i + E_i$, where $E_i$ denotes the update induced by unlearning at layer $i$. The magnitude and spread of these perturbations are expected to increase with the learning rate and the number of unlearning requests. Expanding the perturbed computation into terms involving one or more layer-wise perturbations yields the following intuition:
$\widetilde{f}(x) - f(x) \approx \sum_{i=1}^L (W_L \circ \cdots \circ E_i \circ \cdots \circ W_1)(x) + \sum_{i<j} (W_L \circ \cdots \circ E_j \circ \cdots \circ E_i \circ \cdots \circ W_1)(x) + \cdots$.

When perturbations are small or localized, the change is dominated by low-order terms, so task-level behavior may be distorted while internal representations remain largely recoverable. In contrast, large or distributed perturbations across many layers can accumulate through higher-order interactions, leading to persistent representational drift and irreversible catastrophic forgetting. We can formalize the impact on our diagnostic tools:

**PCA Similarity.** Let $X_i$ and $Y_i = X_i + E_i'$ be the centered activations at layer $i$ before and after unlearning. By the Davis–Kahan theorem (Davis & Kahan, 1970), $\cos \angle(\mathbf{c}_i^{\text{orig}}, \mathbf{c}_i^{\text{upd}}) \approx 1 - O(\|E_i'\|/(\lambda_{1,i} - \lambda_{2,i}))$, with top two eigenvalues $\lambda_{1,i}, \lambda_{2,i}$. The layer-averaged PCA similarity is $\bar{S}_{\text{PCA}} \approx 1 - O((1/L) \sum_i \|E_i'\|)$.
**PCA Shift.** Along the first principal component, the activation-centroid shift is expressed as $p_{i,12} = O(\|E_i'\|)$. Large perturbations $\|E_i'\|$ propagating across multiple layers lead to *irreversible* representational drift, whereas smaller perturbations remain localized and thus *reversible*.
**CKA.** Let $\widetilde{K}_{Y_i} = \widetilde{K}_{X_i} + \Delta K_i$ denote the perturbed Gram matrix at layer $i$. The corresponding CKA score is computed as $\text{CKA}_i = 1 - O\left(\|\Delta K_i\|_*/\|\widetilde{K}_{X_i}\|_*\right)$. Averaging across layers yields $\bar{C} \approx 1 - O\left(\frac{1}{L} \sum_i \|\Delta K_i\|_*\right)$, where $\bar{C}$ denotes the layer-averaged CKA.
**Fisher Information.** Given update $\delta w_i = O(\|E_i\|)$, the Fisher diagonal behaves as $F_{ii}(w + \delta w) = F_{ii}(w) + O(\|\delta w_i\|)$, so the average Fisher becomes $\bar{F} =$

$(1/P) \sum_i F_{ii} = F_0 - O((1/P) \sum_i \|E_i\|)$.

## 5.2. Bridging Representational Drift and Task-Level Metrics

Classic (task-level) metrics can be misleading. They are highly sensitive to small weight changes, particularly in the final layers, which can cause large shifts in output probabilities without altering the model's deeper representations. For a softmax output, a small perturbation $\delta\theta$ to the parameters yields a large change in log-probability: $\log p(y|x; \theta + \delta\theta) \approx \log p(y|x; \theta) + \nabla_\theta \log p(y|x; \theta)^\top \delta\theta + O(|\delta\theta|^2)$. A minor update to the logits can dominate this first-order term, causing a sharp drop in accuracy that suggests catastrophic forgetting, even if the underlying geometry is preserved.

This aligns our theoretical model with the empirical findings in Sections 3 and 4.2. When LR or $N$ is small, changes are confined to first-order effects, feature spaces remain intact, and forgetting is *reversible*. When LR or $N$ is large, higher-order perturbations accumulate across layers, making recovery impossible and leading to *irreversible* forgetting. Figure 3 illustrates such a transition.

Interestingly, relearning can sometimes yield performance that *exceeds* the original model's accuracy on the forget set (Table 3). This suggests unlearning can act as contrastive regularization, reinforcing salient features tied to the forgotten data, which brief relearning can then exploit.

## 5.3. A Failure Mode of Probability-Based MIA

Probability-based membership inference metrics can suffer from similar output-level sensitivity. Recent studies show that probability-based MIAs, such as min-$k$%-prob (Shi et al., 2024), are sensitive to distributional mismatch and may conflate probability shifts with memorization (Duan et al., 2024; Meeus et al., 2025), while stronger shadow-model attacks such as LiRA are often impractical for LLM-scale evaluation (Carlini et al., 2022). For an input sequence $x = (x_1, \ldots, x_T)$, min-$k$%-prob first selects the subset $\mathcal{I}_k(x)$ containing the $k$% token positions with the lowest conditional probabilities, and then computes their average log-probability: min-k% -prob$(x) =$

$\frac{1}{|\mathcal{I}_k(x)|} \sum_{i \in \mathcal{I}_k(x)} \log p_\theta(x_i \mid x_{<i})$. A higher score indicates that even the least likely tokens are assigned relatively high probability, which is often treated as evidence that the sequence is closer to the training distribution. However, the resulting AUC only measures the relative separability between forget and retain examples, rather than whether the forget-set knowledge has been truly deleted.

Under aggressive unlearning, output probabilities can collapse globally, making the min-k% statistic unstable and difficult to interpret. During relearning, forget-set probabilities may recover faster than retain-set probabilities, recreating a probability gap between the two sets. This asymmetric recovery can artificially increase MIA AUC, reflecting restored statistical separability rather than durable erasure.

### 5.4. Probing the Limits of Irreversibility

In our primary experiments, we did not observe *irreversible non-catastrophic forgetting*; even a small fraction (*e.g.*, 10%) of the forget set was sufficient to restore performance. To explore this regime, we conducted extra experiments with more constrained relearning conditions. We used the GA+GD+WAGLE method (Jia et al., 2024), which selectively updates influential parameters, and limited the relearning data to either (i) 50% of the retain set or (ii) an equal-sized, unrelated dataset (Table 5).

Under these conditions, the method exhibited *seemingly irreversible, non-catastrophic forgetting*. The forget set showed large, unrecoverable PCA distances, while the retain set experienced only modest, partially recoverable degradation. This demonstrates that achieving the ideal of targeted, permanent unlearning without collateral damage remains an open challenge. Defining precise thresholds to distinguish the forgetting regimes (*reversible vs. irreversible* and *catastrophic vs. non-catastrophic*) is non-trivial, as they depend on factors like unlearning method and task complexity.

To assess generality, we apply GA+GD+WAGLE to Qwen3-8B-Base and Llama-3-8B, using an unrelated dataset for the relearning phase. Appendix Table 14 again shows *seemingly irreversible yet non-catastrophic forgetting*. We further find model-family differences in hyperparameter sensitivity when reaching comparable behavioral regimes. Such sensitivity likely shapes the boundary between reversible and irreversible forgetting and may guide future work on stable, irreversible, non-catastrophic unlearning.

## 6. Discussion and Conclusion

**Discussion.** (i) **Diagnostic metrics predict reversibility under a fixed protocol.** Under a bounded relearning budget and fixed data source, large layer-wise PCA shifts and high mean PCA distance predict recovery failure, while high CKA and concentrated Fisher spectra indicate reversibility;

*Table 5.* Yi-6B (GA+GD+WAGLE) performance under different relearning settings. Mean PCA distances on the forget and retain sets. Values in parentheses denote the change from the Original baseline (red: negative, blue: positive).

| Phase | F.Acc | R.Acc | PCA dist (forget) | PCA dist (retain) |
|---|---|---|---|---|
| Original model | 78.9 | 65.5 | 0.00 | 0.00 |
| LR=$2 \times 10^{-5}$, $N = 50$, relearned by retain set ($N = 25$) | | | | |
| Unlearn | 37.8 (-41.1) | 55.9 (-9.6) | 11.84 (+11.84) | 6.28 (+6.28) |
| Relearn | 46.9 (-32.0) | 58.3 (-7.2) | 9.00 (+9.00) | 5.91 (+5.91) |
| LR=$4 \times 10^{-5}$, $N = 50$, relearned by unrelated data ($N = 50$) | | | | |
| Unlearn | 27.8 (-51.1) | 51.4 (-14.1) | 26.02 (+26.02) | 8.37 (+8.37) |
| Relearn | 31.5 (-47.4) | 53.5 (-12.0) | 24.56 (+24.56) | 8.12 (+8.12) |

these signatures are consistent across models, datasets, and unlearning methods. (ii) **Practical guidance for controlling unlearning behavior.** Appendix Table 12 shows that mean PCA distance tracks recovery under a different budget: as drift decreases, forgetting accuracy recovers, whereas relearning on retain/unrelated data yields limited recovery. Together with our observations on *seemingly irreversible yet non-catastrophic forgetting* across models under specific settings, these results support using drift to tune learning rates and request counts. (iii) **Unlearning can enhance performance rather than merely erase information.** In several continual runs, relearning on the forget set exceeds the original accuracy, suggesting unlearning can act as an implicit contrastive regularizer that reorganizes representations toward more generalizable patterns (Section 5). (iv) **Toward a more reliable evaluation protocol.** Our findings suggest that durable unlearning cannot be reliably certified by any single metric. A practical protocol should jointly assess task-level behavior, constrained relearning robustness, and representation-level changes to better distinguish genuine erasure from reversible obscuration.

**Conclusion.** This work shows that task-level metrics can mislead LLM unlearning evaluation: apparent collapse may reflect reversible suppression rather than deletion when representations remain recoverable. We introduce a representation-level toolkit based on PCA similarity and shift, CKA, FIM, and mean PCA distance to characterize *reversibility* and *catastrophicity*. Our results suggest localized updates often yield superficial forgetting, while durable erasure requires substantial representational change. Yet irreversible, non-catastrophic forgetting remains challenging, as stronger updates can cause representational collapse and utility loss. Overall, our findings motivate protocols combining task metrics, constrained relearning, and representation-level diagnostics to distinguish erasure from obscuration.

## Acknowledgments

This work was supported by the Ministry of Science and Technology of the People's Republic of China (National Key Research and Development Programme, Grant No: 2025YFE0200100), the National Natural Science Foundation of China (Grant No: 62372130), the Research Grants Council (Grant No: 25207224), and the Innovation and Technology Fund (Grant No: ITS-140-23FP), Hong Kong SAR, China.

## Impact Statement

This work studies unlearning and relearning in large language models to improve the reliability of data removal for privacy and safety. Our representation-level diagnostics are used to evaluate and interpret model behavior (e.g., reversibility and drift), not to extract or infer sensitive information. We analyze recovery attempts (e.g., fine-tuning, prompting, in-context learning, and quantization) only to stress-test robustness and identify failure modes under realistic attacker capabilities, and we do not propose techniques to reconstruct private or copyrighted data. Overall, we expect these findings to enable more trustworthy unlearning evaluation protocols and more robust algorithm design.

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

# A. Appendix

## A.1. Limitations

Our experiments target multiple LLMs and a handful of tasks and unlearning methods; although our diagnostic framework is model-agnostic and designed to scale, empirical validation on much larger models and production-scale pipelines remains to be done. The constrained relearning protocol and selected metrics provide clear insights into representational drift but are not exhaustive and do not offer formal privacy guarantees. In this work, we primarily relied on four diagnostic tools—PCA similarity, PCA shift, CKA, and Fisher information—to capture different aspects of representational drift. Other feature-level methods, such as correlation-based approaches (e.g., SVCCA (Raghu et al., 2017)), offer similar perspectives on subspace similarity. Incorporating a broader suite of analytic tools is an important direction for future work.

## A.2. Related Work

**Machine Unlearning.** Machine unlearning has emerged as a critical direction for addressing privacy, safety, and bias in large language models (LLMs) (Yao et al., 2024; Jang et al., 2023; Li et al., 2024b; Liu et al., 2024; Gao et al., 2025; Shi et al., 2025; Xu et al., 2025; Zhang et al., 2025; Yuan et al., 2025; Wuerkaixi et al., 2025; Bourtoule et al., 2021). It is typically defined as either *exact* or *approximate* (Bourtoule et al., 2021). Exact unlearning requires the resulting model to be indistinguishable from one retrained from scratch on the retain set, fully eliminating any statistical trace of the forget set. Approximate unlearning relaxes this requirement to distributional or behavioral similarity, demanding only comparable outputs (e.g., forget-set accuracy) between unlearned and retrained models (Maini et al., 2024; Shi et al., 2025). For modern LLMs, however, exact unlearning is computationally infeasible, as full retraining or partition-based schemes scale poorly (Bourtoule et al., 2021). Consequently, approximate methods dominate practice in LLMs.

**Single-Shot Unlearning.** Most existing approaches are designed for single deletion events. Gradient-based strategies (e.g., GA) enforce forgetting directly but often incur significant utility loss (Yao et al., 2024). Recent advances such as WAGLE augment these methods with weight attribution (e.g., GA+GD+WAGLE), selectively updating the most influential parameters to enhance forgetting efficacy while mitigating utility degradation (Jia et al., 2024). Prompt-based steering avoids parameter updates, reducing cost, but typically achieves only superficial forgetting with vulnerability to reactivation (Liu et al., 2024). Model-editing methods, such as AlphaEdit (Li et al., 2025a), are lightweight and potentially robust, yet their behavior under sequential or heterogeneous requests remains underexplored.

**Continual Unlearning.** When unlearning requests arrive sequentially, naive extensions of single-shot methods tend to compound damage, leading to catastrophic forgetting and unstable dynamics (Barez et al., 2025; Shi et al., 2025). Each request operates on an already modified model, magnifying utility loss. Recent work has attempted to mitigate this through orthogonal updates (e.g., LoRA-based unlearning (Hu et al., 2022)) and OOD detectors. ALKN (Wuerkaixi et al., 2025) advances this line by providing a principled framework for continual unlearning, introducing parameter-level interventions and adaptive modules to counteract accumulative decline.

**Evaluations.** Evaluating unlearning efficacy remains an open challenge. Existing studies rely on three main classes of metrics. First, *classic (task-level) metrics* such as accuracy, perplexity (Yao et al., 2024; Li et al., 2024b) are widely used but can be misleading, since performance degradation does not guarantee removal of knowledge. Second, *memorization probes* (Lee et al., 2024) assess verbatim recall, offering finer granularity but failing to capture semantic or paraphrased knowledge. Third, robustness-based evaluations examine vulnerabilities to *jailbreaking* (Zou et al., 2023; Liu et al., 2023), *relearning attacks* (Lo et al., 2024), *prompt attack* (Patil et al., 2024) and even *quantization attacks* (Zhang et al., 2025). For *quantization attacks*, low-bit compression restores forget-set behavior without direct access to the forget.

For relearning attacks, RESTOR (Rezaei et al., 2025) evaluates whether an unlearning algorithm can both remove the influence of the forget set and restore the model to the counterfactual parameter state it would have reached without those datapoints. Benign relearning exposes a practical fragility of post-unlearning robustness: lightweight fine-tuning on seemingly innocuous data can "jog" an unlearned model into recovering suppressed behaviors (Hu et al., 2025), and such recovery may depend more on *syntactic similarity* than topical overlap (Yang et al., 2026). Related reversibility phenomena have also been observed beyond LLMs. In vision classifiers and diffusion models, apparent unlearning or concept erasure can be unstable, with forgotten classes or erased concepts re-emerging after adaptation (Siddiqui et al., 2025; George et al., 2025). Together, these studies show that apparent deletion can diverge from durable erasure across modalities.

However, existing studies mainly characterize whether forgotten behaviors can be recovered, while providing limited structural insight into what is altered inside the model. For LLMs, some representation-level methods, such as RMU (Li et al., 2024b), directly suppress target knowledge by steering forget-sample representations toward randomized targets. Such methods aim to improve unlearning algorithms, whereas our work takes a complementary diagnostic perspective. We use constrained relearning together with representation-level measurements to examine whether unlearning induces durable representational change or merely reversible behavioral suppression. This toolkit jointly characterizes *reversibility* and *catastrophicity* in both single-shot and continual unlearning, with the latter reflecting the realistic setting where deletion requests arrive sequentially over a model's lifecycle.

## A.3. Detailed Analysis Tools

**PCA Similarity and PCA Shift.**  For each Transformer layer, we perform PCA on the hidden activations of the *original* and *updated* models. Let $\mathbf{c}_{i,1}^{\text{orig}}$ and $\mathbf{c}_{i,1}^{\text{upd}}$ denote the first principal component (PC1) directions of layer $i$. The *PCA Similarity* is defined as

$$\text{PCA-Sim}(i) = \cos\big(\mathbf{c}_{i,1}^{\text{orig}}, \mathbf{c}_{i,1}^{\text{upd}}\big) = \frac{(\mathbf{c}_{i,1}^{\text{orig}})^\top \mathbf{c}_{i,1}^{\text{upd}}}{\|\mathbf{c}_{i,1}^{\text{orig}}\| \, \|\mathbf{c}_{i,1}^{\text{upd}}\|} \in [-1, 1],$$

where values near $1$ indicate stable directional alignment, values near $0$ suggest a near-orthogonal change in dominant directions, and values near $-1$ indicate a direction flip.

To capture translational drift, we compute the mean projection of activations along PC1 and the second principal component (PC2):

$$PC1^m(i) = \text{mean}\big(\mathbf{H}_i^m \mathbf{c}_{i,1}^{\text{orig}}\big), \qquad PC2^m(i) = \text{mean}\big(\mathbf{H}_i^m \mathbf{c}_{i,2}^{\text{orig}}\big), \qquad m \in \{\text{orig}, \text{upd}\},$$

$$PC1\Delta(i) = PC1^{\text{upd}}(i) - PC1^{\text{orig}}(i), \qquad p_{i,12}^{\text{orig}} = \big(0, \, PC2^{\text{orig}}(i)\big), \qquad p_{i,12}^{\text{upd}} = \big(PC1\Delta(i), \, PC2^{\text{upd}}(i)\big).$$

where $\mathbf{H}_i^{\text{orig}}$ and $\mathbf{H}_i^{\text{upd}}$ are the hidden activations extracted at layer $i$ from the original and updated models over the same set of inputs $\mathcal{X}$, and $\mathbf{c}_{i,2}^{\text{orig}}$ is the PC2 direction obtained by fitting PCA on the original activations at layer $i$.

We also introduce the *mean PCA distance* (PCA dist), defined as the average Euclidean distance between $p_{i,12}^{\text{upd}}$ and $p_{i,12}^{\text{orig}}$ across layers:

$$\text{PCA dist} = \frac{1}{L} \sum_{i=1}^{L} \left\| p_{i,12}^{\text{upd}} - p_{i,12}^{\text{orig}} \right\|_2 = \frac{1}{L} \sum_{i=1}^{L} \sqrt{\big(PC1\Delta(i)\big)^2 + \big(PC2^{\text{upd}}(i) - PC2^{\text{orig}}(i)\big)^2}.$$

where $L$ is the number of layers.

**Centered Kernel Alignment (CKA).**  To assess subspace alignment, we use linear Centered Kernel Alignment (CKA) (Kornblith et al., 2019), which compares activation matrices $X, Y \in \mathbb{R}^{N \times D}$ from before and after unlearning. First, we compute the centered Gram matrices:

$$\widetilde{K}_X = HXX^\top H, \qquad \widetilde{K}_Y = HYY^\top H, \qquad H = I_N - \tfrac{1}{N}\mathbf{1}\mathbf{1}^\top.$$

The CKA score is then given by:

$$\text{CKA}(X, Y) = \frac{\text{Tr}(\widetilde{K}_X \widetilde{K}_Y)}{\sqrt{\text{Tr}(\widetilde{K}_X^2)} \sqrt{\text{Tr}(\widetilde{K}_Y^2)}} \in [0, 1],$$

where values near $1$ indicate highly overlapping subspaces, and values near $0$ signal near-orthogonality.

**Fisher Information.**  To measure parameter-level importance, we compute the diagonal of the empirical Fisher Information Matrix (FIM). For each parameter $w_i$ and input distribution $\mathcal{D}_{\text{dis}}$, the diagonal entry is approximated as:

$$\text{FIM}_{ii} \approx \mathbb{E}_{(\mathbf{x}, y) \sim \mathcal{D}_{\text{dis}}} \left[ \big(\partial_{w_i} \log p(y \mid \mathbf{x}; \mathbf{w})\big)^2 \right].$$

Larger values indicate that $w_i$ has a stronger influence on the model's predictions. A substantial leftward shift in the Fisher spectrum after unlearning implies a flattened loss landscape and diminished parameter sensitivity.

*Table 6.* **Yi-6B simple-task metrics under four** $(LR, N)$ **settings.** For each block: forget/retain perplexity (F.Ppl / R.Ppl), forget/retain accuracy (F.Acc / R.Acc), CommonsenseQA (CSQA), GSM8K, and membership-inference AUC (MIA). The relearning phase uses the cumulative forget set. Values in parentheses denote the change from the Original baseline (red: negative, blue: positive) for accuracy-based metrics and MIA.

| Phase | Method | F.Ppl | R.Ppl | F.Acc | R.Acc | CSQA | GSM8K | MIA |
|---|---|---|---|---|---|---|---|---|
| | | | | LR=$3 \times 10^{-5}$, $N = 100$ | | | | |
| Original | — | 3.8 | 7.8 | 78.9 | 65.5 | 73.1 | 39.6 | 70.9 |
| | GA | ∞ | ∞ | 0.0 (-78.9) | 0.0 (-65.5) | 19.3 (-53.8) | 0.0 (-39.6) | 26.1 (-44.8) |
| | GA+GD | ∞ | ∞ | 9.7 (-69.2) | 2.3 (-63.2) | 19.7 (-53.4) | 0.0 (-39.6) | 16.8 (-54.1) |
| Unlearn | GA+KL | ∞ | ∞ | 9.0 (-69.9) | 6.2 (-59.3) | 19.6 (-53.5) | 0.0 (-39.6) | 17.8 (-53.1) |
| | NPO | 31296.5 | 597.9 | 37.8 (-41.1) | 37.9 (-27.6) | 62.2 (-10.9) | 1.0 (-38.6) | 60.1 (-10.8) |
| | NPO+KL | 348080.2 | 4482.0 | 64.3 (-14.6) | 55.9 (-9.6) | 64.9 (-8.2) | 1.4 (-38.2) | 59.0 (-11.9) |
| | RLabel | 63791.7 | 65903.4 | 0.0 (-78.9) | 0.0 (-65.5) | 20.9 (-52.2) | 0.0 (-39.6) | 65.1 (-5.8) |
| | GA | 137094.5 | 758443.5 | 2.1 (-76.8) | 1.8 (-63.7) | 19.7 (-53.4) | 0.0 (-39.6) | 74.5 (+3.6) |
| | GA+GD | 5274.5 | 9568.6 | 2.2 (-76.7) | 2.6 (-62.9) | 19.6 (-53.5) | 0.0 (-39.6) | 68.1 (-2.8) |
| Relearn | GA+KL | 5037.1 | 15019.9 | 1.7 (-77.2) | 1.6 (-63.9) | 20.6 (-52.5) | 0.0 (-39.6) | 70.7 (-0.2) |
| | NPO | 16.6 | 41.7 | 57.0 (-21.9) | 45.6 (-19.9) | 51.8 (-21.3) | 0.6 (-39.0) | 70.0 (-0.9) |
| | NPO+KL | 21.8 | 16.2 | 60.7 (-18.2) | 54.3 (-11.2) | 48.0 (-25.1) | 0.9 (-38.7) | 67.7 (-3.2) |
| | RLabel | 4056.1 | 15048.6 | 4.3 (-74.6) | 2.8 (-62.7) | 19.7 (-53.4) | 0.0 (-39.6) | 69.5 (-1.4) |
| | | | | LR=$5 \times 10^{-6}$, $N = 100$ | | | | |
| | GA | ∞ | ∞ | 9.1 (-69.8) | 6.2 (-59.3) | 19.6 (-53.5) | 0.0 (-39.6) | 23.2 (-47.7) |
| | GA+GD | ∞ | ∞ | 3.6 (-75.3) | 3.1 (-62.4) | 24.5 (-48.6) | 0.0 (-39.6) | 28.7 (-42.2) |
| Unlearn | GA+KL | ∞ | ∞ | 9.1 (-69.8) | 6.2 (-59.3) | 19.6 (-53.5) | 0.0 (-39.6) | 27.3 (-43.6) |
| | NPO | 3017.7 | 1110.6 | 50.1 (-28.8) | 52.3 (-13.2) | 72.9 (-0.2) | 37.5 (-2.1) | 50.6 (-20.3) |
| | NPO+KL | 38.5 | 232.4 | 77.6 (-1.3) | 64.3 (-1.2) | 73.1 | 37.6 (-2.0) | 65.4 (-5.5) |
| | RLabel | 57035.4 | 53377.1 | 0.1 (-78.8) | 0.4 (-65.1) | 19.1 (-54.0) | 0.0 (-39.6) | 63.6 (-7.3) |
| | GA | 3.7 | 7.8 | 80.0 (+1.1) | 64.9 (-0.6) | 70.2 (-2.9) | 39.9 (+0.3) | 68.0 (-2.9) |
| | GA+GD | 3.6 | 7.6 | 81.2 (+2.3) | 65.1 (-0.4) | 72.1 (-1.0) | 39.0 (-0.6) | 69.8 (-1.1) |
| Relearn | GA+KL | 3.6 | 8.4 | 81.1 (+2.2) | 64.8 (-0.7) | 71.6 (-1.5) | 40.7 (+1.1) | 68.3 (-2.6) |
| | NPO | 3.5 | 7.6 | 82.7 (+3.8) | 65.5 | 74.0 (+0.9) | 39.7 (+0.1) | 68.0 (-2.9) |
| | NPO+KL | 3.5 | 7.8 | 83.8 (+4.9) | 65.6 (+0.1) | 74.1 (+1.0) | 39.7 (+0.1) | 69.5 (-1.4) |
| | RLabel | 3.6 | 7.7 | 80.8 (+1.9) | 65.3 (-0.2) | 71.8 (-1.3) | 39.2 (-0.4) | 70.3 (-0.6) |
| | | | | LR=$3 \times 10^{-6}$, $N = 100$ | | | | |
| | GA | ∞ | ∞ | 16.8 (-62.1) | 14.4 (-51.1) | 69.5 (-3.6) | 12.3 (-27.3) | 25.2 (-45.7) |
| | GA+GD | 3.3 | 7.6 | 78.8 (-0.1) | 65.5 | 77.0 (+3.9) | 37.5 (-2.1) | 69.4 (-1.5) |
| Unlearn | GA+KL | ∞ | ∞ | 35.4 (-43.5) | 40.6 (-24.9) | 63.2 (-9.9) | 18.3 (-21.3) | 18.9 (-52.0) |
| | NPO | 3.7 | 7.9 | 78.3 (-0.6) | 65.0 (-0.5) | 73.3 (+0.2) | 38.7 (-0.9) | 68.4 (-2.5) |
| | NPO+KL | 3.8 | 8.1 | 78.4 (-0.5) | 65.1 (-0.4) | 73.6 (+0.5) | 38.6 (-1.0) | 66.7 (-4.2) |
| | RLabel | 36794.7 | 32562.0 | 3.8 (-75.1) | 3.2 (-62.3) | 19.3 (-53.8) | 2.2 (-37.4) | 61.4 (-9.5) |
| | GA | 3.7 | 7.6 | 80.8 (+1.9) | 65.2 (-0.3) | 73.4 (+0.3) | 39.9 (+0.3) | 68.6 (-2.3) |
| | GA+GD | 3.6 | 7.4 | 81.8 (+2.9) | 65.5 | 72.1 (-1.0) | 39.0 (-0.6) | 70.0 (-0.9) |
| Relearn | GA+KL | 3.6 | 10.3 | 81.0 (+2.1) | 63.3 (-2.2) | 67.2 (-5.9) | 40.7 (+1.1) | 70.7 (-0.2) |
| | NPO | 3.5 | 7.5 | 81.2 (+2.3) | 65.4 (-0.1) | 72.9 (-0.2) | 39.7 (+0.1) | 69.9 (-1.0) |
| | NPO+KL | 3.5 | 7.5 | 83.8 (+4.9) | 65.5 | 73.0 (-0.1) | 39.7 (+0.1) | 69.9 (-1.0) |
| | RLabel | 3.6 | 7.6 | 80.5 (+1.6) | 65.3 (-0.2) | 72.2 (-0.9) | 39.2 (-0.4) | 70.0 (-0.9) |
| | | | | LR=$3 \times 10^{-5}$, $N = 6$ | | | | |
| | GA | ∞ | ∞ | 36.3 (-42.6) | 36.1 (-29.4) | 69.1 (-4.0) | 5.8 (-33.8) | 29.6 (-41.3) |
| | GA+GD | 209.3 | 20.6 | 77.0 (-1.9) | 64.0 (-1.5) | 70.0 (-3.1) | 37.8 (-1.8) | 66.9 (-4.0) |
| Unlearn | GA+KL | ∞ | ∞ | 53.0 (-25.9) | 41.5 (-24.0) | 68.3 (-4.8) | 2.0 (-37.6) | 29.5 (-41.4) |
| | NPO | 12.3 | 10.7 | 71.6 (-7.3) | 59.4 (-6.1) | 71.7 (-1.4) | 24.7 (-14.9) | 68.7 (-2.2) |
| | NPO+KL | 8.9 | 10.7 | 74.7 (-4.2) | 62.1 (-3.4) | 72.8 (-0.3) | 32.2 (-7.4) | 67.9 (-3.0) |
| | RLabel | 51589.2 | 40622.9 | 0.4 (-78.5) | 0.7 (-64.8) | 19.8 (-53.3) | 0.0 (-39.6) | 62.6 (-8.3) |
| | GA | 6.8 | 11.4 | 70.5 (-8.4) | 58.7 (-6.8) | 64.5 (-8.6) | 18.4 (-21.2) | 68.2 (-2.7) |
| | GA+GD | 12.3 | 11.5 | 61.6 (-17.3) | 54.4 (-11.1) | 61.3 (-11.8) | 7.3 (-32.3) | 67.1 (-3.8) |
| Relearn | GA+KL | 17.1 | 11.6 | 66.6 (-12.3) | 56.2 (-9.3) | 60.6 (-12.5) | 3.0 (-36.6) | 65.0 (-5.9) |
| | NPO | 6.0 | 11.6 | 71.2 (-7.7) | 59.4 (-6.1) | 59.4 (-13.7) | 2.0 (-37.6) | 68.4 (-2.5) |
| | NPO+KL | 7.3 | 11.6 | 67.6 (-11.3) | 56.1 (-9.4) | 42.9 (-30.2) | 1.6 (-38.0) | 69.0 (-1.9) |
| | RLabel | 6.4 | 11.4 | 72.7 (-6.2) | 61.1 (-4.4) | 67.5 (-5.6) | 28.9 (-10.7) | 65.2 (-5.7) |

Together, these tools form a feature-space diagnostic suite: FIM captures global sensitivity, CKA measures subspace preservation, and PCA-based metrics expose fine-grained geometric drift across layers—enabling a robust assessment of representational degradation during unlearning.

*Table 7.* Qwen-2.5-7B: MIA / MATH / GSM8K Accuracy (%) for complex task under four settings. Bold numbers indicate improvements over the Original baseline in MATH or GSM8K. The relearning phase uses the cumulative forget set. Values in parentheses denote the change from the Original baseline (red: negative, blue: positive).

| Phase | Method | LR=$3 \times 10^{-5}$, N=6 | | | LR=$3 \times 10^{-6}$, N=6 | | | LR=$5 \times 10^{-6}$, N=6 | | | LR=$5 \times 10^{-6}$, N=100 | | |
|---|---|---|---|---|---|---|---|---|---|---|---|---|---|
| | | MIA | MATH | GSM8K | MIA | MATH | GSM8K | MIA | MATH | GSM8K | MIA | MATH | GSM8K |
| Original | —— | 99.3 | 9.0 | 80.1 | 99.3 | 9.0 | 80.1 | 99.3 | 9.0 | 80.1 | 99.3 | 9.0 | 80.1 |
| Unlearn | GA | 5.9(-93.4) | 0.0(-9.0) | 0.0(-80.1) | 0.9(-98.4) | 0.0(-9.0) | 0.0(-80.1) | 3.8(-95.5) | 0.0(-9.0) | 0.0(-80.1) | 5.5(-93.8) | 0.0(-9.0) | 0.0(-80.1) |
| | NPO | 95.9(-3.4) | 0.0(-9.0) | 0.2(-79.9) | 97.4(-1.9) | 21.5(+12.5) | 74.1(-6.0) | 67.4(-31.9) | 24.1(+15.1) | 71.8(-8.3) | 94.7(-4.6) | 0.0(-9.0) | 0.4(-79.7) |
| | RLabel | 35.5(-63.8) | 0.0(-9.0) | 0.0(-80.1) | 69.6(-29.7) | 0.0(-9.0) | 1.5(-78.6) | 11.2(-88.1) | 0.0(-9.0) | 0.0(-80.1) | 2.9(-96.4) | 0.0(-9.0) | 0.0(-80.1) |
| Relearn | GA | 97.6(-1.7) | 0.0(-9.0) | 1.1(-79.0) | 99.3(0.0) | 5.1(-3.9) | **83.2**(+3.1) | 99.4(+0.1) | **9.3**(+0.3) | 77.8(-2.3) | 99.2(-0.1) | 0.0(-9.0) | 0.0(-80.1) |
| | NPO | 95.8(-3.5) | 0.0(-9.0) | 0.0(-80.1) | 99.4(+0.1) | 4.7(-4.3) | **82.6**(+2.5) | 99.4(+0.1) | **16.5**(+7.5) | 75.7(-4.4) | 99.2(-0.1) | 0.0(-9.0) | 0.0(-80.1) |
| | RLabel | 99.5(+0.2) | 0.0(-9.0) | 0.0(-80.1) | 99.3(0.0) | 5.3(-3.7) | **83.3**(+3.2) | 99.3(0.0) | **10.0**(+1.0) | 77.2(-2.9) | 99.6(+0.3) | 0.0(-9.0) | 0.0(-80.1) |

*Table 8.* Single and continual unlearning results for GA across four models. Bold numbers indicate improvements over the Original baseline. Values in parentheses denote the change from each model's Original baseline (red: negative, blue: positive).

| Single unlearning: Qwen2.5-7B (GA) | | | Continual unlearning: Qwen3-8B-Base (GA) | | | Continual unlearning: Llama-3-8B (GA) | | | Continual unlearning: Qwen2.5-3B (GA) | | |
|---|---|---|---|---|---|---|---|---|---|---|---|
| | MATH | GSM8K | | F.Acc | R.Acc | | F.Acc | R.Acc | | F.Acc | R.Acc |
| Original model | 9.00 | 80.10 | Original model | 78.28 | 62.96 | Original model | 76.41 | 63.50 | Original model | 76.37 | 61.39 |
| $3 \times 10^{-6}$ (unlearn) | 6.24(-2.76) | 73.28(-6.82) | $6 \times 10^{-6}$ (unlearn) | 0.45(-77.83) | 0.21(-62.75) | $6 \times 10^{-6}$ (unlearn) | 0.38(-76.03) | 0.48(-63.02) | $6 \times 10^{-6}$ (unlearn) | 1.45(-74.92) | 2.56(-58.83) |
| $3 \times 10^{-6}$ (relearn) | 8.97(-0.03) | 78.29(-1.81) | $6 \times 10^{-6}$ (relearn) | **79.72**(+1.44) | 62.66(-0.30) | $6 \times 10^{-6}$ (relearn) | **76.49**(+0.08) | 63.21(-0.29) | $6 \times 10^{-6}$ (relearn) | **79.61**(+3.24) | 61.45(+0.06) |
| $6 \times 10^{-6}$ (unlearn) | 1.12(-7.88) | 30.21(-49.89) | $5 \times 10^{-5}$ (unlearn) | 0.02(-78.26) | 0.02(-62.94) | $5 \times 10^{-5}$ (unlearn) | 0.00(-76.41) | 0.00(-63.50) | $5 \times 10^{-5}$ (unlearn) | 0.01(-76.36) | 0.01(-61.38) |
| $6 \times 10^{-6}$ (relearn) | 8.62(-0.38) | 77.63(-2.47) | $5 \times 10^{-5}$ (relearn) | 0.03(-78.25) | 0.03(-62.93) | $5 \times 10^{-5}$ (relearn) | 0.02(-76.39) | 0.04(-63.46) | $5 \times 10^{-5}$ (relearn) | 3.58(-72.79) | 4.27(-57.12) |

*Table 9.* Continual unlearning results on the simple task for Qwen3-14B using GA and NPO under different learning rates.

| Qwen3-14B (GA) | | | | | Qwen3-14B (NPO) | | | | |
|---|---|---|---|---|---|---|---|---|---|
| | F.Acc | R.Acc | PCA dist (forget) | PCA dist (retain) | | F.Acc | R.Acc | PCA dist (forget) | PCA dist (retain) |
| $3 \times 10^{-6}$ (unlearn) | 75.2 | 60.3 | 25.88 | 25.28 | $3 \times 10^{-6}$ (unlearn) | 76.2 | 61.7 | 10.41 | 8.48 |
| $3 \times 10^{-6}$ (relearn) | 77.2 | 62.0 | 1.83 | 1.34 | $3 \times 10^{-6}$ (relearn) | 77.6 | 62.2 | 3.94 | 3.27 |
| $5 \times 10^{-6}$ (unlearn) | 48.4 | 40.5 | 40.36 | 48.23 | $5 \times 10^{-6}$ (unlearn) | 75.3 | 61.4 | 18.47 | 14.43 |
| $5 \times 10^{-6}$ (relearn) | 77.0 | 61.6 | 10.58 | 9.23 | $5 \times 10^{-6}$ (relearn) | 78.6 | 62.4 | 5.94 | 4.57 |
| $6 \times 10^{-5}$ (unlearn) | 2.3 | 0.2 | 77.47 | 78.51 | $6 \times 10^{-5}$ (unlearn) | 30.1 | 18.8 | 49.52 | 32.89 |
| $6 \times 10^{-5}$ (relearn) | 4.5 | 1.3 | 65.24 | 68.67 | $6 \times 10^{-5}$ (relearn) | 45.2 | 31.9 | 41.57 | 26.59 |

*Table 10.* Continual unlearning results on the simple task for Llama-3-8B-Instruct using GA and NPO under different learning rates.

| Llama-3-8B-Instruct (GA) | | | | | Llama-3-8B-Instruct (NPO) | | | | |
|---|---|---|---|---|---|---|---|---|---|
| | F.Acc | R.Acc | PCA dist (forget) | PCA dist (retain) | | F.Acc | R.Acc | PCA dist (forget) | PCA dist (retain) |
| $3 \times 10^{-6}$ (unlearn) | 0.1 | 0.2 | 1.79 | 1.60 | $3 \times 10^{-6}$ (unlearn) | 71.9 | 60.0 | 0.31 | 0.33 |
| $3 \times 10^{-6}$ (relearn) | 73.2 | 59.3 | 0.53 | 0.48 | $3 \times 10^{-6}$ (relearn) | 77.2 | 61.3 | 0.29 | 0.21 |
| $5 \times 10^{-6}$ (unlearn) | 0.0 | 0.0 | 2.56 | 2.25 | $5 \times 10^{-6}$ (unlearn) | 70.3 | 59.3 | 0.49 | 0.52 |
| $5 \times 10^{-6}$ (relearn) | 77.2 | 60.5 | 0.52 | 0.48 | $5 \times 10^{-6}$ (relearn) | 80.1 | 61.4 | 0.38 | 0.25 |
| $6 \times 10^{-5}$ (unlearn) | 0.0 | 0.0 | 33.85 | 33.75 | $6 \times 10^{-5}$ (unlearn) | 20.6 | 21.6 | 1.26 | 1.34 |
| $6 \times 10^{-5}$ (relearn) | 0.2 | 0.4 | 32.21 | 31.29 | $6 \times 10^{-5}$ (relearn) | 43.6 | 24.8 | 1.03 | 1.21 |

## A.4. Different types of relearning and Sample efficiency

### A.4.1. DIFFERENT TYPES OF RELEARNING

Beyond standard relearning attack, we further evaluated the unlearned Yi-6B model (GA-based and simple task setup) under four alternative recovery strategies: quantization attacks (Zhang et al., 2025), prompt attacks (Patil et al., 2024), jailbreaking (Liu et al., 2023), and in-context recovery. For quantization, we applied Int4 quantization directly to the unlearned model. For the other methods, which do not modify parameters, we adapted inputs to interface with our PCA analysis: *prompt attack* used paraphrased variants of the original inputs; *jailbreak attack* prepended the fixed prefix from (Liu et al., 2023); *in-context recovery* supplied five demonstrations from the forget set before evaluating the original inputs.

As shown in Table 11, none of these recovery strategies succeed in restoring the forgotten knowledge. Once the model enters the regime of *reversible catastrophic forgetting*, approaches that do not explicitly update parameters, or that introduce only minor perturbations such as quantization, cannot recover the lost representations. This indicates that the relevant information remains inaccessible under inference-time interventions and lightweight modifications. In contrast, explicit relearning that directly updates model parameters is required to reverse this forgetting state and regain performance on the forget set.

*Table 11.* Different recovery attempts on Yi-6B (GA, LR=$6 \times 10^{-6}$, $N = 100$). Mean PCA distance is computed on the forget set. Values in parentheses denote the change from the Original baseline (red: negative, blue: positive).

| Setting (Yi-6B, GA, LR=$6 \times 10^{-6}$, $N = 100$) | F.Acc | PCA dist (forget) |
|---|---|---|
| **Original model** | 78.90 | 0.00 |
| **Unlearned model** | 0.00 (-78.90) | 31.66 (+31.66) |
| **Quantization attack** | 0.00 (-78.90) | 32.21 (+32.21) |
| **In-context (num_demos = 5)** | 0.01 (-78.89) | 30.83 (+30.83) |
| **Prompt attack** | 0.03 (-78.87) | 29.14 (+29.14) |
| **Jailbreaking** | 0.03 (-78.87) | 30.04 (+30.04) |
| **Relearning attack** | 78.21 (-0.69) | 1.04 (+1.04) |

*Table 12.* Relearning comparison on Yi-6B (GA, LR=$6 \times 10^{-6}$, $N = 100$), evaluating the sample efficiency of different relearning data sources (forget, retain, unrelated). The results show how varying the amount and type of relearning data affects recovery performance and representational drift. Values in parentheses denote the change from each model's Original baseline (red: negative, blue: positive).

| Relearn % | F.Acc | PCA dist (forget) | F.Acc | PCA dist (forget) | F.Acc | PCA dist (forget) |
|---|---|---|---|---|---|---|
| | **Forget set** | | **Retain set** | | **Unrelated data** | |
| **Original model** | 78.90 | 0.00 | 78.90 | 0.00 | 78.90 | 0.00 |
| **Unlearned model** | 0.00 (-78.90) | 31.66 (+31.66) | 0.00 (-78.90) | 31.66 (+31.66) | 0.00 (-78.90) | 31.66 (+31.66) |
| **10%** | 67.28 (-11.62) | 8.49 (+8.49) | 0.05 (-78.85) | 30.57 (+30.57) | 0.02 (-78.88) | 31.02 (+31.02) |
| **30%** | 75.77 (-3.13) | 6.42 (+6.42) | 11.24 (-67.66) | 25.48 (+25.48) | 6.48 (-72.42) | 27.74 (+27.74) |
| **60%** | 77.13 (-1.77) | 4.31 (+4.31) | 45.24 (-33.66) | 14.69 (+14.69) | 38.83 (-40.07) | 17.51 (+17.51) |
| **100%** | 79.20 (+0.30) | 2.16 (+2.16) | 75.86 (-3.04) | 7.51 (+7.51) | 65.66 (-13.24) | 9.14 (+9.14) |

*Table 13.* Yi-6B (GA): Mean PCA distance under different learning rates. The left block uses China Taiwan for *relearning* only, while the right block uses TOFU for both *unlearning relearning*.

| | **Relearning with China Taiwan** | | | **Unlearning + Relearning with TOFU** | |
|---|---|---|---|---|---|
| **Learning Rate** | **Phase** | **PCA dist (forget)** | | **Phase** | **PCA dist (forget)** |
| $3 \times 10^{-6}$ | Unlearn | 17.12 | | Unlearn | 0.51 |
| $3 \times 10^{-6}$ | Relearn | 4.98 | | Relearn | 0.27 |
| $5 \times 10^{-6}$ | Unlearn | 20.27 | | Unlearn | 2.41 |
| $5 \times 10^{-6}$ | Relearn | 10.77 | | Relearn | 1.08 |
| $3 \times 10^{-5}$ | Unlearn | 193.13 | | Unlearn | 11.96 |
| $3 \times 10^{-5}$ | Relearn | 167.32 | | Relearn | 11.02 |

*Table 14.* Relearning on Qwen3-8B-Base and Llama-3-8B (GA+GD+WAGLE) with unrelated data. F.Acc/R.Acc is forget/retain accuracy, with mean PCA distances on the forget and retain sets. Values in parentheses denote the change from each model's Original baseline (red: negative, blue: positive).

| | **Qwen3-8B-Base (relearned by unrelated data)** | | | | **Llama-3-8B (relearned by unrelated data)** | | | |
|---|---|---|---|---|---|---|---|---|
| **Phase** | **F.Acc** | **R.Acc** | **PCA dist (forget)** | **PCA dist (retain)** | **F.Acc** | **R.Acc** | **PCA dist (forget)** | **PCA dist (retain)** |
| Original model | 78.28 | 62.96 | 0.00 | 0.00 | 76.41 | 63.50 | 0.00 | 0.00 |
| | **Relearning by unrelated data (Qwen: LR=$5 \times 10^{-6}$, N=50; Llama: LR=$3 \times 10^{-6}$, N=50)** | | | | | | | |
| Unlearn | 48.52 (-29.76) | 56.47 (-6.49) | 8.49 (+8.49) | 5.98 (+5.98) | 42.59 (-33.82) | 53.47 (-10.03) | 14.29 (+14.29) | 7.12 (+7.12) |
| Relearn | 53.21 (-25.07) | 59.16 (-3.80) | 6.57 (+6.57) | 4.32 (+4.32) | 49.78 (-26.63) | 56.24 (-7.26) | 11.47 (+11.47) | 6.21 (+6.21) |

### A.4.2. SAMPLE EFFICIENCY

To examine sample efficiency, we extend our GA-based relearning experiments ($LR = 6 \times 10^{-6}$, $N = 100$ on simple task) across three data sources, namely the forget set, the retain set, and unrelated data (see Section 2 for details). We evaluate each source using 10%, 30%, 60%, and 100% of the original forget-set size, keeping the unlearning setting fixed. This design isolates how both data type and data volume affect recovery behavior and the degree of representational drift.

As shown in Table 12, these experiments reveal a clear hierarchy in recovery efficiency. Relearning on the forget set provides the strongest and fastest recovery, with PCA distances approaching those of the original model even at moderate sample sizes. In contrast, relearning using the retain set or unrelated data restores performance only gradually; both sources are substantially less sample-efficient and yield slower improvements in representational alignment.

*Table 15.* Comparison of RMU, UnDIAL, AltPO, and PDU under different learning rates for unlearning and relearning on the simple task in continual unlearning.

| Learning Rate | Phase | RMU | | | | UnDIAL | | | | AltPO | | | | PDU | | | |
|---|---|---|---|---|---|---|---|---|---|---|---|---|---|---|---|---|---|
| | | F.Acc | R.Acc | PCA dist (forget) | PCA dist (retain) | F.Acc | R.Acc | PCA dist (forget) | PCA dist (retain) | F.Acc | R.Acc | PCA dist (forget) | PCA dist (retain) | F.Acc | R.Acc | PCA dist (forget) | PCA dist (retain) |
| $3 \times 10^{-6}$ | Unlearn | 48.9 | 58.6 | 1.58 | 0.62 | 46.5 | 59.4 | 1.40 | 0.47 | 41.7 | 56.2 | 2.30 | 0.81 | 36.2 | 53.1 | 3.50 | 1.26 |
| $3 \times 10^{-6}$ | Relearn | 79.8 | 64.6 | 0.33 | 0.18 | 79.5 | 65.1 | 0.30 | 0.12 | 77.6 | 63.5 | 0.60 | 0.24 | 74.8 | 61.7 | 0.90 | 0.38 |
| $5 \times 10^{-6}$ | Unlearn | 22.1 | 38.5 | 10.68 | 3.84 | 27.4 | 49.2 | 8.40 | 2.76 | 19.8 | 43.1 | 12.50 | 4.20 | 14.7 | 35.8 | 15.90 | 6.43 |
| $5 \times 10^{-6}$ | Relearn | 78.5 | 63.8 | 0.90 | 0.73 | 79.1 | 64.3 | 0.80 | 0.41 | 75.9 | 62.1 | 1.72 | 0.69 | 72.1 | 60.4 | 2.50 | 1.14 |
| $3 \times 10^{-5}$ | Unlearn | 2.3 | 5.1 | 40.26 | 24.60 | 7.9 | 18.4 | 24.70 | 10.95 | 4.8 | 11.6 | 31.80 | 18.27 | 1.6 | 4.8 | 45.70 | 27.84 |
| $3 \times 10^{-5}$ | Relearn | 45.5 | 56.8 | 33.35 | 17.90 | 61.3 | 60.2 | 8.80 | 4.22 | 52.4 | 58.7 | 14.90 | 8.42 | 41.2 | 54.9 | 36.30 | 20.63 |

## A.5. Mean PCA distance under Different Dataset

To examine the role of distributional alignment, we evaluate unlearning and relearning under two dataset settings. First, we use the *TOFU* benchmark (Maini et al., 2024), where unlearning and relearning occur within the same distribution. Second, we treat different languages as out-of-distribution (OOD) and relearn on a Traditional-Chinese corpus after unlearning on the simple task. This setup allows us to test whether cross-lingual signals can support effective recovery, and how their efficacy compares to in-distribution relearning.

Table 13 confirms that **cross-lingual relearning improves the model but achieves less complete restoration than English data**: mean PCA distance and related summary metrics move closer to baseline values, yet remain higher. Greater linguistic or domain dissimilarity therefore reduces recovery efficacy, though partial restoration is still attainable.

For the TOFU dataset, the overall pattern holds: **learning rate and the number of unlearning requests ($N$) effectively regulate feature drift and reversibility**. However, the representational shifts induced by TOFU are milder than those observed in our simple and complex tasks. We attribute this to the smaller and less diverse nature of TOFU's corpus; many entries are short and contain only author metadata, making its impact on the model's feature space comparatively limited.

## A.6. Additional Results on Broader Unlearning Paradigms

To further examine whether our taxonomy generalizes beyond the six canonical baselines, we evaluate four additional unlearning methods on the simple task under the same continual-unlearning protocol: RMU (Li et al., 2024b), UnDIAL (Dong et al., 2025), AltPO (Mekala et al., 2025), and PDU (Entesari et al., 2025). These methods span fundamentally different unlearning paradigms. RMU represents a representation-misdirection approach, steering forget-sample representations at an intermediate layer toward a target random representation. UnDIAL adopts self-distillation with adjusted logits, selectively reducing the influence of targeted tokens while preserving general language capability. AltPO frames unlearning as preference optimization, combining negative feedback with in-domain positive feedback on the forget set to avoid incoherent or nonsensical responses. PDU formulates unlearning as constrained primal-dual optimization, enforcing forgetting through a logit-margin flattening loss while preserving retained utility via explicit constraints.

As shown in Table 15, our core findings remain consistent across these diverse approaches. At lower learning rates, $3 \times 10^{-6}$ and $5 \times 10^{-6}$, all four methods exhibit strong post-relearning recovery, accompanied by relatively small mean PCA distances. This indicates a reversible regime, where unlearning mainly suppresses task-level behavior while preserving recoverable representation geometry. At the aggressive learning rate $3 \times 10^{-5}$, post-relearning recovery drops substantially and PCA distances remain large, suggesting a shift toward irreversible forgetting. Notably, UnDIAL and AltPO show greater structural robustness under aggressive hyperparameters, avoiding the near-total representational collapse observed for GA under aggressive settings. Overall, these results suggest that the proposed reversibility–catastrophicity taxonomy captures a broader phenomenon across representation engineering, distillation, preference optimization, and constrained optimization methods.

## A.7. Detailed Analysis Results

### A.7.1. PRINCIPAL COMPONENT ANALYSIS: SIMILARITY AND SHIFT

Across the same hyper-parameter grid, Figure 7 and Figure 11 (PCA Similarity and Shift) provide complementary views of representational drift. For GA, higher learning rates drive unlearned states far from the original, while relearning fails to return, producing long rays of *irreversible* drift. GA+GD narrows the spread but still collapses at $3 \times 10^{-5}$.

On Qwen-2.5-7B, GA shifts span thousands of PC1 units and drive PC2 to extreme negatives (Figure 13c,f,i), consistent

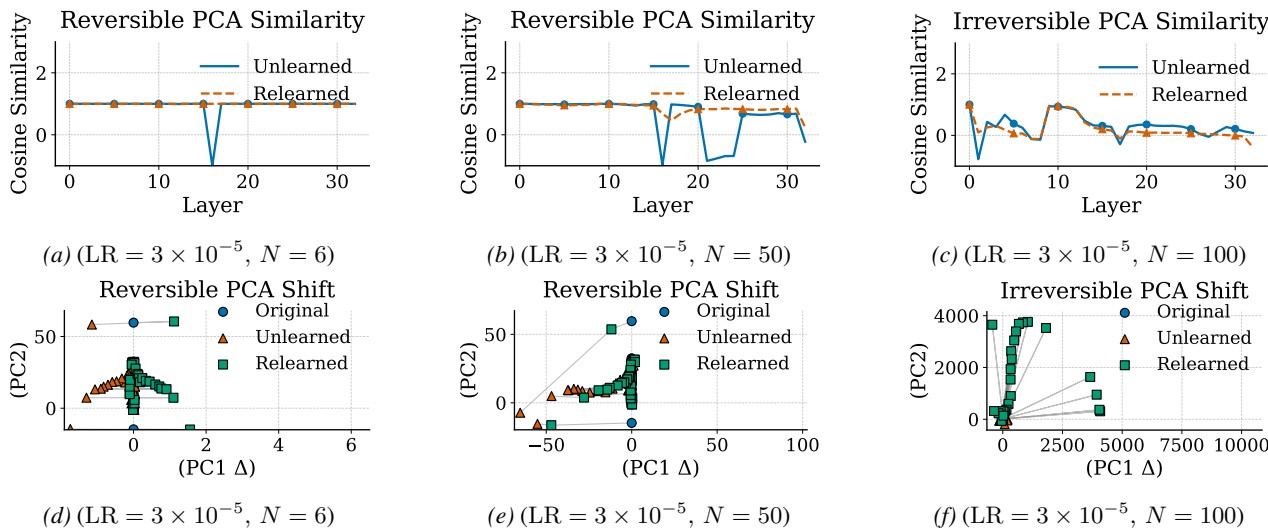

*Figure 5.* Layer-wise PCA Similarity and Shift for GA on Yi-6B (simple task). vary $N \in \{6, 50, 100\}$ at $\text{LR} = 3 \times 10^{-5}$. Sustained low similarity or large shifts signal severe, irreversible catastrophic forgetting, whereas partial similarity or small shifts indicate mild, reversible catastrophic forgetting. Input queries are drawn from the forget set.

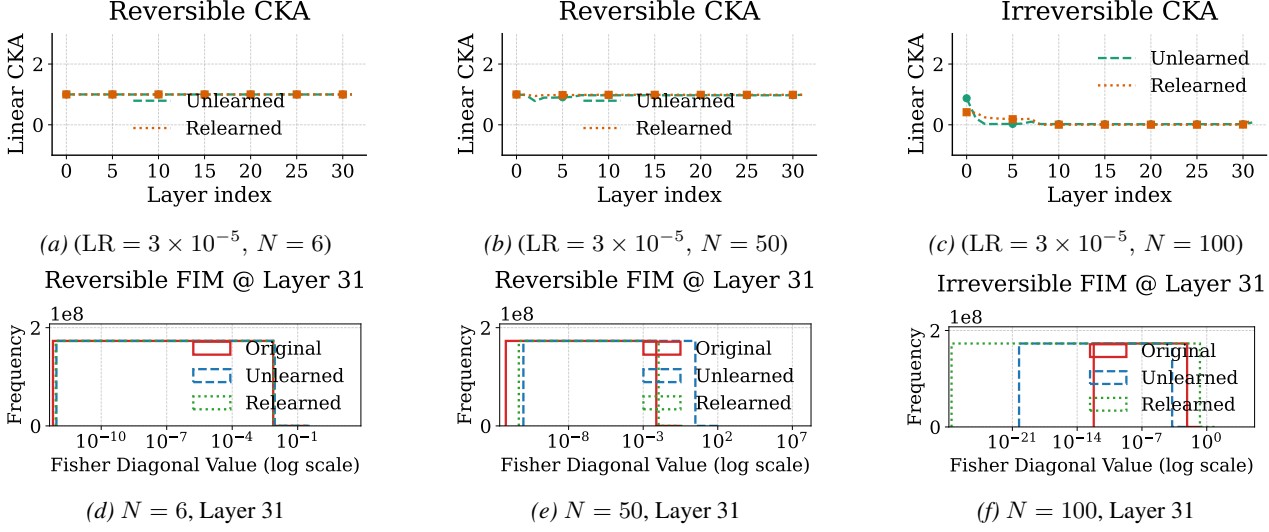

*Figure 6.* CKA and FIM for GA on Yi-6B, simple task. Vary $\text{LR} = 3 \times 10^{-5}$ with $N \in \{6, 50, 100\}$. High CKA ( 1) and concentrated FIM spectra indicate reversible catastrophic forgetting, while persistently low CKA and large-shifted, flattened spectra denote severe representational drift and irreversible catastrophic forgetting. Input queries are drawn from the forget set.

with the multi-layer perturbations predicted in Section 4. In complex tasks such as mathematical reasoning, even small perturbations in hidden states can lead to substantial performance differences. This is reflected in our PCA–Similarity analysis, where seemingly minor changes in hidden state geometry correspond to meaningful behavioral variations. Besides, PCA-Similarity captures global alignment, whereas PCA–Shift highlights fine-grained translational drift. This distinction also explains why Figure 9h,i show only moderate misalignment under similarity but reveal pronounced displacements under shift (cf. Figure 13). Using both metrics thus provides a more complete characterization of reversibility. Overall, these results confirm that GA, with or without GD or KL, induces large and often irreversible displacements, whereas NPO variants, and to a lesser extent RLabel, constrain less shifts, consistent with our utility findings.

A.7.2. CENTERED KERNEL ALIGNMENT ANALYSIS

Figures 15–17 report layer-wise linear CKA between the original model and its unlearned or relearned counterparts. Across both Yi-6B and Qwen-2.5-7B, GA stands out: as the learning rate or $N$ increases, its CKA curve drops close to zero in most layers and fails to recover, revealing a deep subspace fracture consistent with the irreversible PCA trends. GA+GD and GA+KL mitigate this decline to some extent but do not restore full alignment after relearning.

Task complexity does not alter the ordering but amplifies the differences. On the math-heavy, difficult benchmark, GA's tail layers fall almost to zero at high learning rates, whereas NPO maintains significantly higher alignment. Taken together with the PCA-Shift results, these findings show that GA-style objectives consistently break subspace alignment, NPO variants preserve much greater stability, and RLabel induces moderate but partly recoverable distortions.

A.7.3. FISHER INFORMATION ANALYSIS

Figures 19–33 plot the empirical Fisher spectra layer by layer. Across both Yi-6B (simple) and Qwen-2.5-7B (complex), GA and its variants exhibit a pronounced leftward shift of the diagonal histogram as LR or $N$ increase. The peaks move several orders of magnitude in middle and deep layers, reflecting a flattened loss surface and diminished parameter salience. Crucially, these shifts persist after relearning, marking the onset of irreversible forgetting.

NPO, NPO+KL, and RL produce smaller leftward displacements under moderate LR or $N$, and their Fisher spectra recenter after relearning, indicating primarily reversible drift. Under extreme settings (e.g., LR $= 3 \times 10^{-5}$ or $N = 100$), these methods also show persistent displacement in some layers, suggesting milder but still irreversible forgetting.

Figures 14, 10, 18, and 34 examine relearning dynamics when the fine-tuning data and input query are drawn from the forget set, the retain set, or an unrelated data: i) across all sources, the overall trends are similar: alignment can be partially restored, but recovery is consistently weaker with unrelated data, underscoring that effective relearning depends on both the size and the relevance of the training set; ii) the observed behavior also varies with the choice of input queries. In the case of *reversible catastrophic forgetting*, all forget set, retain set, and unrelated data undergo the similar feature drifts.

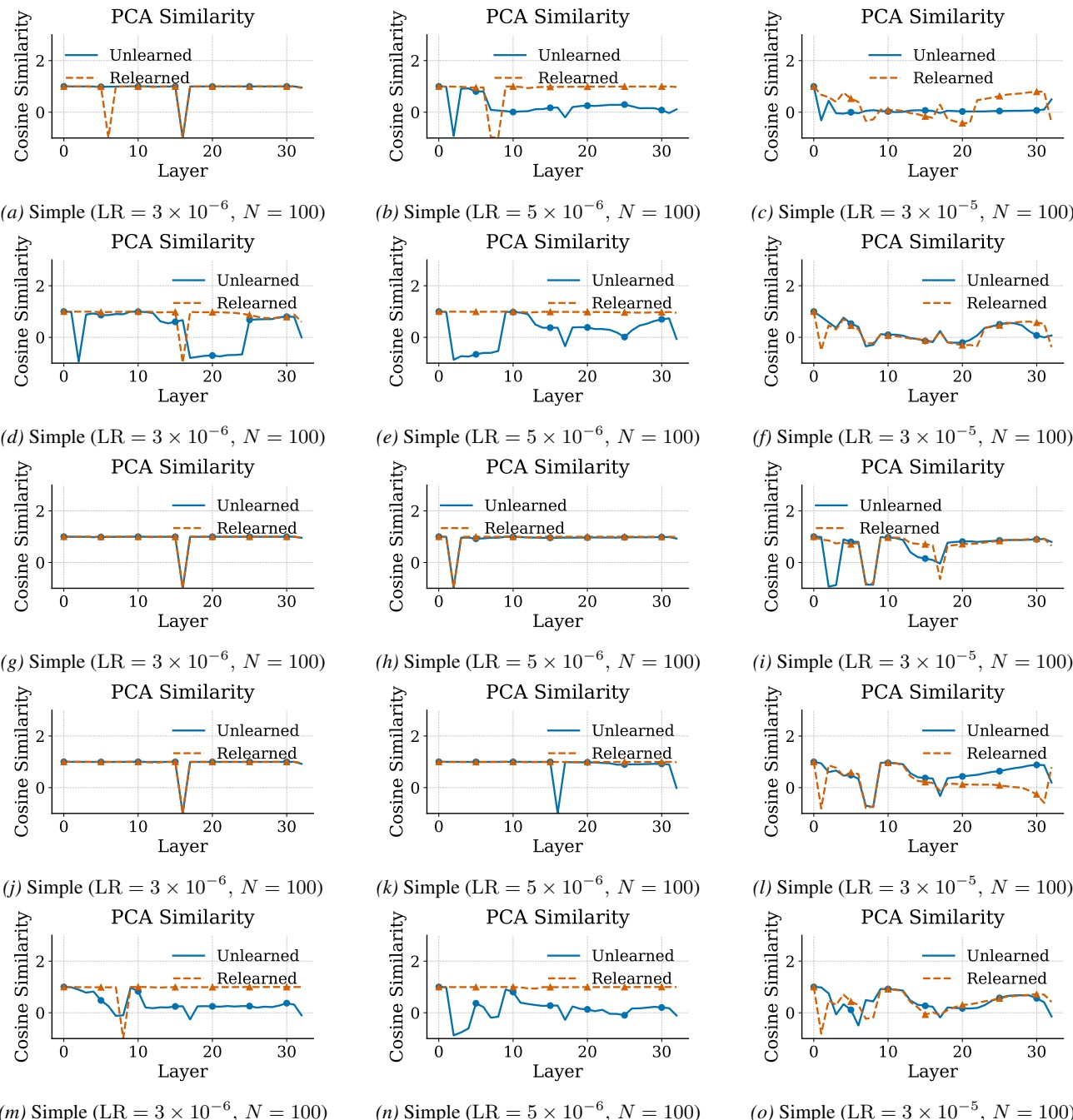

*Figure 7.* PCA Similarity Across Layers. Each row shows results under different unlearning methods: GA+GD (a–c), GA+KL (d–f), NPO (g–i), NPO+KL (j–l), and Rlable (m–o). All plots are for the simple task on Yi-6B, using three learning rates $\{3 \times 10^{-6}, 5 \times 10^{-6}, 3 \times 10^{-5}\}$ and fixed $N = 100$.

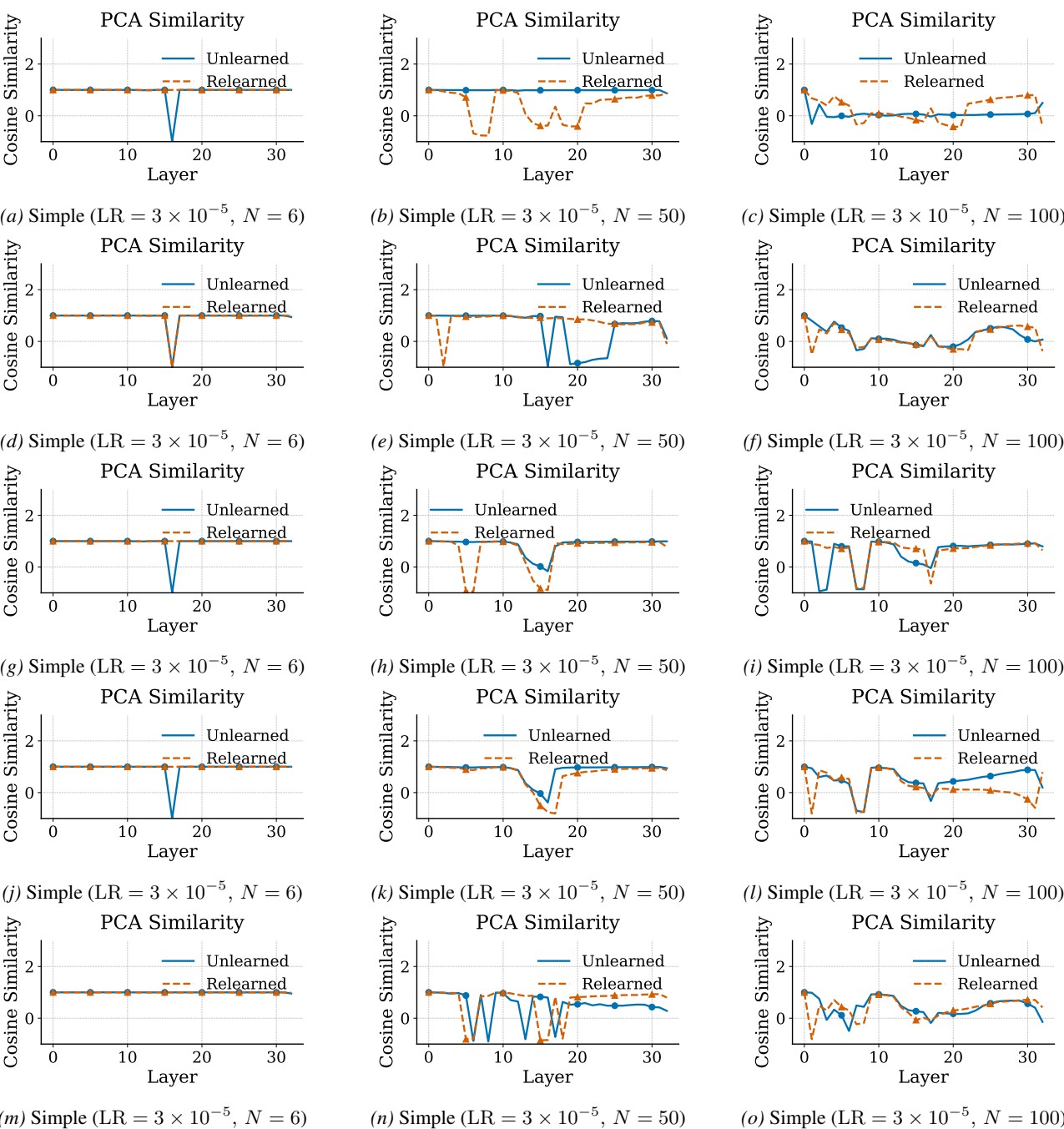

*Figure 8.* PCA Similarity Across Layers. Each row shows results under different unlearning methods: GA+GD (a–c), GA+KL (d–f), NPO (g–i), NPO+KL (j–l), and Rlable (m–o). Simple task on Yi-6B with fixed learning rate LR $= 3 \times 10^{-5}$ and varying unlearning requests $N \in \{6, 50, 100\}$.

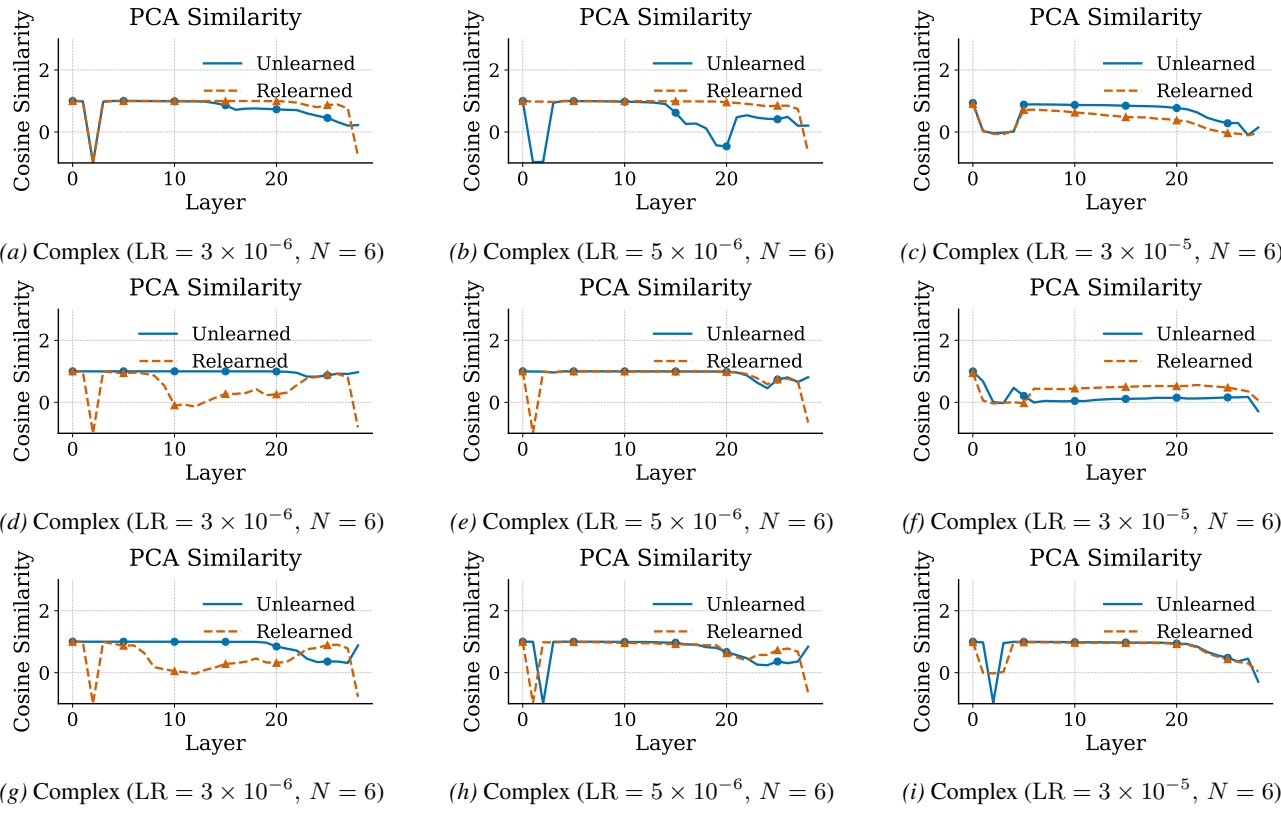

*Figure 9.* PCA Similarity Across Layers. Each row shows results under different unlearning methods: GA (a-c) NPO (d–f), Rlable (g–j). All plots are for the complex task on Qwen2.5-7B, using three learning rates $\{3 \times 10^{-6}, 5 \times 10^{-6}, 3 \times 10^{-5}\}$ and fixed $N = 6$.

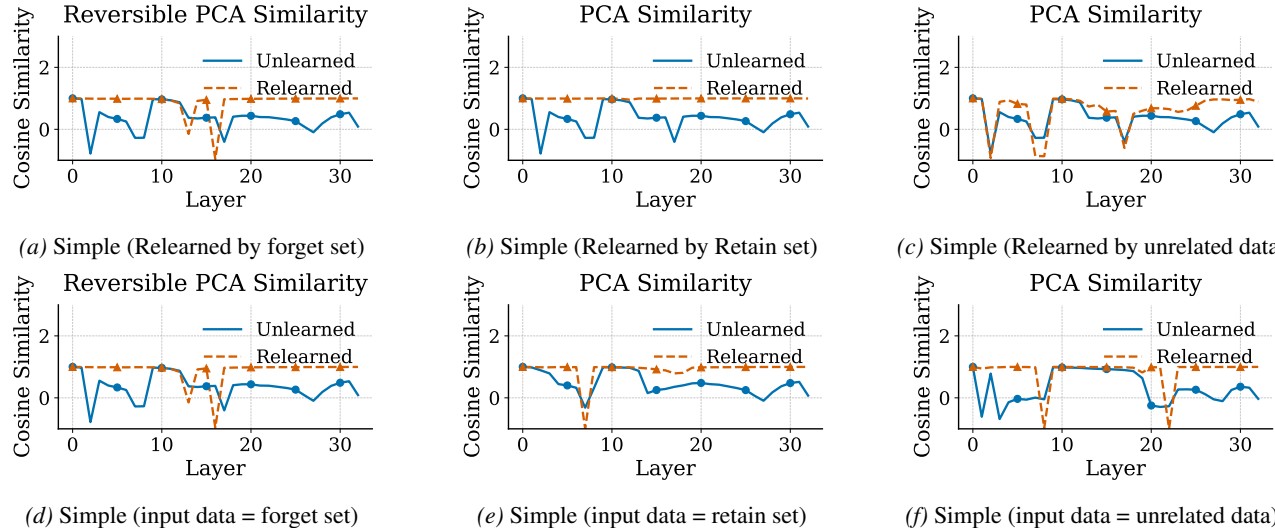

*Figure 10.* PCA Similarity Analysis for GA under Varied Relearning and Evaluation Inputs on Yi-6B (Simple Task). (a–c): Relearning is performed using the forget set, retain set, or unrelated data respectively. (d–f): PCA similarity is measured using the forget set, retain set, or unrelated data as evaluation input.

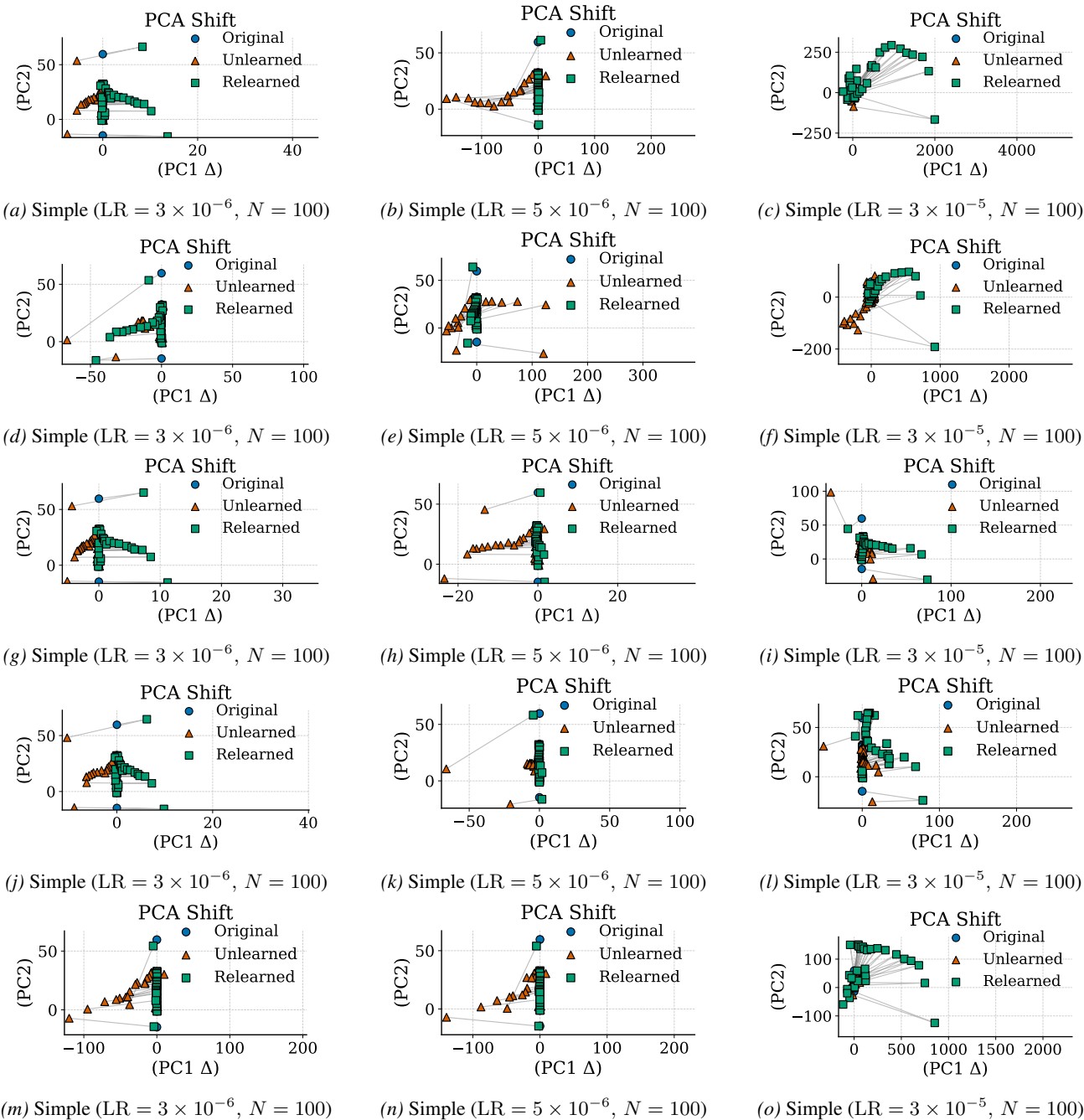

*Figure 11.* PCA Shift Across Layers. Each row shows results under different unlearning methods: GA+GD (a–c), GA+KL (d–f), NPO (g–i), NPO+KL (j–l), and Rlable (m–o). All plots are for the simple task on Yi-6B, using three learning rates $\{3 \times 10^{-6}, 5 \times 10^{-6}, 3 \times 10^{-5}\}$ and fixed $N = 100$.

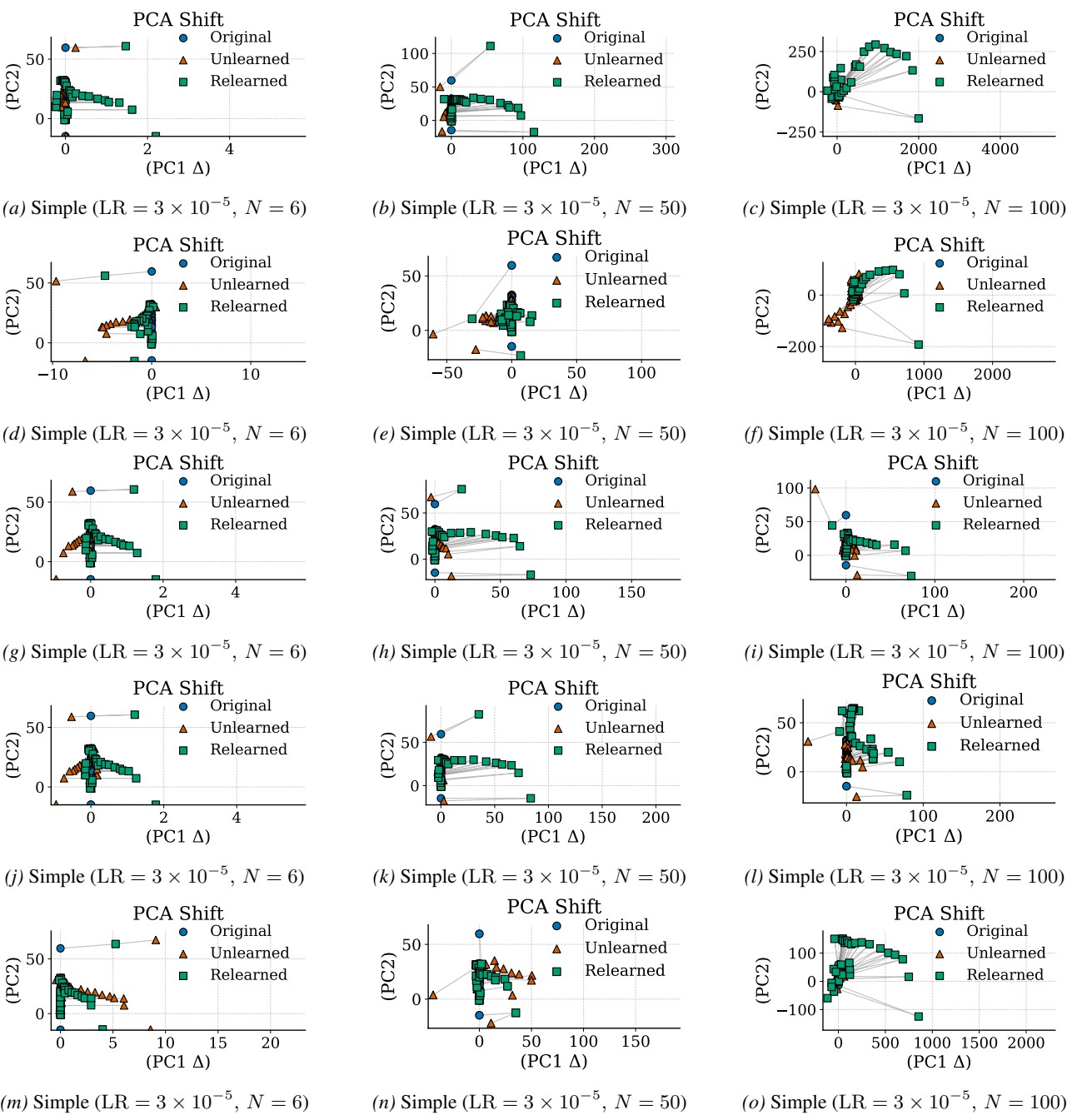

*Figure 12.* PCA Shift Across Layers. Each row shows results under different unlearning methods: GA+GD (a–c), GA+KL (d–f), NPO (g–i), NPO+KL (j–l), and Rlable (m–o). Simple task on Yi-6B with fixed learning rate LR $= 3 \times 10^{-5}$ and varying unlearning requests $N \in \{6, 50, 100\}$.

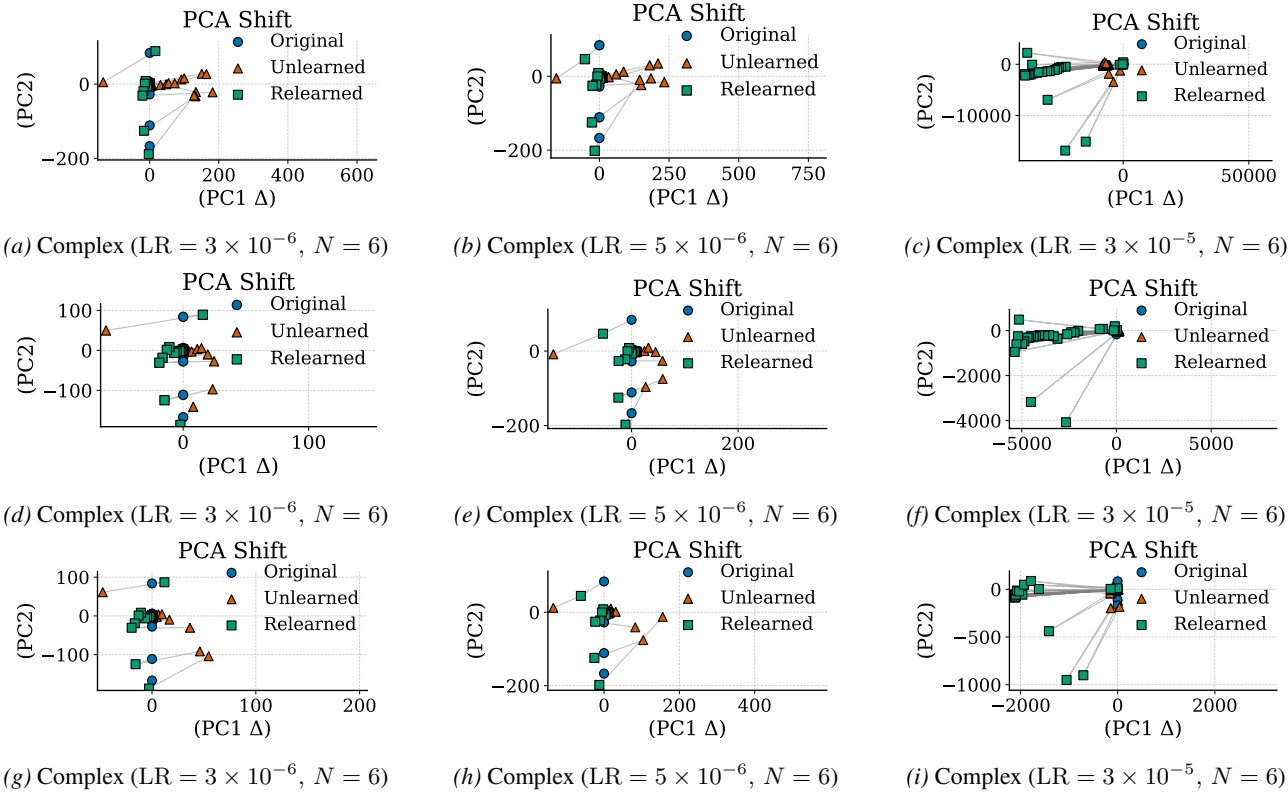

*Figure 13.* PCA Shift Across Layers. Each row shows results under different unlearning methods: GA (a-c) NPO (d–f), Rlable (g–j). All plots are for the complex task on Qwen2.5-7B, using three learning rates $\{3 \times 10^{-6}, 5 \times 10^{-6}, 3 \times 10^{-5}\}$ and fixed $N = 6$.

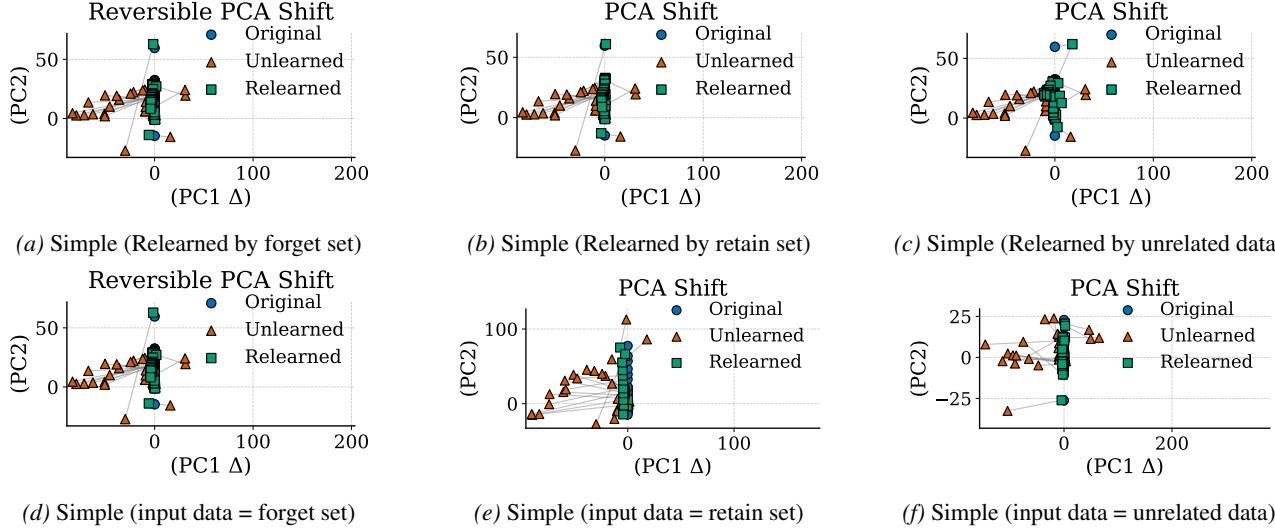

*Figure 14.* PCA Shift Analysis under Varied Relearning and Evaluation Inputs on Yi-6B (Simple Task). (a–c): Relearning is performed using the forget set, retain set, or unrelated data respectively. (d–f): PCA shift is measured using the forget set, retain set, or unrelated data as evaluation input.

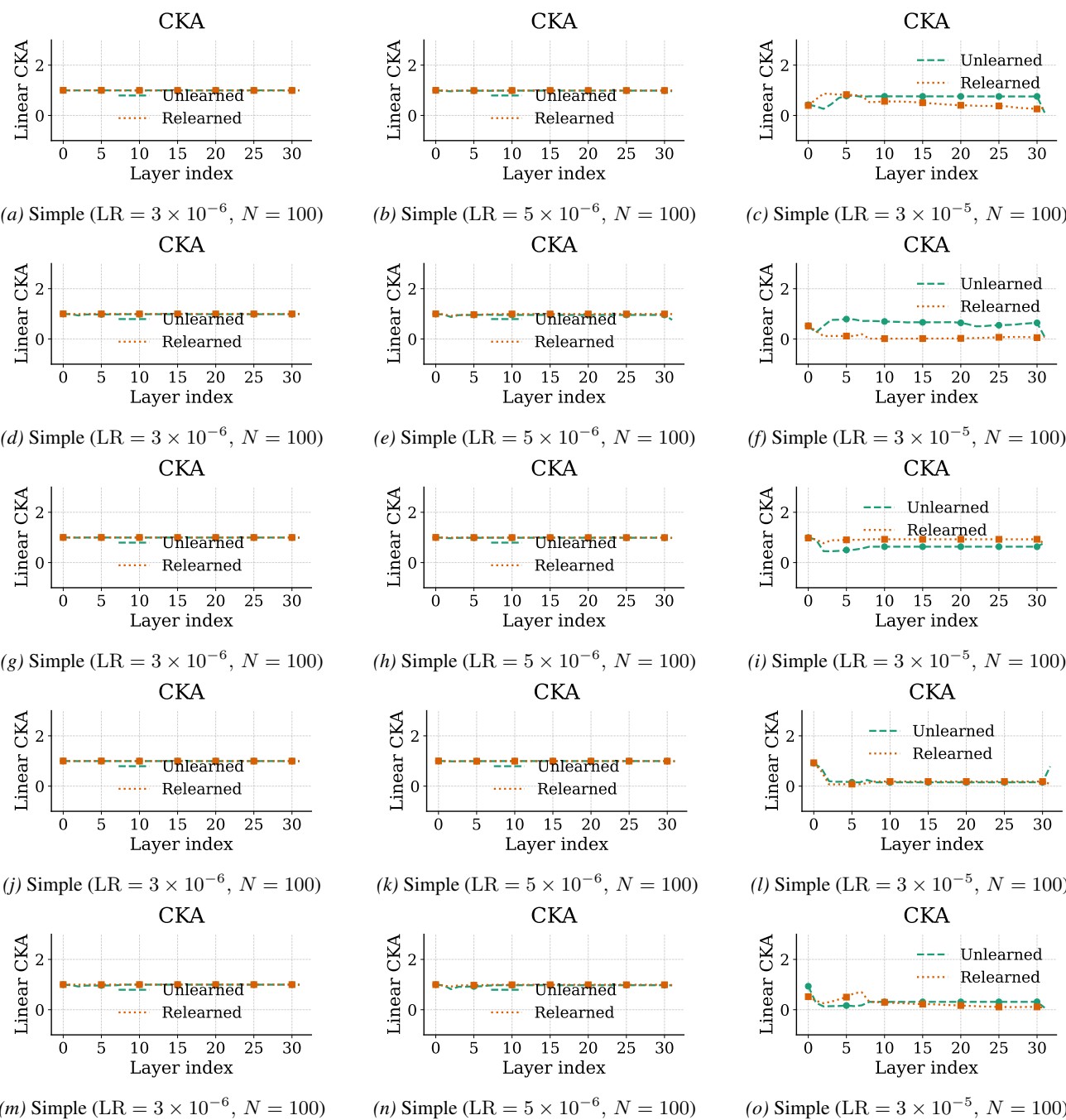

*Figure 15.* CKA Across Layers. Each row shows results under different unlearning methods: GA+GD (a–c), GA+KL (d–f), NPO (g–i), NPO+KL (j–l), and Rlable (m–o). All plots are for the simple task on Yi-6B, using three learning rates $\{3 \times 10^{-6}, 5 \times 10^{-6}, 3 \times 10^{-5}\}$ and fixed $N = 100$.

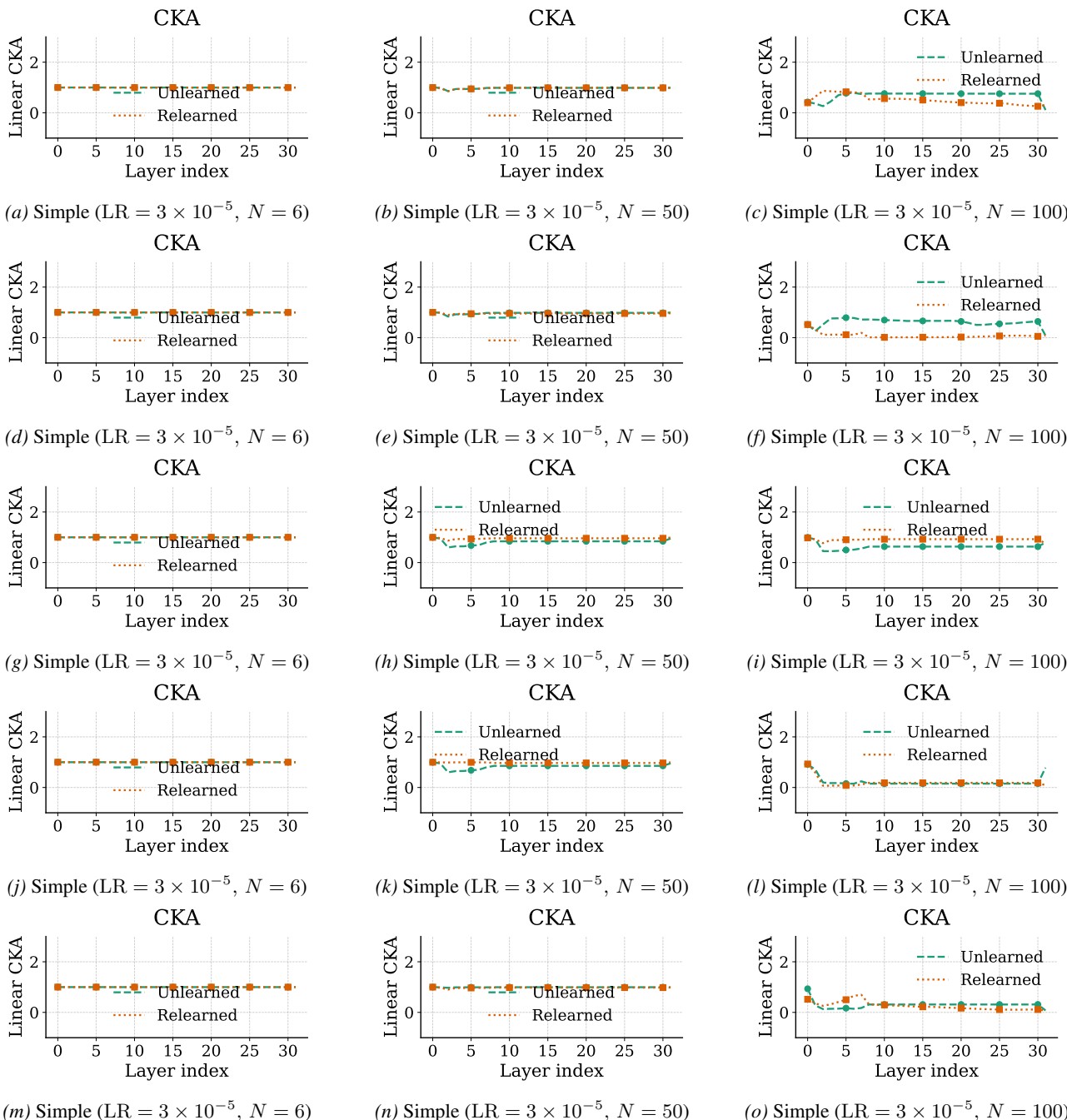

*Figure 16.* CKA Across Layers. Each row shows results under different unlearning methods: GA+GD (a–c), GA+KL (d–f), NPO (g–i), NPO+KL (j–l), and Rlable (m–o). Simple task on Yi-6B with fixed learning rate LR $= 3 \times 10^{-5}$ and varying unlearning requests $N \in \{6, 50, 100\}$.

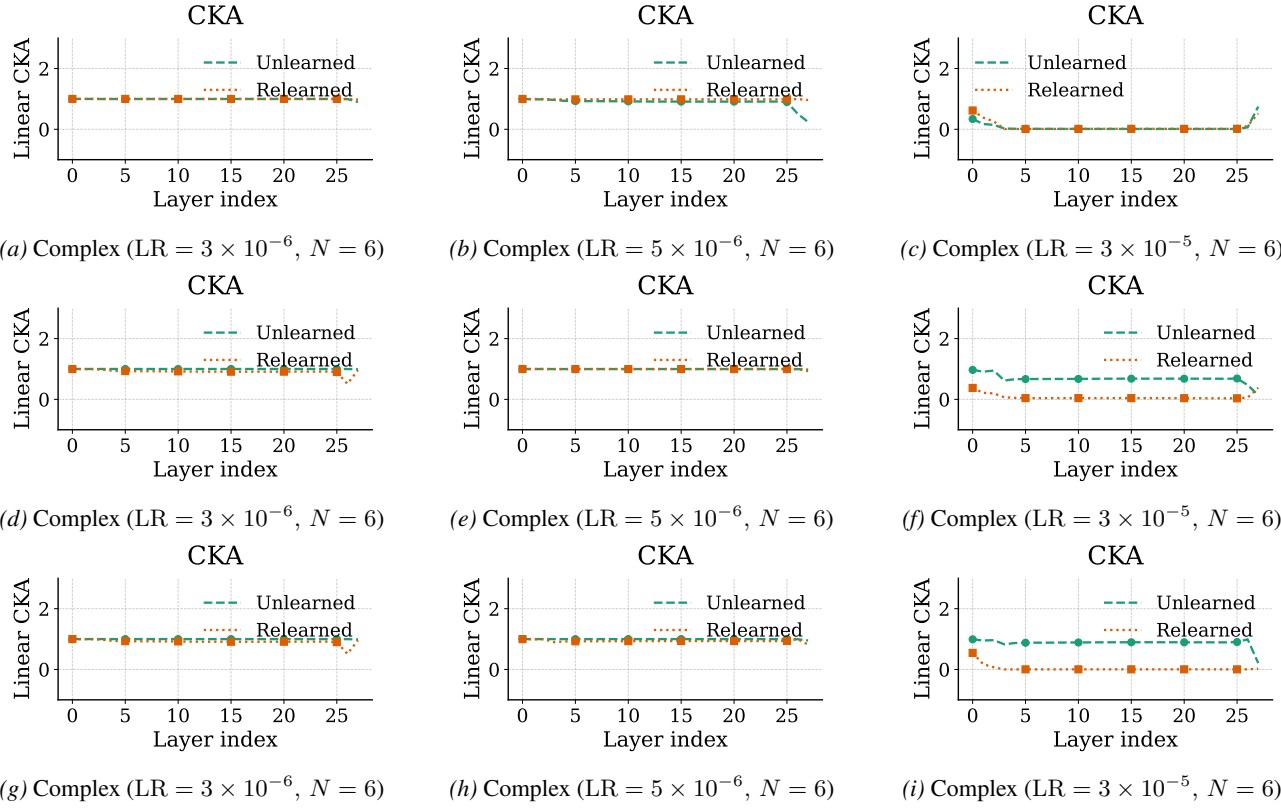

*Figure 17.* CKA Across Layers. Each row shows results under different unlearning methods: GA (a-c) NPO (d–f), Rlable (g–j). All plots are for the complex task on Qwen2.5-7B, using three learning rates $\{3 \times 10^{-6}, 5 \times 10^{-6}, 3 \times 10^{-5}\}$ and fixed $N = 6$.

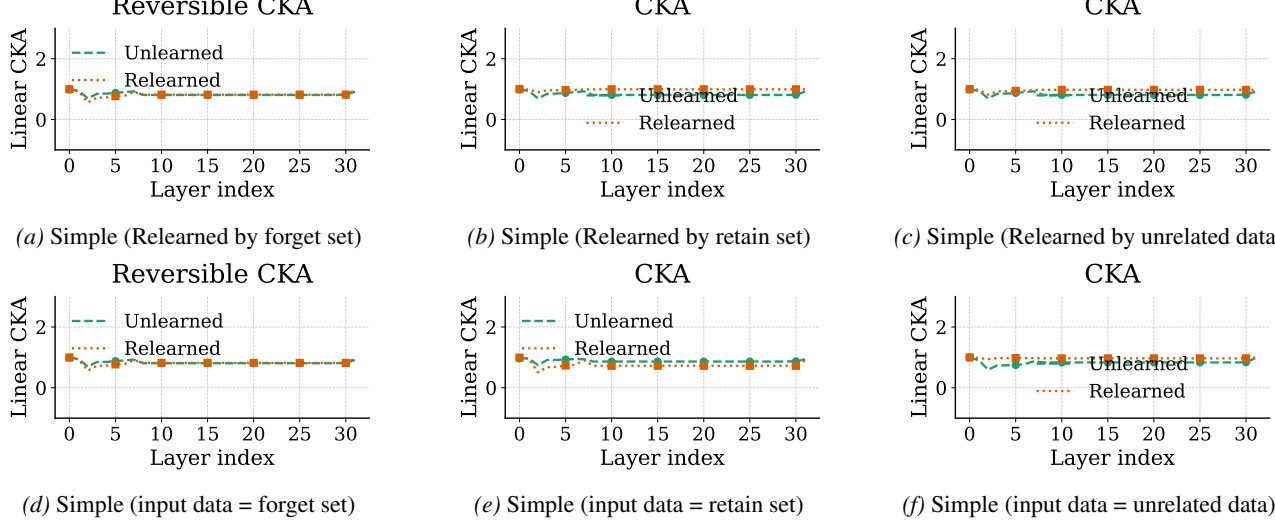

*Figure 18.* CKA Analysis under Varied Relearning and Evaluation Inputs on Yi-6B (Simple Task). (a–c): Relearning is performed using the forget set, retain set, or unrelated data respectively. (d–f): CKA is measured using the forget set, retain set, or unrelated data as evaluation input.

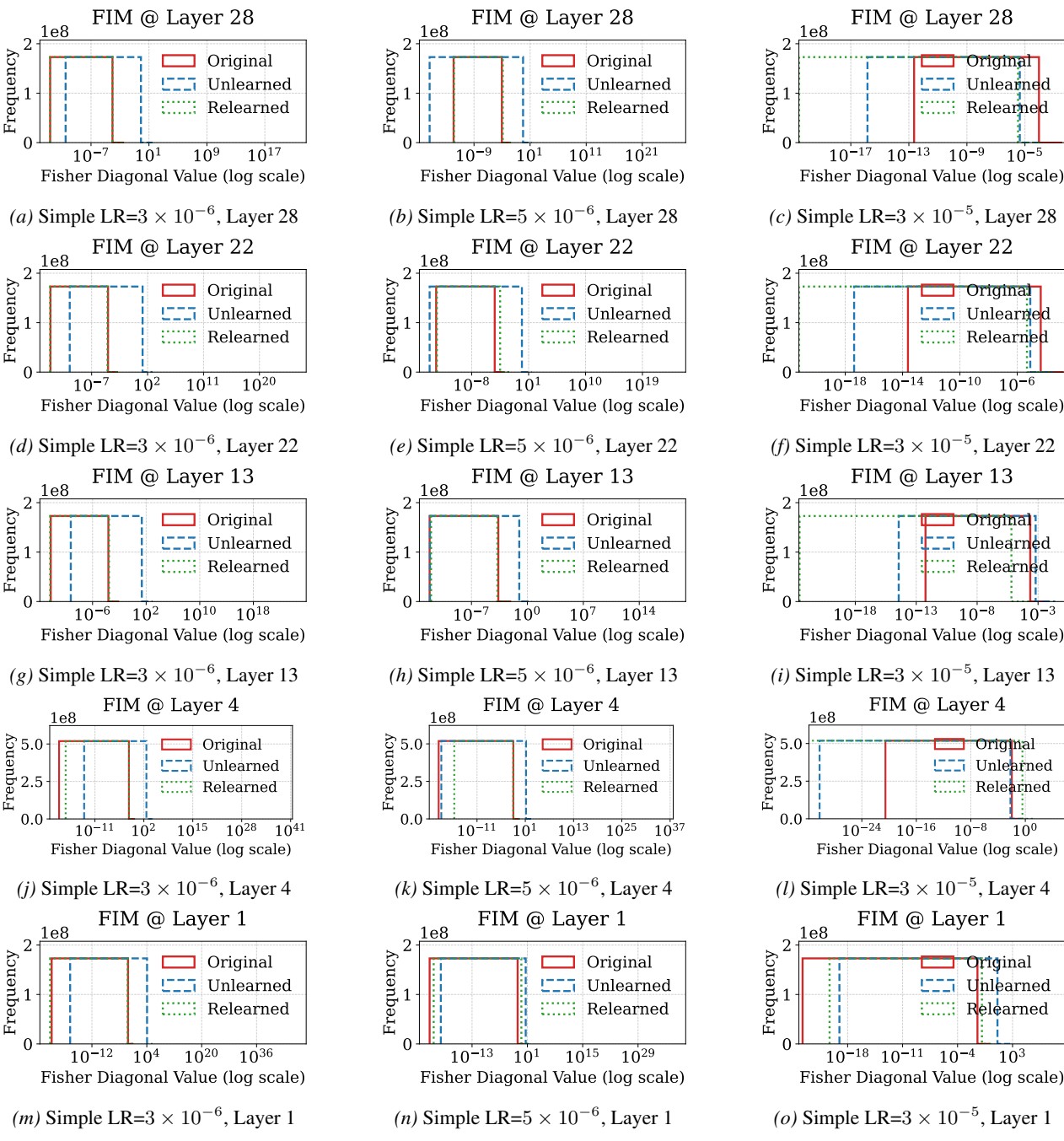

*Figure 19.* FIM for GA Across Layers. All plots are for the simple task on Yi-6B, using three learning rates $\{3 \times 10^{-6},\ 5 \times 10^{-6},\ 3 \times 10^{-5}\}$ and fixed $N = 100$.

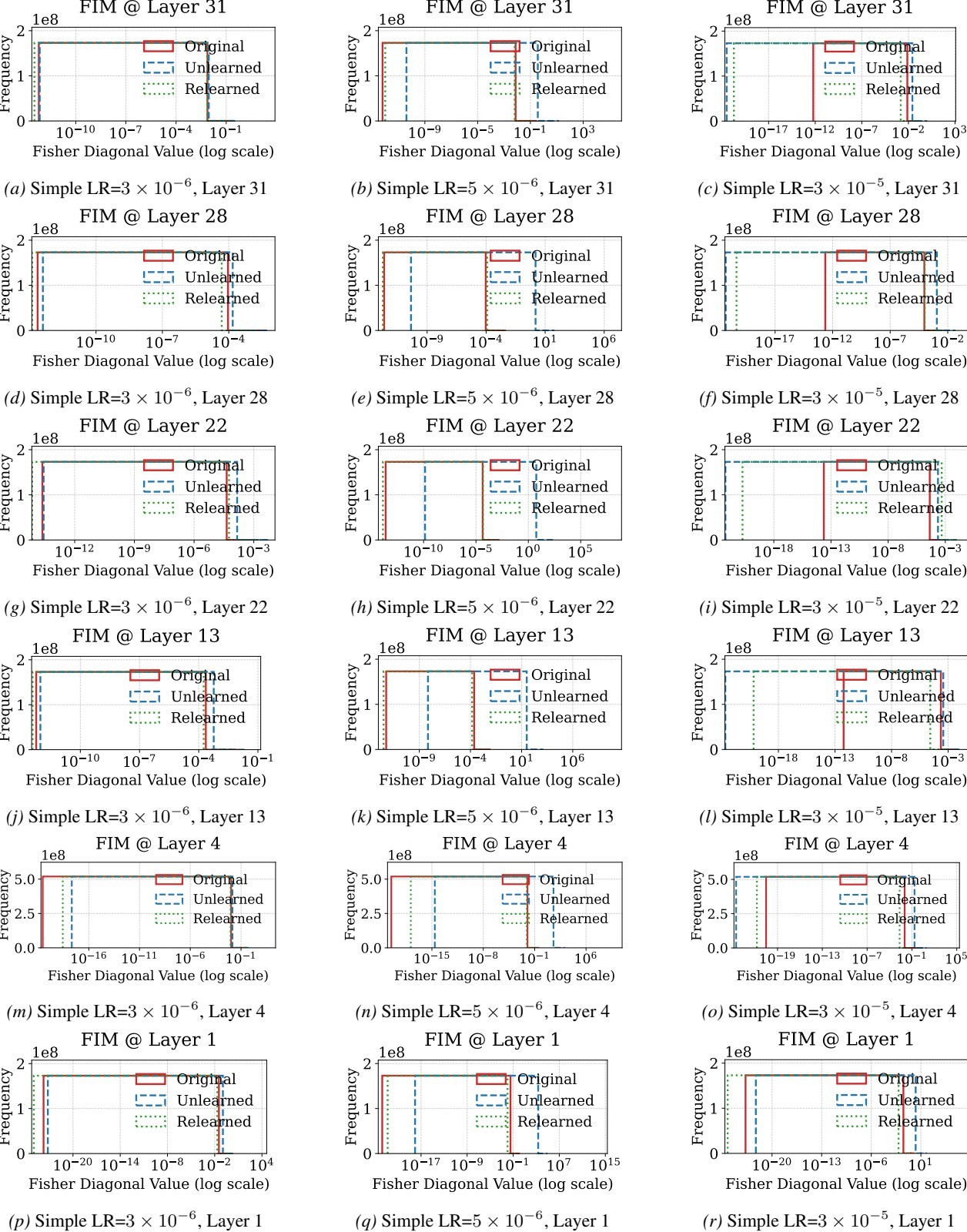

*Figure 20.* FIM for GA+GD Across Layers. All plots are for the simple task on Yi-6B, using three learning rates $\{3 \times 10^{-6}, 5 \times 10^{-6}, 3 \times 10^{-5}\}$ and fixed $N = 100$.

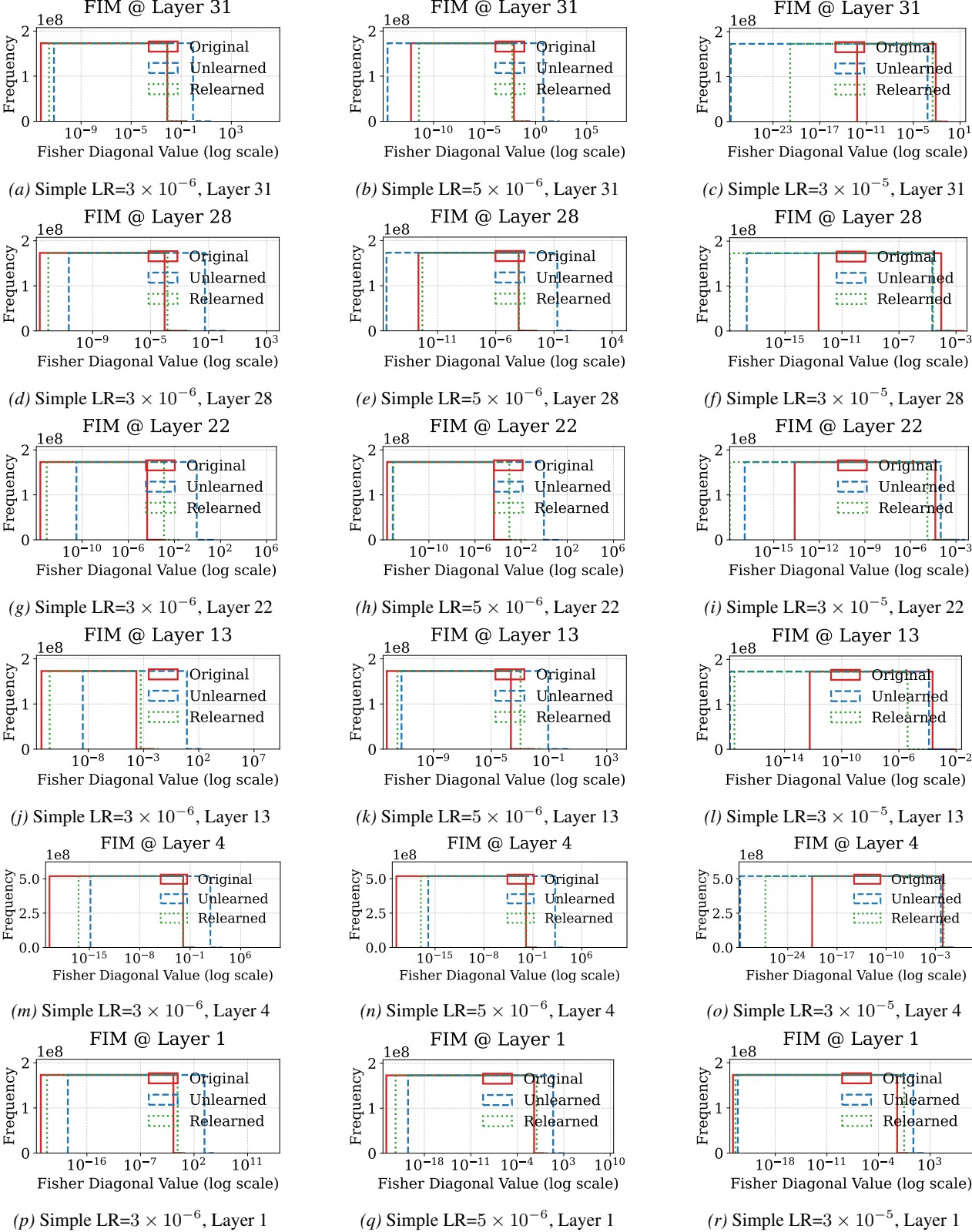

*Figure 21.* FIM for GA+KL Across Layers. All plots are for the simple task on Yi-6B, using three learning rates $\{3 \times 10^{-6}, 5 \times 10^{-6}, 3 \times 10^{-5}\}$ and fixed $N = 100$.

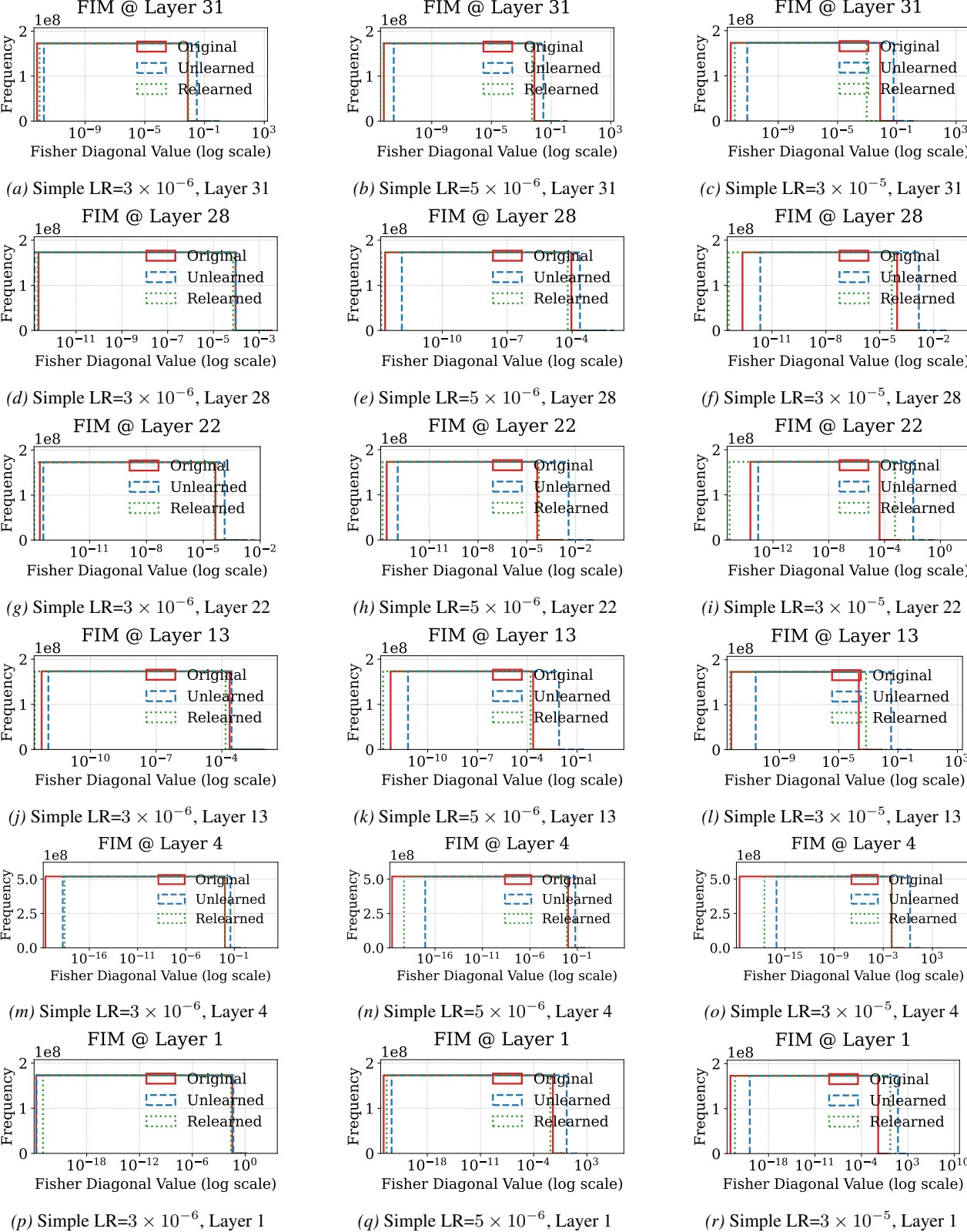

*Figure 22.* FIM for NPO Across Layers. All plots are for the simple task on Yi-6B, using three learning rates $\{3 \times 10^{-6}, 5 \times 10^{-6}, 3 \times 10^{-5}\}$ and fixed $N = 100$.

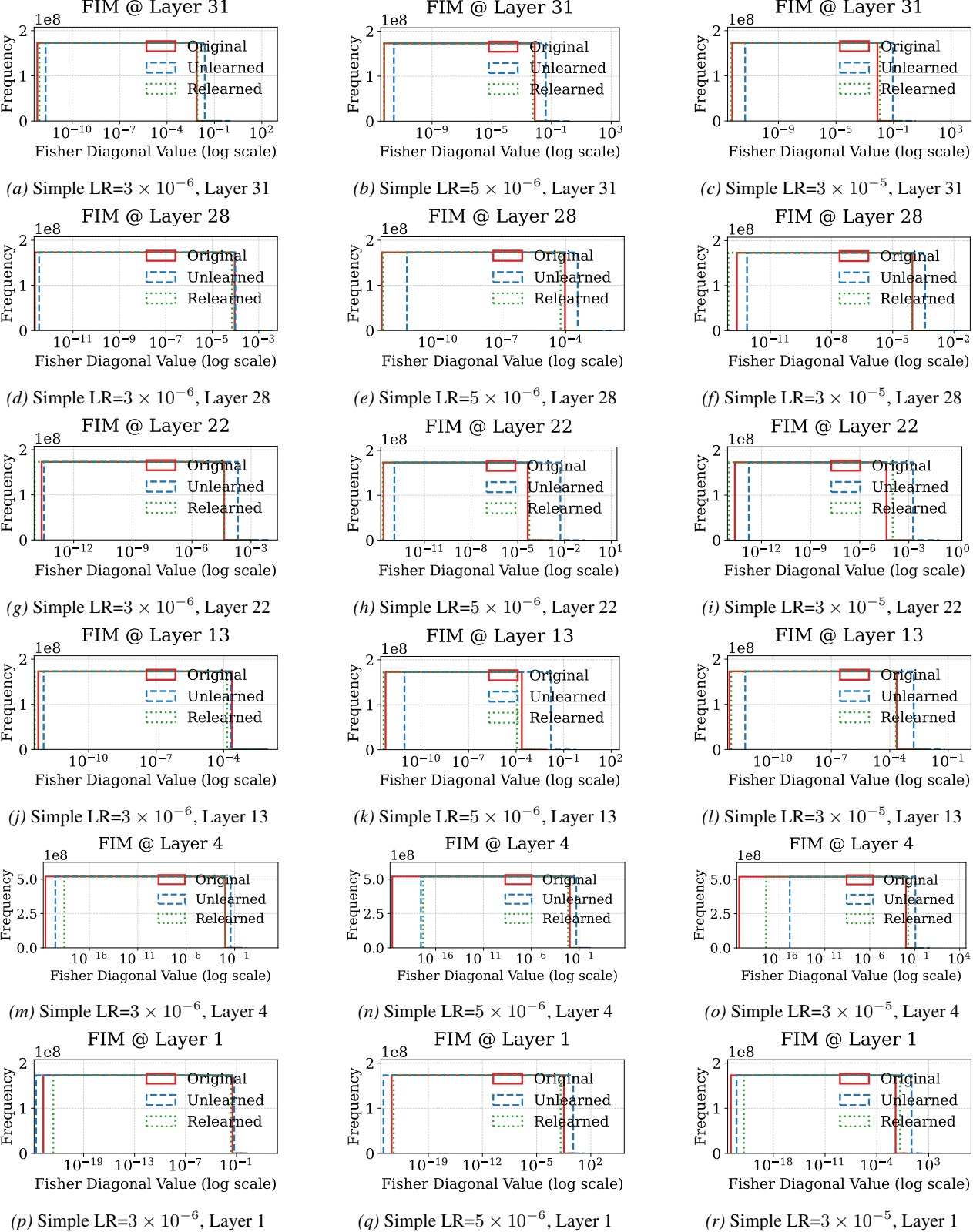

*Figure 23.* FIM for NPO+KL Across Layers. All plots are for the simple task on Yi-6B, using three learning rates $\{3 \times 10^{-6}, 5 \times 10^{-6}, 3 \times 10^{-5}\}$ and fixed $N = 100$.

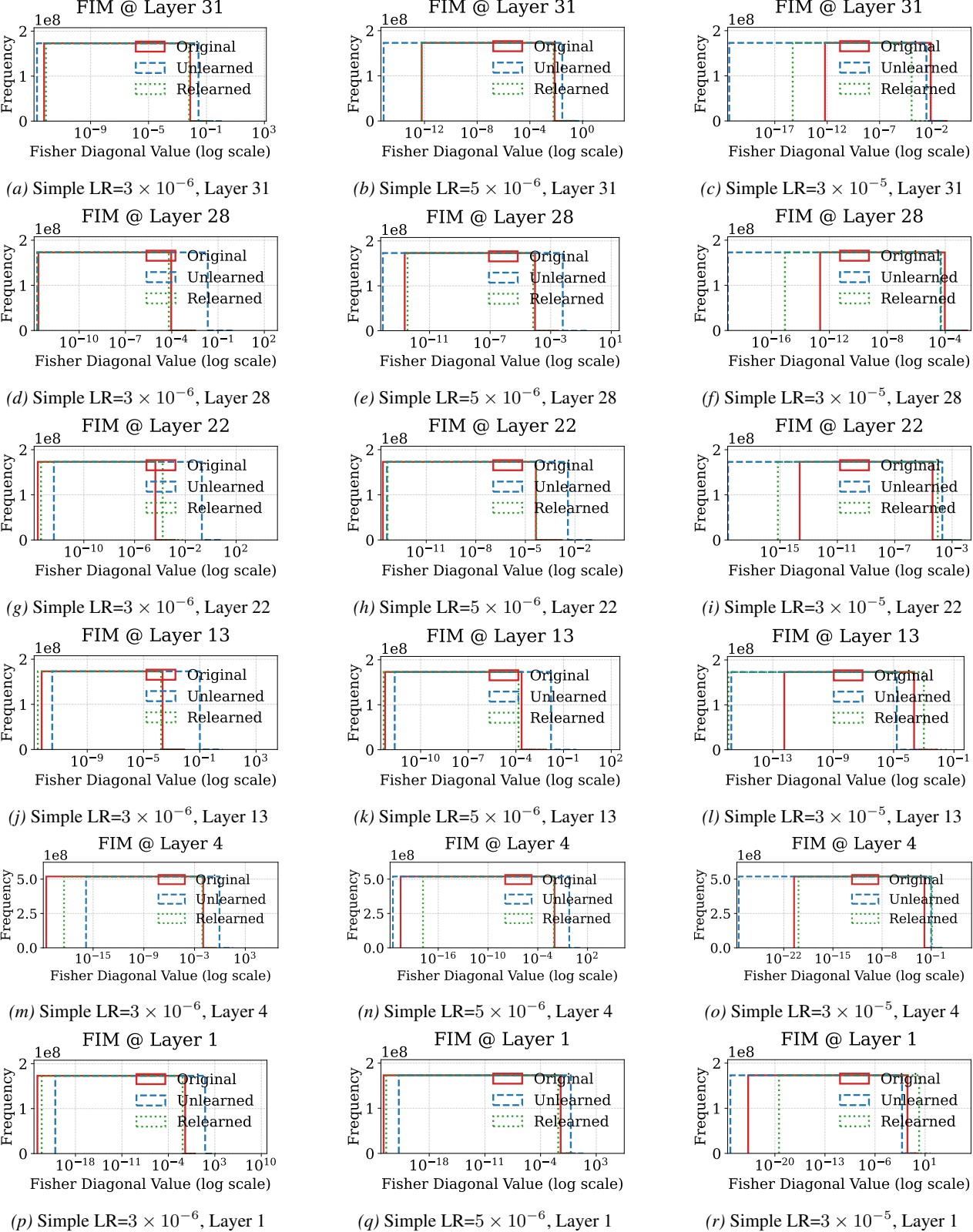

*Figure 24.* FIM for Rlable Across Layers. All plots are for the simple task on Yi-6B, using three learning rates $\{3 \times 10^{-6}, 5 \times 10^{-6}, 3 \times 10^{-5}\}$ and fixed $N = 100$.

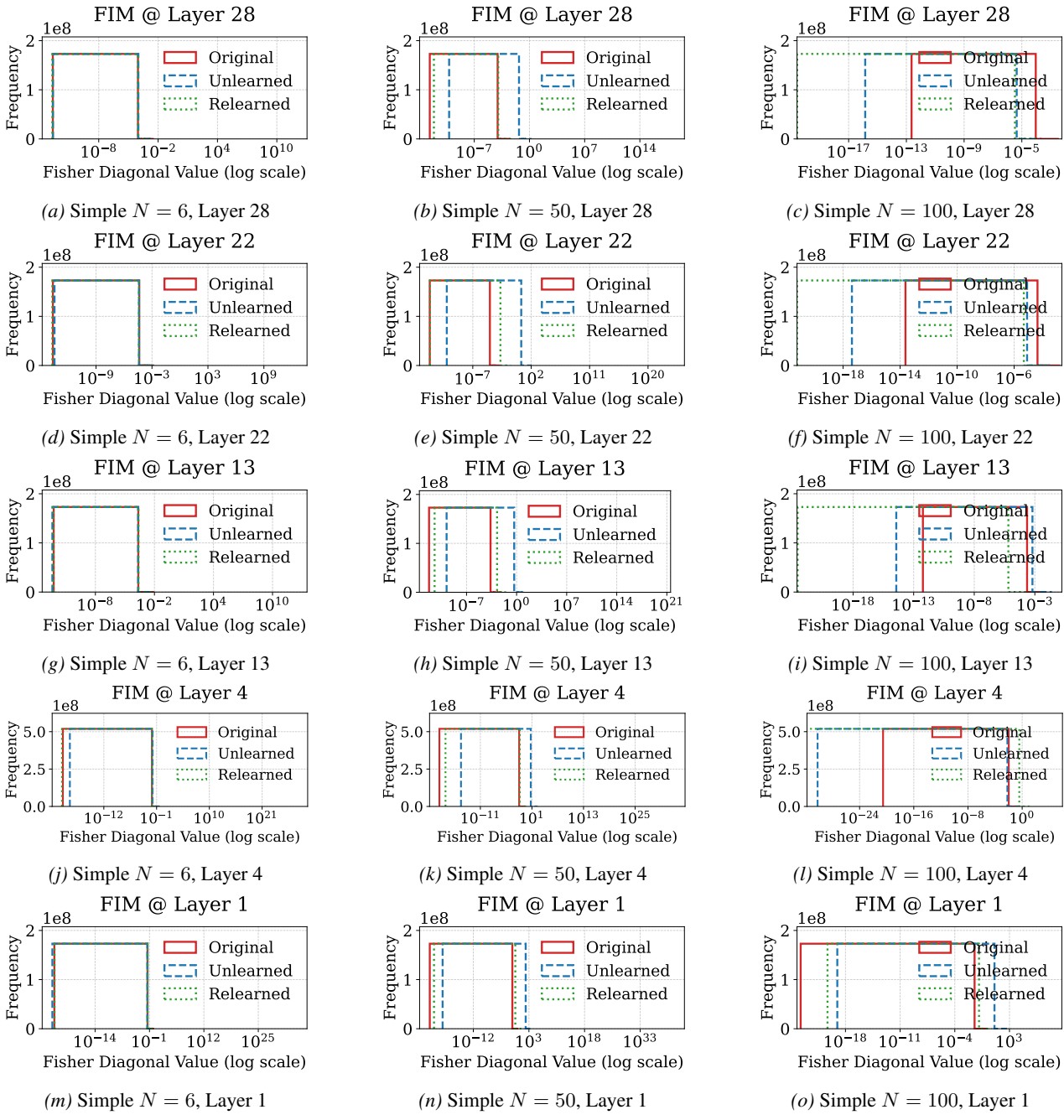

*Figure 25.* FIM for GA Across Layers. Simple task on Yi-6B with fixed learning rate LR $= 3 \times 10^{-5}$ and varying unlearning requests $N \in \{6, 50, 100\}$.

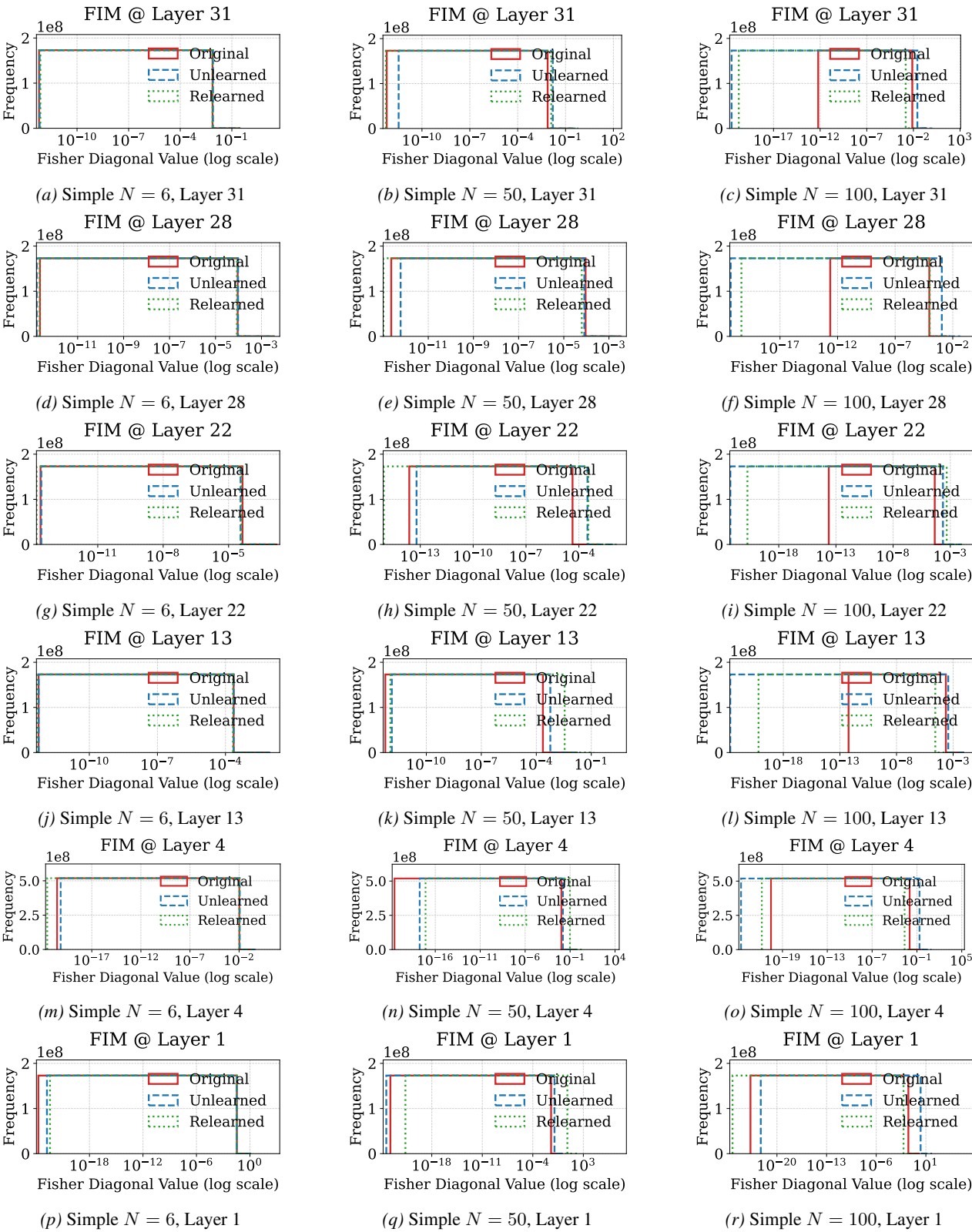

*Figure 26.* FIM for GA+GD Across Layers. Simple task on Yi-6B with fixed learning rate LR $= 3 \times 10^{-5}$ and varying unlearning requests $N \in \{6, 50, 100\}$.

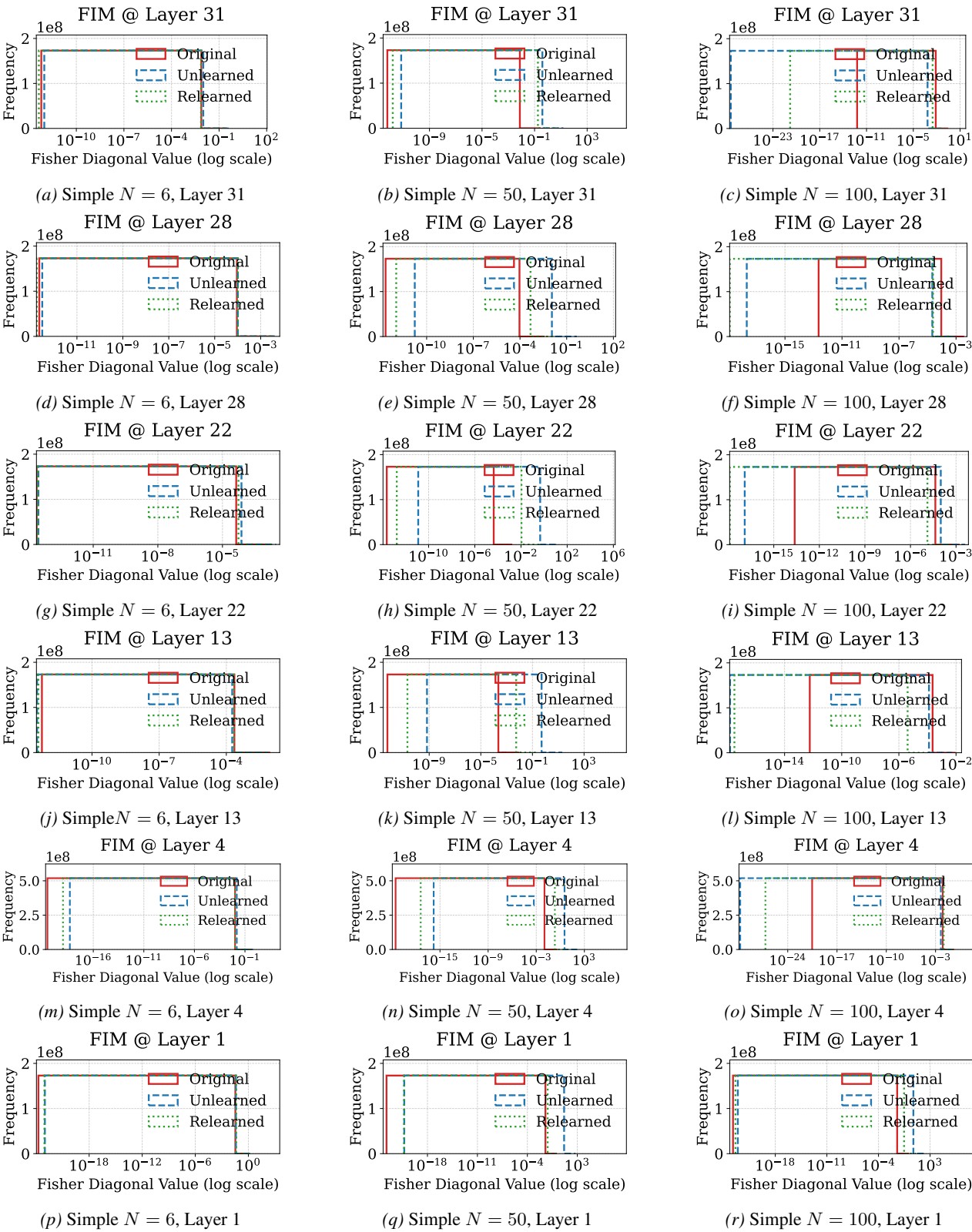

*Figure 27.* FIM for GA+KL Across Layers. Simple task on Yi-6B with fixed learning rate LR = $3 \times 10^{-5}$ and varying unlearning requests $N \in \{6, 50, 100\}$.

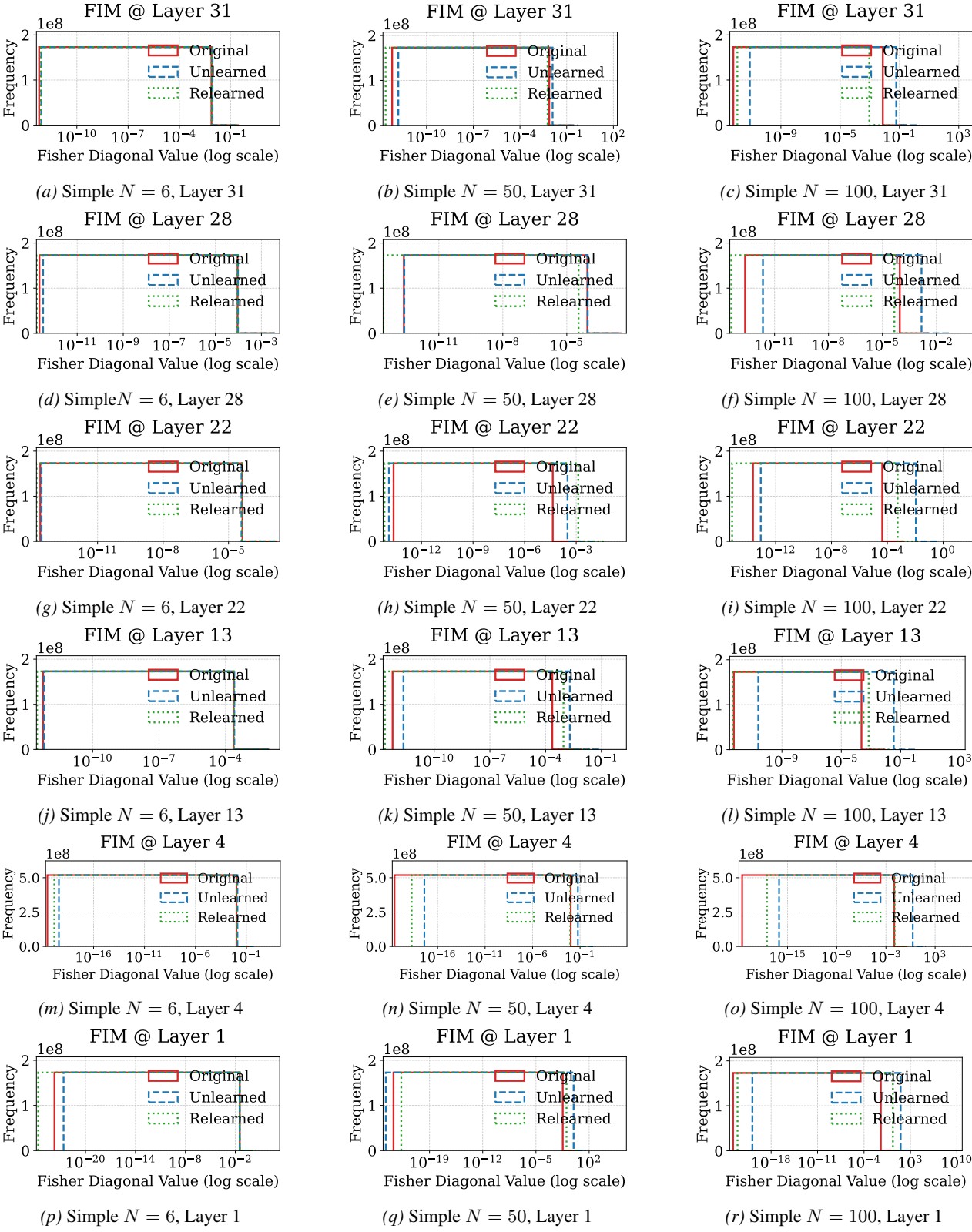

*Figure 28.* FIM for NPO Across Layers. Simple task on Yi-6B with fixed learning rate $\mathrm{LR} = 3 \times 10^{-5}$ and varying unlearning requests $N \in \{6, 50, 100\}$.

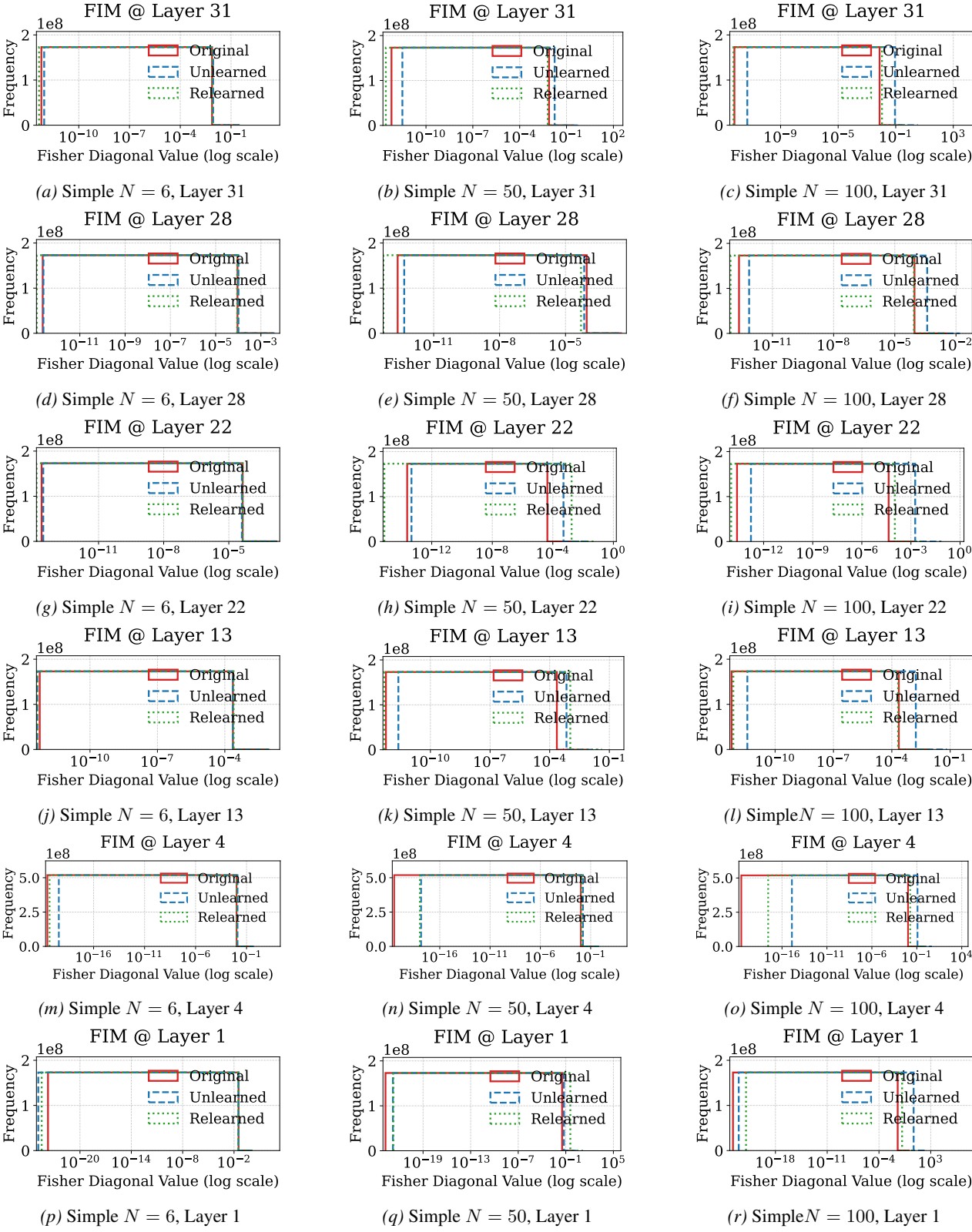

*Figure 29.* FIM for NPO+KL Across Layers. Simple task on Yi-6B with fixed learning rate LR $= 3 \times 10^{-5}$ and varying unlearning requests $N \in \{6, 50, 100\}$.

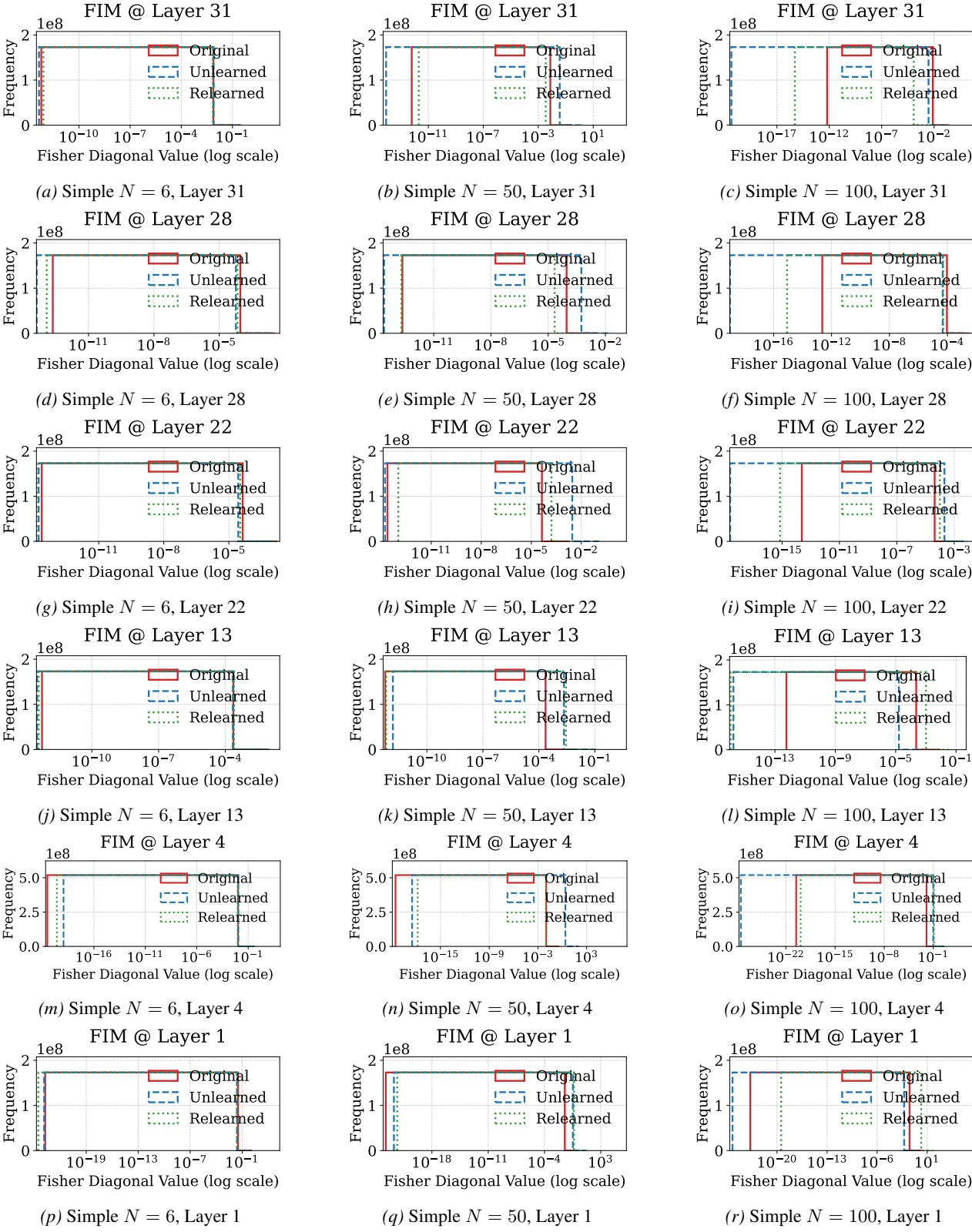

*Figure 30.* FIM for Rlable Across Layers. Simple task on Yi-6B with fixed learning rate LR $= 3 \times 10^{-5}$ and varying unlearning requests $N \in \{6, 50, 100\}$.

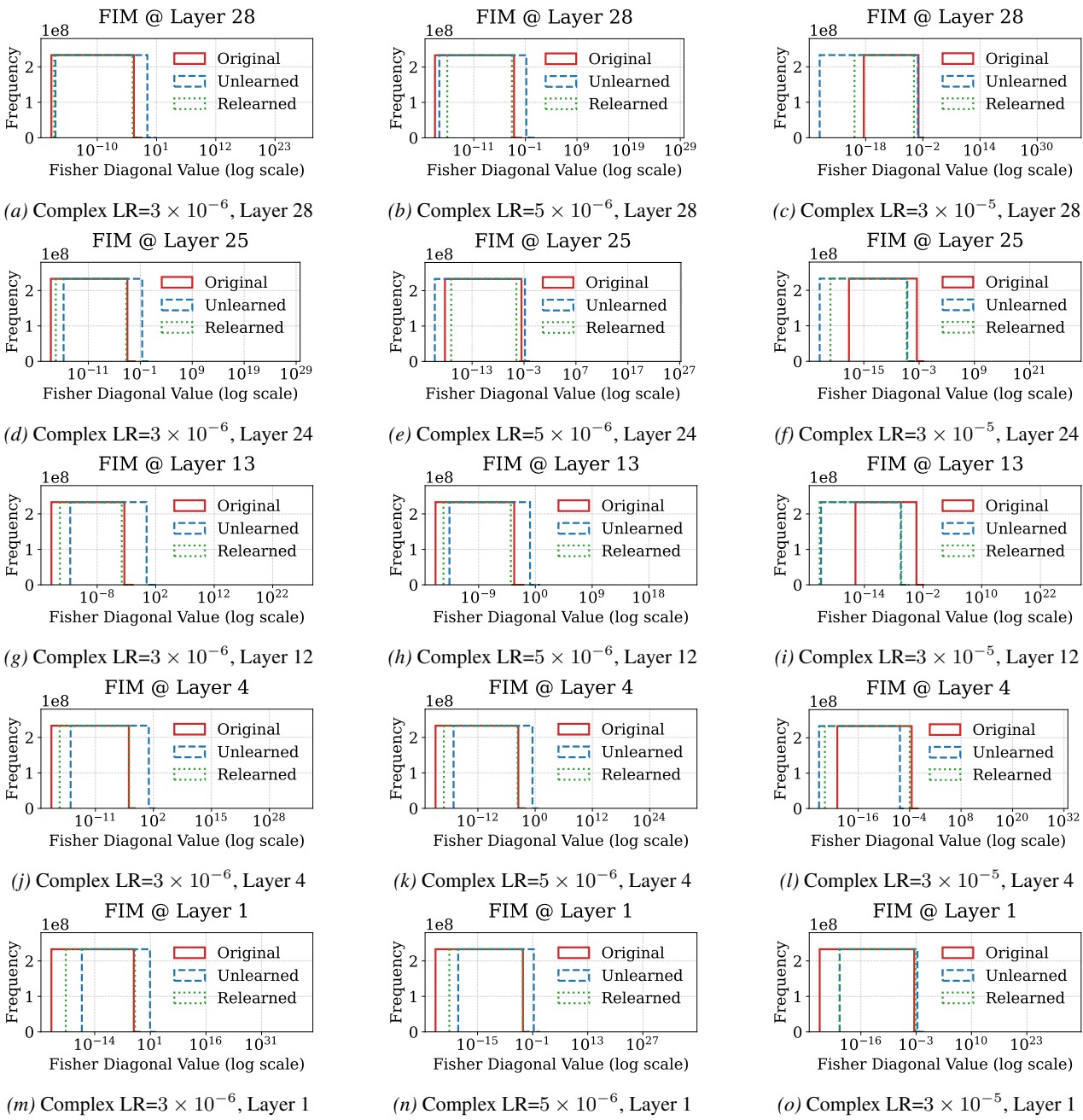

*Figure 31.* FIM for GA Across Layers. All plots are for the complex task on Qwen2.5-7B, using three learning rates $\{3 \times 10^{-6}, 5 \times 10^{-6}, 3 \times 10^{-5}\}$ and fixed $N = 6$.

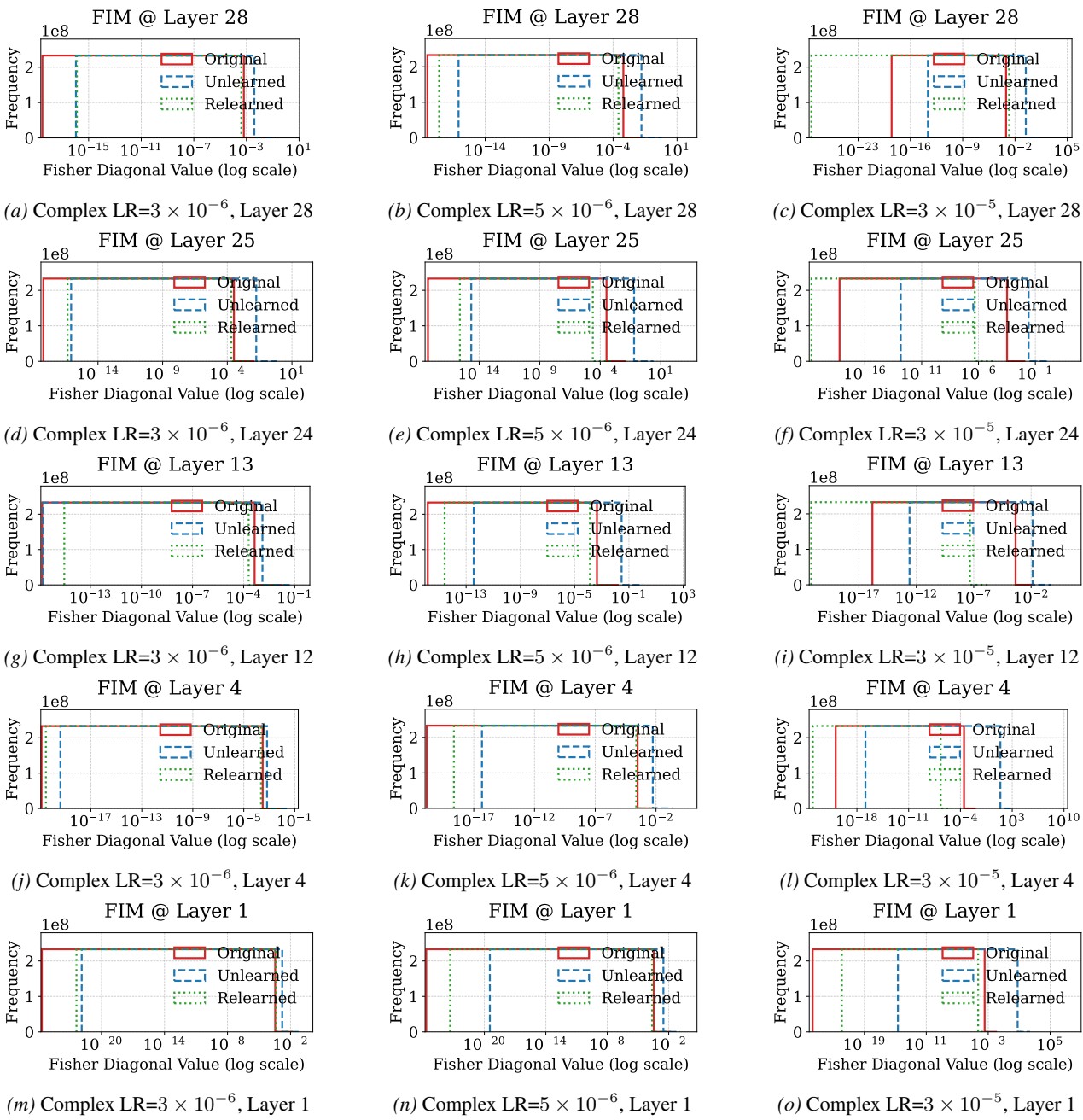

*Figure 32.* FIM for NPO Across Layers. All plots are for the complex task on Qwen2.5-7B, using three learning rates $\{3 \times 10^{-6}, 5 \times 10^{-6}, 3 \times 10^{-5}\}$ and fixed $N = 6$.

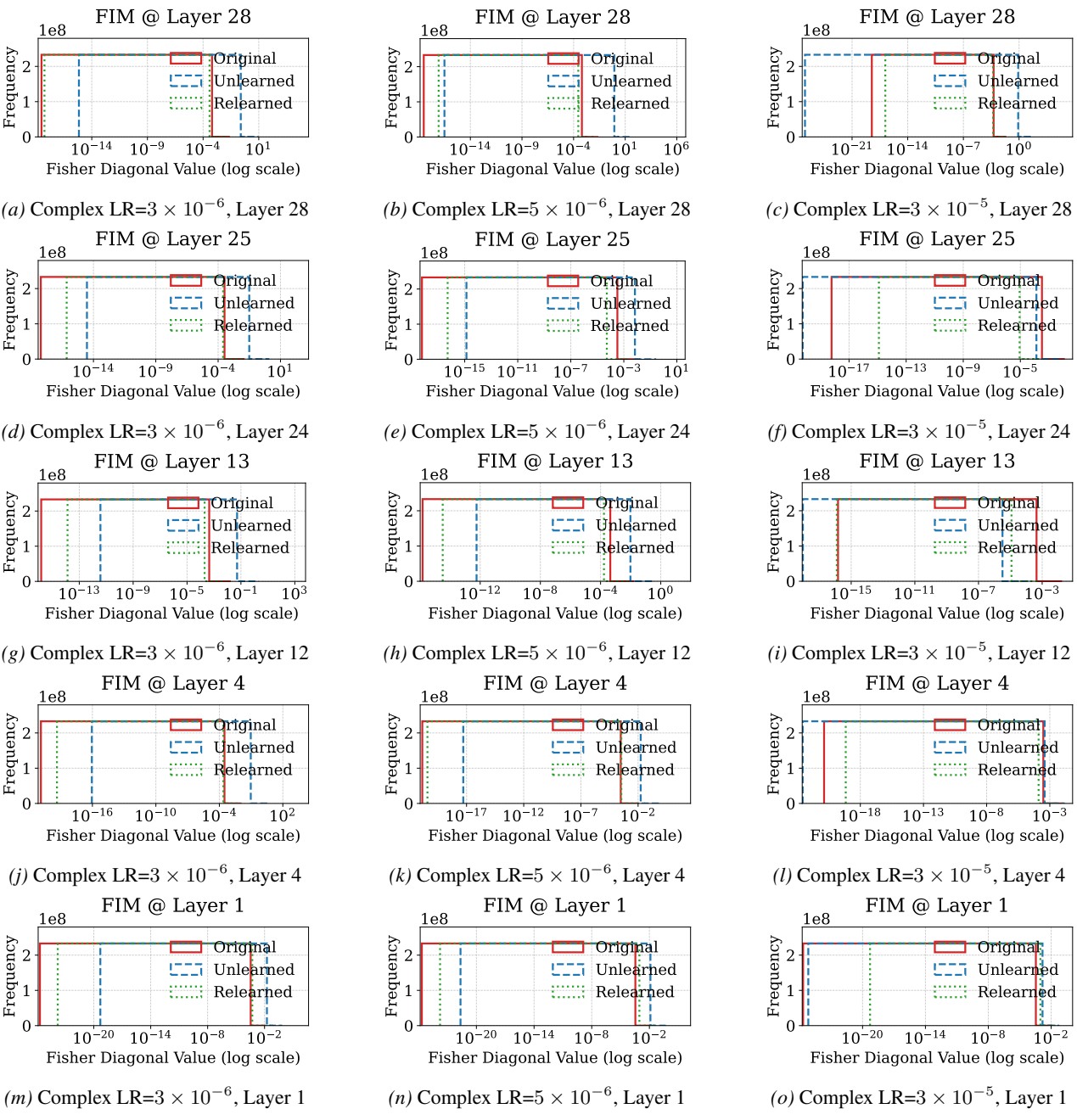

*Figure 33.* FIM for Rlable Across Layers. All plots are for the complex task on Qwen2.5-7B, using three learning rates $\{3 \times 10^{-6}, 5 \times 10^{-6}, 3 \times 10^{-5}\}$ and fixed $N = 6$.

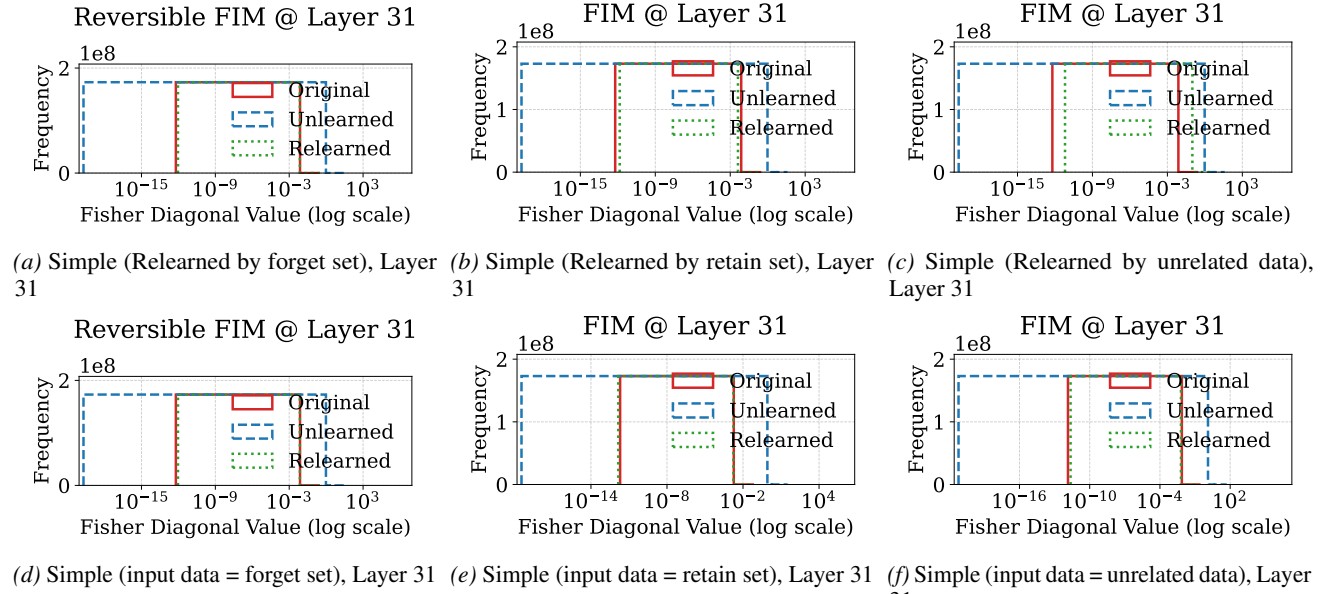

*(a)* Simple (Relearned by forget set), Layer 31

*(b)* Simple (Relearned by retain set), Layer 31

*(c)* Simple (Relearned by unrelated data), Layer 31

*(d)* Simple (input data = forget set), Layer 31

*(e)* Simple (input data = retain set), Layer 31

*(f)* Simple (input data = unrelated data), Layer 31

*Figure 34.* FIM in layer 31 under Varied Relearning and Evaluation Inputs on Yi-6B (Simple Task). (a–c): Relearning is performed using the forget set, retain set, or unrelated data respectively. (d–f): FIM is measured using the forget set, retain set, or unrelated data as evaluation input.

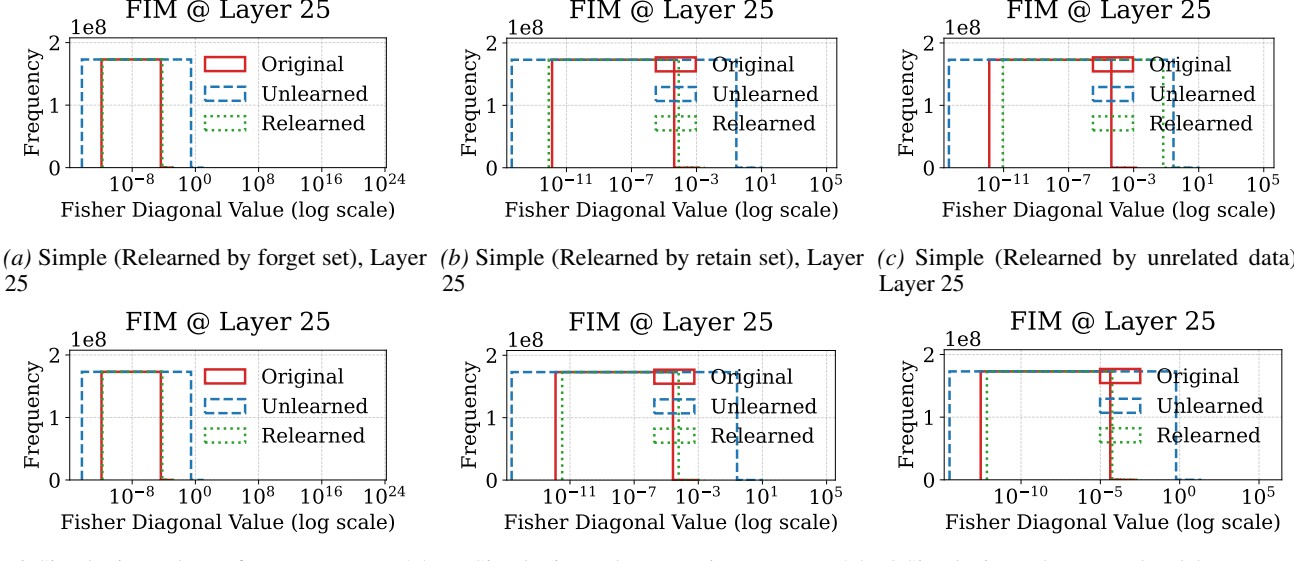

*(a)* Simple (Relearned by forget set), Layer 25

*(b)* Simple (Relearned by retain set), Layer 25

*(c)* Simple (Relearned by unrelated data), Layer 25

*(d)* Simple (input data = forget set), Layer 25

*(e)* Simple (input data = retain set), Layer 25

*(f)* Simple (input data = unrelated data), Layer 25

*Figure 35.* FIM in layer 25 under Varied Relearning and Evaluation Inputs on Yi-6B (Simple Task). (a–c): Relearning is performed using the forget set, retain set, or unrelated data respectively. (d–f): FIM is measured using the forget set, retain set, or unrelated data as evaluation input.

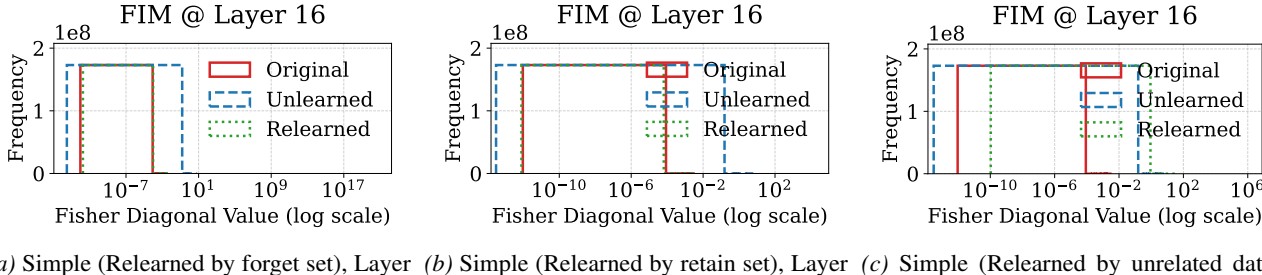

*(a)* Simple (Relearned by forget set), Layer 16

*(b)* Simple (Relearned by retain set), Layer 16

*(c)* Simple (Relearned by unrelated data), Layer 16

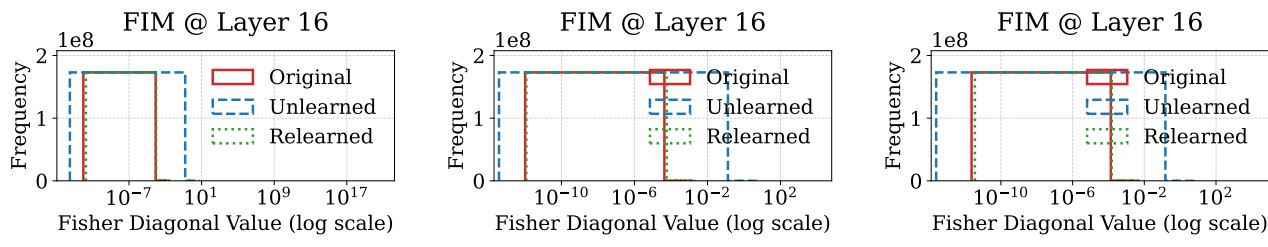

*(d)* Simple (input data = forget set), Layer 16

*(e)* Simple (input data = retain set), Layer 16

*(f)* Simple (input data = unrelated data), Layer 16

*Figure 36.* FIM in layer 16 under Varied Relearning and Evaluation Inputs on Yi-6B (Simple Task). (a–c): Relearning is performed using the forget set, retain set, or unrelated data respectively. (d–f): FIM is measured using the forget set, retain set, or unrelated data as evaluation input.

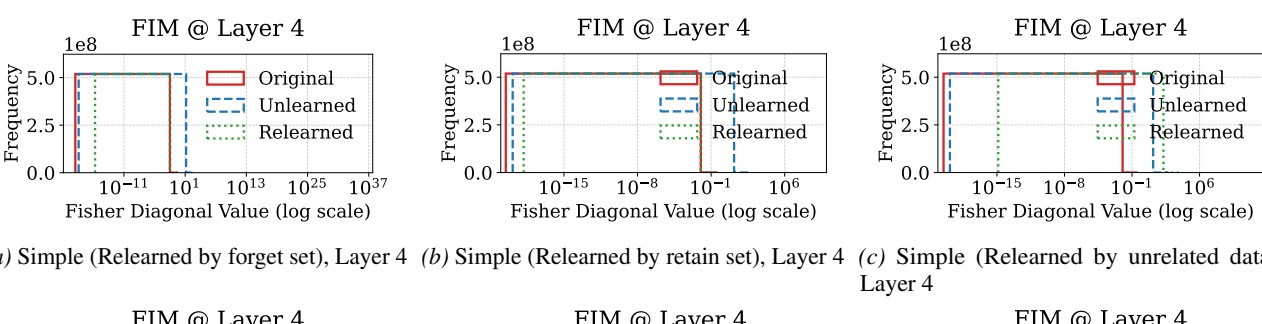

*(a)* Simple (Relearned by forget set), Layer 4

*(b)* Simple (Relearned by retain set), Layer 4

*(c)* Simple (Relearned by unrelated data), Layer 4

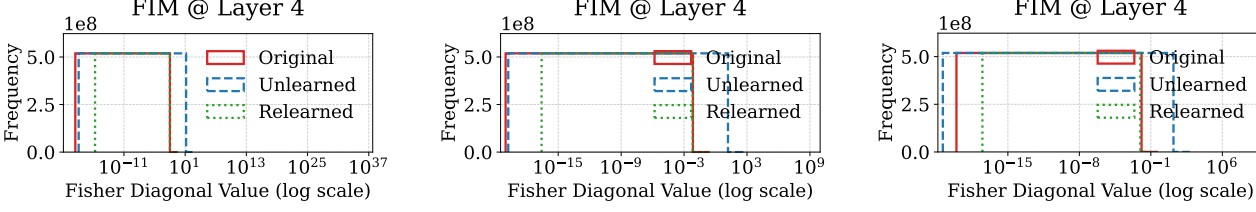

*(d)* Simple (input data = forget set), Layer 4

*(e)* Simple (input data = retain set), Layer 4

*(f)* Simple (input data = unrelated data), Layer 4

*Figure 37.* FIM in layer 4 under Varied Relearning and Evaluation Inputs on Yi-6B (Simple Task). (a–c): Relearning is performed using the forget set, retain set, or unrelated data respectively. (d–f): FIM is measured using the forget set, retain set, or unrelated data as evaluation input.

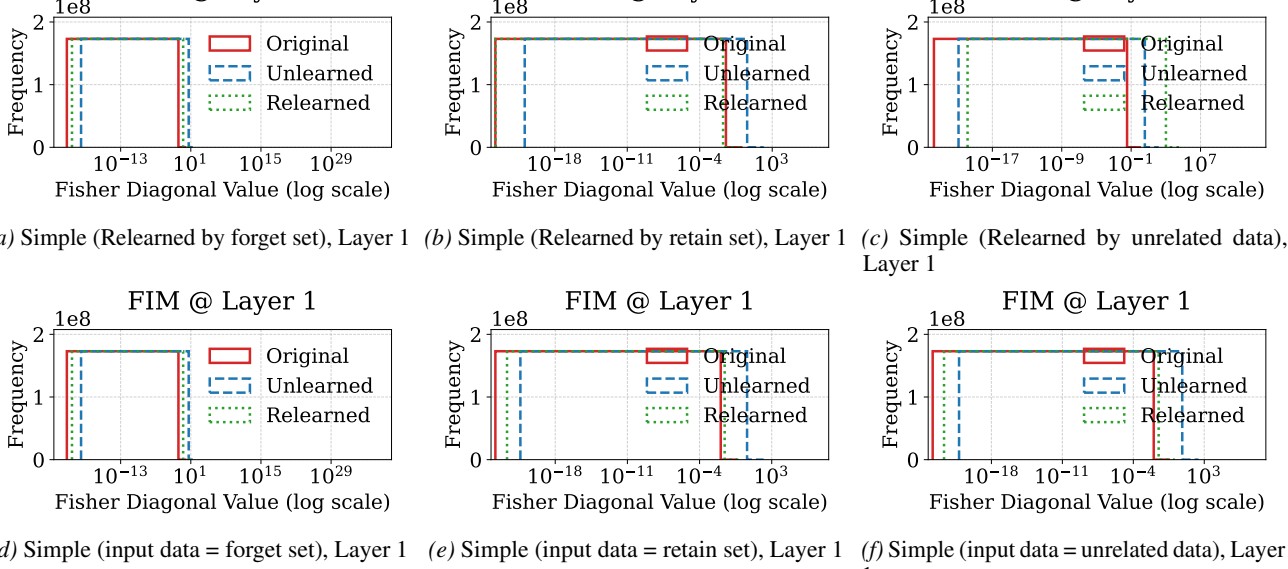

*(a)* Simple (Relearned by forget set), Layer 1    *(b)* Simple (Relearned by retain set), Layer 1    *(c)* Simple (Relearned by unrelated data), Layer 1

*(d)* Simple (input data = forget set), Layer 1    *(e)* Simple (input data = retain set), Layer 1    *(f)* Simple (input data = unrelated data), Layer 1

*Figure 38.* FIM in layer 1 under Varied Relearning and Evaluation Inputs on Yi-6B (Simple Task). (a–c): Relearning is performed using the forget set, retain set, or unrelated data respectively. (d–f): FIM is measured using the forget set, retain set, or unrelated data as evaluation input.

