# OpenReview forum: "Unlearning Isn't Deletion: Investigating Reversibility of Machine Unlearning in LLMs"
_ICML.cc/2026/Conference — ICML 2026 regular_

### Official Review · Reviewer_5fBz · 2026-03-06

**Soundness:** 3
**Presentation:** 3
**Significance:** 2
**Originality:** 2
**Overall Recommendation:** 4
**Confidence:** 3

**Summary:**

The paper investigates the efficacy of current machine unlearning methods for LLM. The authors argue that traditional task-level metrics (e.g., accuracy and perplexity) are insufficient for verifying genuine data erasure, as the suppressed knowledge can often be recovered via minimal fine-tuning. To address this, the authors propose a representation-level analysis framework using tools such as PCA similarity/shift, CKA, and FIM. They establish a taxonomy of four forgetting regimes based on reversibility and catastrophicity.

**Compliance With Llm Reviewing Policy:**

Affirmed.

**Final Justification:**

The paper is well presented and includes comprehensive experiments. The authors’ responses have addressed my concerns, so I am willing to adjust my score accordingly.

**Key Questions For Authors:**

Please refer to Weaknesses.

**Limitations:**

Yes

**Strengths And Weaknesses:**

**Strengths**

The shift from surface-level task metrics to internal representational dynamics is timely and critical for the field of trustworthy unlearning.

The use of a constrained relearning protocol as a diagnostic probe, rather than just checking post-unlearning accuracy, effectively exposes the "fragility" of current methods.


**Weaknesses**

The paper is well written and the experiments are thorough, but I still have some concerns.

1. Existing work [1] also proposed a similar idea, where data with different representation (long-Term Memory) is hard to be forgotten. The authors should include them in related work and comparison.  While the paper provides a systematic analysis, the overall contribution is somewhat incremental. Similar observations regarding the reversibility of unlearning and the limitations of task-level metrics have been previously discussed in recent literature

2. It would be convincing to propose a new unlearning method/framework where consider the metrics proposed in your work.

3. It is not novel that consider representation-level unlearning [2,3]. Also it is not reasonable that the authors did not include RMU[3] which is a commonly used unleraning method and a representation-level method as a baseline in their work.

4. The author identified a single case (using GA+GD+WAGLE) that showed signs of "irreversible, non-catastrophic" forgetting. Could you elaborate on why weight attribution (WAGLE) specifically contributes to this irreversibility? Is it because it targets the "feature" neurons rather than just the "logit" layers?

5. The paper acknowledges that defining precise thresholds for distinguishing between regimes is non-trivial and depends on task complexity, leaving a slight ambiguity for practitioners wanting to apply the framework.


[1] https://arxiv.org/abs/2510.03263

[2] https://openreview.net/forum?id=KzSGJy1PIf

[3] https://arxiv.org/pdf/2403.03218

---

> ### Author Rebuttal · Authors · 2026-03-31
>
> Thanks for your thoughtful feedback. We appreciate recognition of our framework's value:  **the shift to representation-level evaluation** and **use of constrained relearning to expose unlearning fragility**.
>
> **#W1**
> We agree that [1] is relevant and should be discussed.
>
> [1] studies **text-to-image unlearning** via memory self‑regeneration, showing that some visual concepts are harder to forget and easier to recover, and distinguishing short‑ vs. long‑term memory. Our work is complementary but differs in scope and focus:
>
> - **Setting:** [1] addresses diffusion‑based text‑to‑image models, whereas we study **LLM unlearning**, where forgetting is reflected in **text generation and token prediction**, and where evaluation has relied almost on task‑level metrics.
>
> - **Goal:** [1] studies heterogeneous recoverability of visual concepts. We aim to provide a systematic taxonomy of reversibility and catastrophicity in LLM unlearning, across single and continual deletion, and across data types.
>
> - **Methodology:** Prior work [a,b] on LLM unlearning (discussed in Lines 653-709) shows that unlearned behaviors can often be recovered, but treats reversibility mainly as a **behavioral phenomenon**. Our contribution is to introduce a **representation‑level diagnostic toolkit** and a **perturbation‑based analysis** to characterize what changes inside models, distinguishing superficial suppression from genuine representational drift.
>
> We'll include [1] in the related work, clarify how our representation‑level analysis extends prior observations on reversibility and the limitations of task‑level metrics for LLM unlearning.
>
> **#W2**
> We agree that using our metrics to design new unlearning methods is a valuable next step. Nevertheless, our current focus: i) show **why task-level evaluation is insufficient**, and ii) develop a representation‑level **diagnostic framework** to distinguish **reversibility** from **catastrophicity**. Directly turning these diagnostics into training objectives is currently non‑trivial, as the regime boundaries are **method-, task-, and model-dependent**, and our metrics are calibrated for **analysis** rather than **optimization**.
>
> Please also see our response to **Reviewer YUKm #W2**.
>
> **#W3**
> We agree that representation-level unlearning methods, such as domain-adversarial training in [2] and RMU in [3], are established in the literature. Yet, our contribution lies in a fundamentally different direction:
>
> - **Evaluation vs. Methodology**: While [2] and [3] propose algorithms to execute unlearning, our work introduces a diagnostic framework to evaluate it.
>
> - **Mechanistic Focus**: We don't propose a new representation-level update rule. Instead, we utilize PCA similarity/shift, CKA, and FIM to mechanistically expose whether the forgetting achieved by any given method is genuinely durable or merely superficial (i.e., reversible vs. catastrophic).
>
> We agree that RMU [3] is a highly relevant baseline. We've now added RMU to our extended evaluation (see response to **Reviewer BdDZ #W4**). The new results remain fully consistent with our main conclusions.
>
> **#W4**
> Rather than strictly targeting "feature" neurons over "logit" layers, our analysis indicates this near-ideal regime emerges from synergy of **precise weight localization** and **controlled update magnitude**.
>
> Specifically:
>
> - **Selective Localization (Update Direction)**: WAGLE avoids dense updates, selectively modifying internal weights highly specific to the forget set. This disrupts deep knowledge structures rather than superficially masking the output layer.
>
> - **Calibrated Step Size (Update Magnitude)**:  Localization alone is insufficient; overly large steps still cause collateral damage. A tuned learning rate ensures the update is strong enough to permanently erase target knowledge, yet mild enough to prevent global representational collapse.
>
> In short, WAGLE optimizes the update direction while the learning rate controls its magnitude. Together, they achieve durable, localized forgetting without the severe damage of standard GA updates.
>
> **#W5**
> We agree that defining universal numerical thresholds is challenging, as boundaries naturally vary across models and tasks.
>
> Hence, our framework aims to provide **relative diagnostic signals**, not rigid decision rules, combining behavioral recovery with representational drift:
>
> - **Reversible Forgetting**: High behavioral recovery + minimal post-relearning drift.
>
> - **Irreversible Forgetting**: Weak recovery + persistently large drift.
>
> Even without "absolute cutoffs," practitioners can reliably use these paired metrics as a comparative tool: to easily benchmark algorithms, tune hyperparameters, and objectively evaluate whether unlearning is merely superficial, catastrophic, or genuinely durable.
>
> [a] Unlearning or Obfuscating? Jogging the Memory of Unlearned LLMs via Benign Relearning. ICLR 2025.
>
> [b] Erase or Hide? Suppressing Spurious Unlearning Neurons for Robust Unlearning. ICLR 2026.

---

> > ### Author Rebuttal · Reviewer_5fBz · 2026-04-03
> >
> > I thank the authors for their response to my concerns. I will adjust the score.

---

> > > ### Author Response · Authors · 2026-04-03
> > >
> > > We sincerely thank the reviewer for the positive follow-up and greatly appreciate the raised score.

---

### Official Review · Reviewer_BdDZ · 2026-03-09

**Soundness:** 2
**Presentation:** 3
**Significance:** 4
**Originality:** 2
**Overall Recommendation:** 4
**Confidence:** 4

**Summary:**

This paper argues that current LLM unlearning methods mostly suppress information rather than genuinely erase it, and that standard task-level metrics (accuracy, perplexity) are insufficient to detect this. The authors propose a representation-level diagnostic toolkit (PCA similarity/shift, CKA, Fisher information, mean PCA distance) to measure internal representational drift after unlearning. They formalize a taxonomy of four forgetting regimes along two axes — reversibility and catastrophicity — and provide a perturbation-theoretic analysis linking the magnitude and distribution of weight changes to whether forgetting is reversible or permanent.

**Compliance With Llm Reviewing Policy:**

Affirmed.

**Final Justification:**

Overall, the paper tackles an important question, and the rebuttal addresses my main concerns. I'm recommending acceptance with my current score.

**Key Questions For Authors:**

Weaknesses.

**Limitations:**

Yes

**Strengths And Weaknesses:**

The core question — whether unlearning actually erases information or just suppresses it — is important and timely. Showing that task-level metrics can be misleading has direct implications for anyone deploying unlearning in practice. The existence of these issues have been established previously, though.

**Weaknesses:**

- **The proposed metrics don't clearly separate regimes in the figures.** In many plots (e.g., Figures 2–4), the PCA similarity, CKA, and FIM curves for unlearned vs. relearned models look nearly identical, particularly in the reversible regimes. There are no confidence intervals, p-values, or other statistical tests to confirm that the observed differences are meaningful. This raises the question: how much discriminative power do these metrics actually add over standard task-level evaluation? The paper would benefit from a rigorous statistical comparison showing when and where the representation-level metrics diverge from task-level ones in a statistically significant way.

- **Practical applicability has real gaps despite not requiring retraining.** To be fair, the toolkit does not require the prohibitively expensive retrain-from-scratch baseline — all diagnostics compare the original model against the unlearned (and optionally relearned) model, and the relearning probe is cheap since it's size-matched to the forget set. If you have original and unlearned weights, you can compute PCA similarity, CKA, and FIM on any probe set immediately: near-unity CKA on the forget set is a red flag that forgetting is superficial, which is actionable without any relearning. However, the paper never provides calibrated thresholds for distinguishing regimes. The authors acknowledge this (Section 5.3): the boundary between reversible and irreversible depends on method, task, and model family. A practitioner seeing a mean PCA distance of 15 has no way to know which regime they're in without running the full relearning experiment themselves. More critically, the paper doesn't address the external auditing scenario, where only the unlearned model is available and the original weights are not. Every metric is defined relative to the original model's activations, so a regulator trying to verify GDPR compliance — arguably the most policy-relevant use case — gets nothing from this toolkit. The methodology is useful as an internal tool for comparing unlearning methods during development, but it falls short as a standalone certification mechanism.

- **The theoretical analysis doesn't match the experimental setup.** Section 5 models unlearning as perturbations to a feedforward network f(x) = σ(W_L σ(···σ(W_1 x)···)). The actual experiments use transformers with attention, residual connections, and layer normalization, all of which fundamentally change how perturbations propagate through layers. The paper then invokes a Neumann-series expansion of the perturbed output but never defines the expansion properly, states convergence conditions, or derives it from first principles. The theoretical claims appear in big-O notation without the underlying math being established. This section feels disconnected from the rest of the paper and would need substantially more rigor to support the conclusions drawn from it.

- **Important recent methods are missing from the evaluation.** The six methods tested (GA, GA+GD, GA+KL, NPO, NPO+KL, RLabel) all belong to the gradient-ascent or preference-based families. Several well-known methods from different paradigms are absent, including RMU [1], which steers representations toward random targets; UnDIAL [2], which uses self-distillation with adjusted logits; AltPO [3], which combines negative and positive feedback through preference optimization; and PDU [4], which formulates unlearning as constrained primal-dual optimization. These cover representation engineering, distillation, and constrained optimization — fundamentally different approaches whose inclusion would be necessary to support the paper's claims about the generality of its taxonomy.

- The diagnostic tools themselves (PCA similarity, CKA, FIM) are all borrowed from prior work. The novelty is in assembling them, not in the tools themselves.

**References:**

[1] Li N, Pan A, Gopal A, Yue S, Berrios D, Gatti A, Li JD, Dombrowski AK, Goel S, Phan L, Mukobi G. The wmdp benchmark: Measuring and reducing malicious use with unlearning. arXiv preprint arXiv:2403.03218. 2024 Mar 5.


[2] Dong YR, Lin H, Belkin M, Huerta R, Vulić I. Undial: Self-distillation with adjusted logits for robust unlearning in large language models. InProceedings of the 2025 Conference of the Nations of the Americas Chapter of the Association for Computational Linguistics: Human Language Technologies (Volume 1: Long Papers) 2025 Apr (pp. 8827-8840).

[3] Mekala A, Dorna V, Dubey S, Lalwani A, Koleczek D, Rungta M, Hasan SA, Lobo E. Alternate preference optimization for unlearning factual knowledge in large language models. InProceedings of the 31st International Conference on Computational Linguistics 2025 Jan (pp. 3732-3752).

[4] Entesari T, Hatami A, Khaziev R, Ramakrishna A, Fazlyab M. Constrained Entropic Unlearning: A Primal-Dual Framework for Large Language Models. arXiv preprint arXiv:2506.05314. 2025 Jun 5.

---

> ### Author Rebuttal · Authors · 2026-03-31
>
> Appreciate your thoughtful feedback and praises: **the importance and practical relevance of the core question** and **significance of showing that task-level metrics can be misleading**.
>
> New results of Tables*1-*2:
> `https://anonymous.4open.science/r/Reviewer_BdDZ-38DC/README.md`
>
> **#W1**
> We'd like to clarify that our analysis is quantitative, not merely visual:
>
> - **Statistical Stability & Direct Comparison**: To show statistical robustness, Table 4 reports **mean ± standard deviation across four random seeds**. Moreover, our new Table*1 explicitly places it alongside corresponding **task-level metrics** for GA under identical settings.
>
> - **Discriminative Power over Task-Level Metrics**: Task-level metrics often collapse identically regardless of whether knowledge is erased or merely suppressed: only revealing the truth after relearning. Conversely, PCA distance provides a predictive **pre-relearning signal**. At a high learning rate $3\times10^{-5}$, the pre-relearning PCA distance is large (predicting irreversible collapse); at lower rates $5\times10^{-6}$, $3\times10^{-6}$, it remains small (predicting observed strong recovery).
>
> - **CKA and FIM**: We present them as unaggregated layer-wise curves to localize where structural and loss-landscape changes occur across the network's depth.
>
> We'll emphasize these statistical comparisons and clarify our metrics' predictive power.
>
> **#W2**
> Thanks for the accurate summary. We agree: it's designed as an **internal diagnostic toolkit** for method development, not a standalone certification mechanism for external auditing.
>
> To address specific limitations:
>
> - **Universal Thresholds**: As regime boundaries depend heavily on the specific model, task, and unlearning method (Sect-5.3), we don't prescribe one-size-fits-all cutoffs. Instead, the metrics are intended for **relative comparison** and early robustness testing during algorithm design.
>
> - **External Auditing**: We acknowledge that requiring original model weights precludes black-box regulatory auditing. Yet, comparing states before and after unlearning is the standard paradigm for developing and evaluating unlearning methods [a,b].
>
> - **Practical Efficiency**: As correctly noted, the toolkit is highly practical for developers. It avoids prohibitively expensive retrain-from-scratch baselines (e.g., by utilizing proxies like TOFU retain model [b]), and metrics like CKA can provide immediate, actionable "red flags" without even running the relearning probe.
>
> We'll explicitly clarify this intended scope in revision.
>
> **#W3**
> Our intention was not to provide a formal proof, but rather an **illustrative, heuristic perturbation analysis**. This conceptual toy model serves only to provide an intuition for our core empirical findings: minor weight updates can drastically distort final task-level outputs (logits) while leaving deeper internal representations largely intact. We acknowledge that the current framing overstates the mathematical rigor of this section.
>
> We'll extensively reframe Sect-5 to properly reflect its scope. Specifically:
>
> - Explicitly recharacterize the analysis as an intuitive conceptual illustration rather than an architecture-complete derivation.
>
> - Remove/clarify unsupported formalisms (e.g., unqualified Neumann-series expansions and big-O notation).
>
> - Soften the theoretical claims to ensure this heuristic motivation supports, rather than distracts from, our primary empirical contributions.
>
> **#W4**
> Thanks for suggesting these baselines. To ensure our taxonomy generalizes across diverse paradigms, we've extended our evaluation to include **RMU** (representation engineering), **UnDIAL** (distillation), **AltPO** (preference optimization), and **PDU** (constrained optimization)
>
> The new results (**Table*2**) confirm that our core findings hold across these fundamentally different approaches:
>
> - **Reversible Regime (Low LR: $3\times10^{-6}$, $5\times10^{-6}$)**: All four methods exhibit strong task-level recovery post-relearning, accurately predicted by small pre-relearning PCA distances.
>
> - **Irreversible Regime (High LR: $3\times10^{-5}$)**: Post-relearning recovery drops significantly while PCA distances remain large. Notably, our metrics reveal that UnDIAL and AltPO show greater structural robustness under aggressive hyperparameters, avoiding total representational collapses in GA.
>
> **#W5**
> We agree that PCA, CKA, and FIM are established tools. Our contribution lies in synthesizing them into a ** diagnostic framework** to evaluate LLM unlearning reversibility.
>
> As task-level metrics are often misleading, assembling this suite is necessary to capture **complementary views** of representational drift.
>
> Please also see our **response to Reviewer p5G2 (#W1 & #Q2)** for a more detailed justification.
>
> [a] Machine Unlearning of Pre-trained Large Language Models. ACL 2024.
>
> [b] TOFU: A Task of Fictitious Unlearning for LLMs. COLM 2024.

---

> > ### Author Rebuttal · Reviewer_BdDZ · 2026-04-03
> >
> > With the rebuttal and the new additions and revisions to the paper, I am inclined to suggest the paper for acceptance. Thank you.

---

> > > ### Author Response · Authors · 2026-04-03
> > >
> > > Thank you very much for your positive assessment and for considering our paper for acceptance. We sincerely appreciate your time, thoughtful feedback, and engagement throughout the review process.

---

### Official Review · Reviewer_p5G2 · 2026-03-10

**Soundness:** 3
**Presentation:** 3
**Significance:** 3
**Originality:** 2
**Overall Recommendation:** 4
**Confidence:** 4

**Summary:**

In the paper "Unlearning Isn't Deletion: Investigating Reversibility of Machine Unlearning in LLMs," the authors challenge the current reliance on task-level metrics (such as accuracy and perplexity) to evaluate machine unlearning in Large Language Models (LLMs). They demonstrate that these metrics can be deceptive: a model may show a complete performance collapse on a forget set, suggesting successful unlearning, while the underlying knowledge remains latent and easily recoverable through minimal fine-tuning. The authors introduce a representation-level diagnostic toolkit (PCA similarity/shift, CKA, Fisher information) and propose a taxonomy of four forgetting regimes along reversibility and catastrophicity axes. Experiments span six unlearning methods, two LLMs, and three data domains.

**Compliance With Llm Reviewing Policy:**

Affirmed.

**Final Justification:**

I thank the authors for their thoughtful and detailed responses. The rebuttal addresses several concerns meaningfully, but some issues remain partially unresolved. Therefore, my ratings remain unchanged. Here is my assessment:

#W1 & #Q2: The authors provide a reasonable qualitative argument for why each metric captures a complementary dimension. The four-point breakdown (PCA similarity → global alignment; PCA shift → translational drift; CKA → subspace preservation; FIM → parameter sensitivity) is intuitive. However, the rebuttal stops short of providing stronger justification of why these four metrics are necessary and sufficient versus alternatives.

W3 & Q1: partially resolved, but the remaining concerns are not easily addressed in a short rebuttal.

#W2 & #Q3: largely resolved

**Key Questions For Authors:**

1. Can the authors provide a formal or intuitive explanation for why MIA AUC rebounds under irreversible catastrophic forgetting? A satisfactory theoretical account would significantly strengthen the paper's case against MIA as an evaluation metric.

2. Can the authors demonstrate cases where any single metric or a subset of three metrics would lead to incorrect regime classification, and where the full four-metric suite is strictly necessary?

3. Do the authors have any theoretical or empirical evidence suggesting their four-regime taxonomy and the diagnostic thresholds generalize to models at the 70B+ scale or to instruction-tuned/RLHF-aligned variants?

**Limitations:**

Yes

**Strengths And Weaknesses:**

Strengths:

1. The paper introduces a multi-faceted diagnostic toolkit (PCA similarity/shift, CKA, Fisher Information) that provides a holistic view of model internal state changes.
2. The paper features an extensive empirical evaluation across six methods, multiple LLMs (Yi, Qwen, Llama), and diverse data domains (arXiv, code, math)
3. Provides a weight perturbation model that explains why small updates can cause accuracy collapse while preserving internal feature geometry.
4. The four-regime framework (reversible/irreversible × catastrophic/non-catastrophic) is clean, well-motivated, and provides a useful conceptual vocabulary for future work.

Weaknesses:

1. PCA-based representation analysis, CKA, and FIM are all established tools. The contribution is their combination for unlearning evaluation. The paper would benefit from stronger justification of why these four metrics are necessary and sufficient versus alternatives.
2. Empirical validation is restricted to intermediate-sized models (up to 8B parameters), leaving the dynamics in frontier-scale LLMs (>70B) unexplored.
3. The paper identifies Membership Inference Attacks (MIA) as unreliable in the irreversible catastrophic regime. However, the theoretical explanation for this specific phenomenon is thin. Given that this finding undermines a widely used privacy evaluation standard, the erratic behavior of MIA deserves a dedicated theoretical analysis.

---

> ### Author Rebuttal · Authors · 2026-03-31
>
> We thank you for the constructive feedback. We're greatly encouraged that you found value in our **multi-faceted diagnostic toolkit**, **breadth of our empirical evaluation**, **intuition from our weight perturbation analysis**, and **usefulness of our four-regime framework**.
>
> Below, we address your remaining concerns. The additional experiment results referenced as Table*1 are at: `https://anonymous.4open.science/r/Reviewer_p5G2-E611/README.md`
>
> **#W1 & #Q2**
> Thanks for the insightful comment. While the metrics used are established, our contribution lies in their integration into a **joint diagnostic framework** to overcome the limitations of misleading task-level scores.
>
> We justify the four-metric suite as a necessity for capturing **complementary dimensions** [a,b,c]:
>
> - **PCA similarity**: global alignment of dominant directions,
>
> - **PCA shift**: translational displacement in representation space,
>
> - **CKA**: preservation of the activation subspace,
>
> - **FIM**: parameter sensitivity and loss-landscape changes not visible from activations alone.
>
> Our claim is practical rather than absolute: we do not assert that every single metric, or every subset of three, is always incorrect. Rather, our experiments show that:
>
> - Task (single or a subset of) metrics alone can lead to "regime misclassification," e.g., in Table 3 (GA, $LR=5\times10^{-6}$), task-level accuracy collapses to 9.1%, suggesting "true forgetting."
>
> - PCA similarity alone can understate the drift that PCA shift reveals.
>
> - PCA-only views can miss subspace changes captured by CKA.
>
> - Activation geometry can appear partially recovered while FIM indicates persistent, irreversible parameter-level damage.
>
> We'll explicitly clarify these in revision.
>
> **#W2 & #Q3**
> Thanks for highlighting the importance of evaluating frontier-scale LLMs. While scaling to 70B+ models remains an important future direction due to computational constraints, we've extended our validation to a larger model (Qwen3-14B) and an instruction-tuned variant (Llama-3-8B-Instruct).
>
> Our new results (Table*1) show that our four-regime taxonomy generalizes well beyond smaller, base models. Specifically, we observe the same qualitative patterns:
>
> - At **lower learning rates**, relearning recovery is robust and post-relearning PCA drift is minimal, indicating a **more reversible** regime.
>
> - At **higher learning rates**, recovery significantly weakens and representation drift remains large, indicating a **less reversible** regime.
>
> - Methodologically, NPO consistently proves more robust than GA across settings, yielding better recovery and smaller PCA drift.
>
> Because these trends hold across both the 14B and the instruction-tuned models, we believe our taxonomy captures a fundamental phenomenon in LLM unlearning rather than an artifact of model scale. While the precise diagnostic thresholds (e.g., specific learning rates) will naturally vary by model and setting, the structural progression of these forgetting regimes remains highly consistent.
>
> **#W3 # Q1**
> As our MIA evaluation relies on **Min-K% Prob** [d], it fundamentally measures the statistical separability of token probabilities, rather than true knowledge deletion.
> The erratic rebound of the MIA AUC in the irreversible catastrophic regime stems from two interacting effects:
>
> - **Surface-level probability sensitivity（Sect-5.2**: Output logits are highly sensitive to parameter perturbations. Min-K% Prob tracks these distorted output confidences, which can shift dramatically even if underlying features are just corrupted.
>
> - **Asymmetric probability increasing**: In catastrophic regime, the model's accuracy drops globally. After relearning using forget set, yet, the token-probability distributions increase **asymmetrically**: forget-set probabilities typically increase far more quickly than retain-set probabilities. As MIA is a relative discrimination task; this emerging probability gap makes the sets easily separable again, artificially inflating the AUC score.
>
> In short, a rebounding MIA AUC in this regime does not indicate robust privacy preservation; it merely reflects catastrophic, asymmetric model relearning that inadvertently widens the probability gap between the two sets. We'll add this dedicated explanation to strengthen our critique of probability-based MIA metrics.
>
> [a] Spurious Forgetting in Continual Learning of Language Models. ICLR 2025.
>
> [b] Similarity of Neural Network Representations Revisited. ICML 2019.
>
> [c] Towards Robust and Cost-Efficient Knowledge Unlearning for Large Language Models. ICLR 2025.
>
> [d] Detecting Pretraining Data from Large Language Models. ICLR 2024.

---

> > ### Author Rebuttal · Reviewer_p5G2 · 2026-04-01
> >
> > I thank the authors for their thoughtful and detailed responses. The rebuttal addresses several concerns meaningfully, but some issues remain partially unresolved. Therefore, my ratings remain unchanged. Here is my assessment:
> >
> > #W1 & #Q2: The authors provide a reasonable qualitative argument for why each metric captures a complementary dimension.  The four-point breakdown (PCA similarity → global alignment; PCA shift → translational drift; CKA → subspace preservation; FIM → parameter sensitivity) is intuitive. However, the rebuttal stops short of providing stronger justification of why these four metrics are necessary and sufficient versus alternatives.
> >
> > W3 & Q1: partially resolved, but the remaining concerns are not easily addressed in a short rebuttal.
> >
> > #W2 & #Q3: largely resolved

---

> > > ### Author Response · Authors · 2026-04-02
> > >
> > > We sincerely thank you for the quick, constructive follow-up. Appreciate that our response to #W2 and #Q3 largely addresses your concern.
> > >
> > > **#W1 & #Q2**
> > > We don't claim formal sufficiency over all possible representation-level metrics (e.g., SVCCA [e]). Instead, we justify our four-metric suite as a compact, empirically effective, and theoretically necessary toolkit to operationalize the perturbation framework in Section 5:
> > >
> > > i) **The insufficiency of alternatives (Section 5.2)**:
> > >
> > > Our theoretical analysis demonstrates why task-level metrics are misleading. Small weight perturbations near output layers can cause sharp accuracy drops, falsely implying true erasure even when deeper representations remain entirely intact.
> > >
> > > ii) **The necessity of our suite (Section 5.1)**:
> > >
> > > When **perturbations are small and localized**, forgetting remains **reversible**, whereas **distributed perturbations across many layers** accumulate into **irreversible representational drift**. Our four metrics are necessary to probe the specific dimensions of this drift:
> > >
> > > - FIM is necessary to track underlying changes to the loss landscape, distinguishing temporary parameter suppression from permanent erasure.
> > >
> > > - CKA is necessary to measure whether distributed errors have structurally fractured the internal activation subspace.
> > >
> > > - PCA similarity & shift are jointly necessary to disambiguate whether representations have fundamentally altered their principal directions, or just experienced recoverable translational displacement.
> > >
> > > Omitting any of them compromises our ability to reliably distinguish among the four forgetting regimes (e.g., mistaking reversible translational shift for irreversible subspace collapse); see Section 4.2.2.
> > >
> > > We'll explicitly frame our suite as a minimal, theoretically grounded diagnostic baseline, while expanding our "Limitation" to acknowledge that other feature-level metrics (like SVCCA) can serve as valid alternatives/extensions.
> > >
> > > **#W3 & #Q1**
> > > Appreciate the chance to elaborate. Recent work [f,g] shows that probability-based or threshold-based MIAs (e.g., **Min-K\% Prob**) are highly sensitive to distributional mismatches, such as temporal or lexical distribution shifts. Relying on raw probabilities, they often conflate distribution shifts with true memorization, artificially inflating AUC scores. While robust shadow-model MIAs like LiRA [h] avoid this pitfall, they remain prohibitive for LLMs. Building on this **empirical intuition**, we theoretically formalize a failure mode: under irreversible catastrophic forgetting, a rebounding MIA AUC is a deceptive artifact of relative probability shifts.
> > >
> > > MIA via Min-K\% Prob evaluates the mean log-probability of a sequence's least likely tokens:
> > > $\mathrm{Min\text{-}K\%\ Prob}(x)=\frac{1}{E}\sum_{x_i\in \mathrm{Min\text{-}K\%}(x)} \log p(x_i\mid x_{<i})$. Crucially, its AUC measures _relative statistical separability_ between sets, not absolute knowledge retention. The erratic rebound in the **irreversible catastrophic** regime is driven by three interacting factors:
> > >
> > > i) **Surface-level sensitivity (Section 5.2)**: As shown in our perturbation model, log-probabilities are highly sensitive to parameter shifts:
> > > $\log p(y\mid x;\theta+\delta\theta)\approx \log p(y\mid x;\theta)+\nabla_\theta \log p(y\mid x;\theta)^\top \delta\theta+O(\|\delta\theta\|^2)$. Accumulated perturbations can completely shatter output logits, even while deeper representations irreversibly drift.
> > >
> > > ii) **Global probability collapse**: Catastrophic unlearning destroys predictive capacity globally. As shown in Appendix Table 6, both forget and retain perplexities (F.Ppl and R.Ppl) explode toward infinity. With uniformly abysmal token probabilities across all data, the baseline Min-K\% statistic collapses and becomes uninformative.
> > >
> > > iii) **Asymmetric relearning recovery**: When targeted relearning is applied, the loss gradient causes F.Ppl to increase significantly faster than R.Ppl (Appendix Table 6). As MIA AUC is a _relative ranking_ metric, this asymmetric recovery forces a new probability gap. The sets become statistically separable again, which artificially inflates the AUC score.
> > >
> > > In short, a rebounding MIA AUC in this regime doesn't indicate true privacy preservation; it just reflects asymmetric restoration of statistical separability during relearning. By identifying this unlearning-specific artifact, our theoretical and empirical analysis grounds prior concerns regarding probability-based or threshold-based MIAs. We'll **explicitly include and extend this formal breakdown in our revision** to thoroughly address this vulnerability.
> > >
> > > [e] SVCCA: Singular Vector Canonical Correlation Analysis for Deep Learning Dynamics and Interpretability. NeurIPS 2017.
> > >
> > > [f] Do Membership Inference Attacks Work on Large Language Models? COLM 2024.
> > >
> > > [g] SoK: Membership Inference Attacks on LLMs are Rushing Nowhere (and How to Fix It). SaTML 2025.
> > >
> > > [h] Membership Inference Attacks From First Principles. S&P 2022.

---

### Official Review · Reviewer_YUKm · 2026-03-23

**Soundness:** 3
**Presentation:** 3
**Significance:** 3
**Originality:** 2
**Overall Recommendation:** 4
**Confidence:** 3

**Summary:**

The paper studies the reliability of machine unlearning in LLMs, asking whether current methods truly remove information or instead suppress it in a way that can later be reversed. This is a relevant question, and the paper makes a convincing case that apparent forgetting can be misleading when the original behavior is easily recovered through limited relearning. Its main contribution is a representation-level evaluation framework based on PCA similarity and shift, together with related statistical analyses. The experiments also introduce a useful taxonomy of four forgetting regimes and show that achieving irreversible unlearning without catastrophic side effects remains challenging.

**Compliance With Llm Reviewing Policy:**

Affirmed.

**Final Justification:**

The authors provided a strong and convincing rebuttal that addresses my main concerns and improves my understanding of the work. As a result, I am more confident in the contribution and raise my score to 4.

**Key Questions For Authors:**

No questions.

**Limitations:**

Yes.

**Strengths And Weaknesses:**

**Strengths**

- The paper addresses an important and timely question: whether current unlearning methods genuinely remove knowledge from LLMs or merely suppress it temporarily.
- Evaluating unlearning through recoverability, rather than relying only on post-unlearning performance, is a convincing and well-motivated choice.
- The representation-level analysis, based on tools such as PCA-based similarity and CKA, is one of the strongest aspects of the paper.

**Weaknesses**

- The paper does not engage enough with prior work on unlearning in vision models and ViT architectures, where notions such as relearning time and apparent unlearning are already well established.
- At times, the paper is more effective at diagnosing the limitations of current unlearning methods than at offering concrete technical advances toward better approaches; as a result, the methodological contribution feels somewhat limited.
- The proposed evaluation protocol is difficult to follow. The number of possible experimental combinations makes it hard to extract the main takeaways and assess some of the conclusions clearly.
- Although the paper makes a strong case against relying solely on task-level metrics, it remains unclear what a practical evaluation standard should look like going forward.

---

> ### Author Rebuttal · Authors · 2026-03-31
>
> Thanks for constructive feedbacks. We're glad that you found **the problem important, recoverability perspective valuable**, and **representation-level analysis insightful**. We address the remaining concerns below.
>
> **#W1**
> We appreciate pointer to work on reversibility in vision models. While our focus is LLMs, we agree that explicitly situating our results in this broader line of work strengthens our work.
>
> In revision, we'll connect more clearly to, e.g.:
>
> - **Apparent unlearning in vision classifiers**, where forget-set accuracy recovers after fine-tuning on retain data [a].
>
> - **Diffusion instability**, where erased concepts re-emerge in text-to-image models after downstream fine-tuning [b].
>
> Our contribution is complementary: we provide the **first systematic, representation-level study of reversibility in LLM unlearning**, integrating task-level recovery, representation drift, and sample efficiency under controlled relearning, and we introduce a unified diagnostic toolkit (PCA similarity/shift, CKA, FIM, mean PCA distance) for LLMs.
>
> **#W2**
> Our focus is methodological rather than algorithmic. Beyond showing task-level metrics are insufficient, we offer a concrete framework for unlearning evaluation:
>
> - Introduce a **representation-level diagnostic toolkit** and a **summary metric** that quantifies representational drift and predicts reversibility under a fixed relearning protocol.
>
> - Define & empirically validate a **taxonomy of four forgetting regimes**, and map standard unlearning methods, hyperparameters, and request volumes into these regimes.
>
> - Develop a **perturbation-based theoretical model** that links weight updates to changes in feature geometry and explains why task-level collapse can coexist with largely intact internal representations.
>
> - Identify a **seemingly irreversible, non‑catastrophic regime** under GA+GD+WAGLE with constrained relearning, showing that genuinely targeted, durable erasure is possible but tightly constrained by method and hyperparameters.
>
> - Provide **practical guidance** for future unlearning methods, including using mean PCA distance or Fisher spectra to tune learning rates/request counts and to distinguish superficial suppression from genuine erasure.
>
> Altogether, they form a concrete, technically grounded framework for analyzing and comparing unlearning methods, and they expose failure modes that task-level metrics alone systematically miss. We view this as groundwork for designing more robust unlearning ones.
>
> **#W3**
> Our evaluation is broad, as reversibility in LLM unlearning depends jointly on the unlearning method, relearning setting, sample budget, attack type, and representation-level behavior. The intent is not to enumerate configurations, but to stress-test methods and extract **a small set of robust conclusions**.
>
> Across these settings, we consistently find that:
>
> - **Task-level forgetting is misleading**: large post‑unlearning drops in accuracy or MIA AUC do not reliably indicate durable deletion.
>
> - **Relearning has distinct sample efficiencies**: how easily forgotten knowledge returns depends strongly on relearning data sources.
>
> - **Representation-level diagnostics are more informative for reversibility**: Our toolkit clearly separates superficial suppression from genuine representational change.
>
> - **Irreversible yet non‑catastrophic forgetting is rare and fragile**, highlighting the need for stronger evaluation protocols beyond standard task metrics.
>
> We agree that the current presentation can be dense. We'll i) reorganize experiments by question, not by configuration, ii) state main takeaways at each subsection, with a brief summary of which ablations support it, and iii) move secondary settings and extended ablations to Appendix, with clear cross-references.
>
> **#W4**
> Beyond critiquing task-level metrics, we aim to **outline a practical evaluation standard** for future unlearning work. We argue that no single metric is reliable; a robust protocol should jointly assess: **Task-level behavior**, **Relearning robustness**, and **Representation-level change**.
>
> Concretely, it requires:
>
> - **Task-level metrics** to capture the immediate behavioral effect of unlearning.
>
> - **Relearning-based robustness tests** to measure how sample‑efficiently "forgotten" knowledge can be recovered under constrained fine-tuning on different data sources.
>
> - **Representation-level diagnostics** to detect whether internal representations remain close to the pre‑unlearning model.
>
> Our GA+GD+WAGLE results fit this template and show how behavior, relearning outcomes, and representational drift jointly characterize unlearning quality. In revision, we'll make this **three-part evaluation standard** explicit as a forward-looking recommendation.
>
> [a] From Dormant to Deleted: Tamper-Resistant Unlearning Through Weight-Space Regularization. NeurIPS 2025.
>
> [b] The Illusion of Unlearning: The Unstable Nature of Machine Unlearning in Text-to-Image Diffusion Models. CVPR 2025.

---

> > ### Author Rebuttal · Reviewer_YUKm · 2026-04-02
> >
> > I thank the authors for their rebuttal. I have carefully read their responses as well as the other reviews. I prefer to keep my original score. The paper would benefit from a clearer positioning with respect to the existing literature, as well as a more precise definition of the key takeaways and novel contributions. A number of settings and metrics are introduced, often building on prior work, but it is not entirely clear to me whether this substantially advances the literature on unlearning. That said, I do appreciate the effort put into the work, and I recognize that its innovative aspect lies in the systematic study of this phenomenon on LLMs.

---

> > > ### Author Response · Authors · 2026-04-03
> > >
> > > We sincerely thank you for timely follow-up and for recognizing the value of our systematic study of this phenomenon in LLMs. Appreciate another round of opportunity to elaborate. For remaining concerns, we clarify our positioning, core contribution, and key takeaways below.
> > >
> > > **1. Positioning: Moving from Behavioral to Mechanistic Reversibility**
> > >
> > > While prior work (lines 653--709, e.g., [c,d]) and [a] observe reversibility in LLMs, it primarily treat it as a behavioral phenomenon, showing that unlearned knowledge can often be recovered via fine-tuning / prompt attacks. Our work fundamentally advances this literature by providing the first _mechanistic_ and _representational explanation_ for _why_ reversibility occurs. By introducing a perturbation-based theoretical model (Sec. 5) paired with a unified diagnostic toolkit (Sec. 4.2.2.), **we bridge the gap between weight-level updates, representational drift, and task-level outcomes**.
> > >
> > > **2. Core Contribution: The Necessity of a Unified Diagnostic Suite**
> > >
> > > Prior work relies heavily on task-level metrics (e.g., accuracy, probability-based MIA) that are easily deceived by output-layer perturbations. Our technical contribution is not just combining existing tools, but establishing theoretically and empirically _why the full suite is necessary_ to correctly diagnose unlearning:
> > >
> > > - **FIM** tracks loss landscape changes, distinguishing temporary parameter suppression from permanent erasure.
> > >
> > > - **CKA** measures whether the activation subspace remains intact or has been structurally fractured by distributed errors.
> > >
> > > - **PCA Similarity & Shift** must be used _jointly_ to separate recoverable translational displacement from irreversible changes in principal directions.
> > >
> > > Relying on task metrics or any single representation metric risks conflating superficial, reversible suppression with genuine representational erasure, e.g., by mistaking a reversible shift for an irreversible collapse. More importantly, our contribution is not simply to introduce these diagnostics, but to establish, both theoretically and empirically, why each is necessary for reliably characterizing reversibility in LLM unlearning.
> > >
> > > **3. Key Takeaways Advancing the Unlearning Literature**
> > >
> > >
> > > Our systematic analysis yields concrete insights that challenge prevailing assumptions in the unlearning literature:
> > >
> > > - **Task-Level Collapse is Deceptive**: Unlearning often induces small perturbations near output layers. Our theoretical analysis (Sec. 5.2) explains why this causes drastic task-level failure (e.g., sharp drops in accuracy or MIA) while preserving deep representations, explaining why supposedly "deleted" knowledge is easily recovered.
> > >
> > > - **The "Hierarchy" of Relearning Efficiency**: Reversibility is not binary. The speed of recovery depends on data source (forget set > retain set > unrelated data), showing that unlearning often leaves latent knowledge primed for reactivation rather than destroyed.
> > >
> > > - **Irreversible yet Non-Catastrophic Forgetting is Very Rare**: Even advanced methods struggle to achieve genuine erasure without collateral damage. We identify a seemingly successful case (GA+GD+WAGLE under strictly constrained relearning), proving that targeted, durable erasure is possible but currently highly fragile.
> > >
> > > - **A Robust New Standard for Evaluation**: Relearning is the ultimate probe for unlearning robustness. Inference-time attacks (e.g., jailbreaks, prompting) fail once a model enters catastrophic forgetting, but relearning can uniquely verify whether representations are permanently destroyed or just temporarily dormant.
> > >
> > > We hope this clarifies how our representational framework and findings substantially advance the understanding and evaluation of LLM unlearning. We'll ensure these distinctions and takeaways are explicitly formalized in our final version.
> > >
> > > We would greatly appreciate it if you could kindly adjust the score in the final decision. Once again, we sincerely thank you for the thoughtful feedback and for helping improve our work.
> > >
> > > [c] Unlearning or Obfuscating? Jogging the Memory of Unlearned LLMs via Benign Relearning. ICLR 2025.
> > >
> > > [d] Erase or Hide? Suppressing Spurious Unlearning Neurons for Robust Unlearning. ICLR 2026.

---

### Decision · Program_Chairs · 2026-04-30

**Decision:**

Accept (regular)

**Comment:**

The paper shows that task-level metrics can make LLM unlearning look successful when the knowledge is only suppressed, and offers a representation-level toolkit and a four-regime taxonomy to diagnose this. Reviewers found the question important and the empirical study broad and convincing. The recurring reservation was novelty: the individual tools are established, and the theoretical section claimed more rigour than it delivers.

The rebuttal handled this appropriately. The new experiments show the taxonomy holds across unlearning paradigms the paper had not originally covered, and the authors agreed to recast the theory as a heuristic motivation rather than a formal result. After discussion all four reviewers support acceptance, and so do I.

For camera-ready, please incorporate the rebuttal experiments and the reframing of Section 5, and tighten the related-work positioning as discussed.

Congratulations!